# Deceptive Fairness Attacks on Graphs via Meta Learning

**Jian Kang[1], Yinglong Xia[2], Ross Maciejewski[3], Jiebo Luo[1], Hanghang Tong[4]**
[1]University of Rochester, {jian.kang@, jluo@cs.}rochester.edu
[2]Meta, yxia@meta.com
[3]Arizona State University, rmacieje@asu.edu
[4]University of Illinois Urbana-Champaign, htong@illinois.edu

## Abstract

We study deceptive fairness attacks on graphs to answer the following question: *How can we achieve poisoning attacks on a graph learning model to exacerbate the bias deceptively?* We answer this question via a bi-level optimization problem and propose a meta learning-based framework named Fate. Fate is broadly applicable with respect to various fairness definitions and graph learning models, as well as arbitrary choices of manipulation operations. We further instantiate Fate to attack statistical parity or individual fairness on graph neural networks. We conduct extensive experimental evaluations on real-world datasets in the task of semi-supervised node classification. The experimental results demonstrate that Fate could amplify the bias of graph neural networks with or without fairness consideration while maintaining the utility on the downstream task. We hope this paper provides insights into the adversarial robustness of fair graph learning and can shed light on designing robust and fair graph learning in future studies.

## 1 Introduction

Algorithmic fairness on graphs has received much research attention (Bose & Hamilton, 2019; Dai & Wang, 2021; Kang et al., 2020; Li et al., 2021; Kang et al., 2022). Despite its substantial progress, existing studies mostly assume the benevolence of input graphs and aim to ensure that the bias would not be perpetuated or amplified in the learning process. However, malicious activities in the real world are commonplace. For example, consider a financial fraud detection system which utilizes a transaction network to classify whether a bank account is fraudulent or not (Zhang et al., 2017; Wang et al., 2019). An adversary may manipulate the transaction network (e.g., malicious banker with access to the demographic and transaction data), so that the graph-based fraud detection model would exhibit unfair classification results with respect to people of different demographic groups. Consequently, a biased fraud detection model may infringe civil liberty to certain financial activities and impact the well-being of an individual negatively (Bureau, 2022). It would also make the graph learning model fail to provide the same quality of service to individual(s) of certain demographic groups, causing the financial institutions to lose business in the communities of the corresponding demographic groups. Thus, it is critical to understand how resilient a graph learning model is with respect to adversarial attacks on fairness, which we term as *fairness attacks*.

Fairness attack has not been well studied, and sporadic literature often follows two strategies. The first strategy is adversarial data point injection, which is often designed for tabular data rather than graphs (Solans et al., 2021; Mehrabi et al., 2021; Chhabra et al., 2021; Van et al., 2022). However, in addition to only inject adversarial node(s), it is crucial to connect the injected adversarial node(s) to nodes in the original graph, which requires non-trivial modifications to existing methods, to effectively attack graph learning models. Another strategy is adversarial edge injection, which to date only attacks the group fairness of graph neural networks (Hussain et al., 2022). It is thus crucial to study how to attack different fairness definitions for a variety of graph learning models.

To achieve this goal, we study deceptive fairness attacks on graphs. We formulate it as a bi-level optimization, where the lower-level problem optimizes a task-specific loss function to maintain the performance of the downstream learning task and enforces budgeted perturbations to make the fairness attacks deceptive, and the upper-level problem leverages the supervision to modify the input

graph and maximize the bias function corresponding to a user-defined fairness definition. To solve the bi-level optimization problem, we propose a meta learning-based solver (FATE), whose key idea is to compute the meta-gradient of the upper-level bias function with respect to the input graph to guide the fairness attacks. Compared with existing works, our proposed FATE framework has two major advantages. First, it is capable of attacking *any* fairness definition on *any* graph learning model, as long as the corresponding bias function and the task-specific loss function are differentiable. Second, it is equipped with the ability for either continuous or discretized poisoning attacks on the graph topology. We also briefly discuss its ability for poisoning attacks on node features in a later section.

The major contributions of this paper are: (A) **Problem definition.** We study the problem of deceptive fairness attacks on graphs. Based on the definition, we formulate it as a bi-level optimization problem, whose key idea is to maximize a bias function in the upper level while minimizing a task-specific loss function for a graph learning task in the lower level; (B) **Attacking framework.** We propose an end-to-end attacking framework named FATE. It learns a perturbed graph topology via meta learning, such that the bias with respect to the learning results trained with the perturbed graph will be amplified; (C) **Empirical evaluation.** We conduct experiments on three benchmark datasets to demonstrate the efficacy of our proposed FATE framework in amplifying the bias while being the most deceptive method (i.e., achieving the highest micro F1 score) on semi-supervised node classification.

## 2 PRELIMINARIES AND PROBLEM DEFINITION

**A – Notations.** We use bold upper-case, bold lower-case, and calligraphic letters for matrix, vector, and set, respectively (e.g., $\mathbf{A}$, $\mathbf{x}$, $\mathcal{G}$). $^T$ denotes matrix/vector transpose (e.g., $\mathbf{x}^T$ is the transpose of $\mathbf{x}$). Matrix/vector indexing is similar to NumPy in Python, e.g., $\mathbf{A}[i, j]$ is the entry of $\mathbf{A}$ at the $i$-th row and $j$-th column; $\mathbf{A}[i, :]$ and $\mathbf{A}[:, j]$ are the $i$-th row and $j$-th column of $\mathbf{A}$, respectively.

**B – Algorithmic fairness.** The general principle of algorithmic fairness is to ensure the learning results would not favor one side or another.[1] Among several fairness definitions that follow this principle, group fairness (Feldman et al., 2015; Hardt et al., 2016) and individual fairness (Dwork et al., 2012) are the most widely studied ones. Group fairness splits the entire population into multiple demographic groups by a sensitive attribute (e.g., gender) and ensure the parity of a statistical property among learning results of those groups. For example, statistical parity, a classic group fairness definition, guarantees the statistical independence between the learning results (e.g., predicted labels) and the sensitive attribute (Feldman et al., 2015). Individual fairness suggests that similar individuals should be treated similarly. It is often formulated as a Lipschitz inequality such that distance between the learning results of two data points should be no larger than the difference between these two data points (Dwork et al., 2012). More details are provided in Appendix H.

**C – Problem definition.** Existing work (Hussain et al., 2022) for fairness attacks on graphs randomly injects adversarial edges so that the disparity between the learning results of two different demographic groups would be amplified. However, it suffers from three major limitations. (1) First, it only attacks statistical parity while overlooking other fairness definitions (e.g., individual fairness (Dwork et al., 2012)). (2) Second, it only considers adversarial edge injection, excluding other manipulations like edge deletion or reweighting. Hence, it is essential to investigate the possibility to attack other fairness definitions on real-world graphs with an arbitrary choice of manipulation operations. (3) Third, it does not consider the utility of graph learning models when attacking fairness, resulting in performance degradation in the downstream tasks. However, an institution that applies the graph learning models are often utility-maximizing (Liu et al., 2018; Baumann et al., 2022). Thus, a performance degradation in the utility would make the fairness attacks not deceptive from the perspective of a utility-maximizing institution.

In this paper, we seek to overcome the aforementioned limitations. To be specific, given an input graph, an optimization-based graph learning model, and a user-defined fairness definition, we aim to learn a modified graph such that a bias function of the corresponding fairness definition would be maximized for *effective* fairness attacks, while minimizing the task-specific loss function with respect to the graph learning model for *deceptive* fairness attacks.

Formally, we define the problem of deceptive fairness attacks on graphs. We are given (1) an undirected graph $\mathcal{G} = \{\mathbf{A}, \mathbf{X}\}$, (2) a task-specific loss function $l(\mathcal{G}, \mathbf{Y}, \Theta_{\mathrm{vic}}, \theta_{\mathrm{vic}})$, where $\mathbf{Y}$ is the graph learning model output, $\Theta_{\mathrm{vic}}$ is the set of learnable variables of the victim model targeted for

---

[1] https://www.merriam-webster.com/dictionary/fairness

attacking, and $\theta_{\text{vic}}$ is the set of hyperparameters of the victim model, (3) a bias function $b(\mathbf{Y}, \Theta^*, \mathbf{F})$, where $\Theta^*_{\text{vic}} = \arg\min_{\Theta_{\text{vic}}} l(\mathcal{G}, \mathbf{Y}, \Theta_{\text{vic}}, \theta_{\text{vic}})$, and $\mathbf{F}$ is the matrix that contains auxiliary fairness-related information (e.g., sensitive attribute values of all nodes in $\mathcal{G}$ for group fairness, pairwise node similarity matrix for individual fairness), and (4) an integer budget $B$. And our goal is to learn a poisoned graph $\widetilde{\mathcal{G}} = \left\{ \widetilde{\mathbf{A}}, \widetilde{\mathbf{X}} \right\}$, such that (1) $d\left(\mathcal{G}, \widetilde{\mathcal{G}}\right) \leq B$, where $d\left(\mathcal{G}, \widetilde{\mathcal{G}}\right)$ is the distance between the input graph $\mathcal{G}$ and poisoned graph $\widetilde{\mathcal{G}}$ (e.g., the total weight of perturbed edges $\left\| \mathbf{A} - \widetilde{\mathbf{A}} \right\|_{1,1} = \left\| \text{vec}\left(\mathbf{A} - \widetilde{\mathbf{A}}\right) \right\|_1$), (2) the bias function $b(\mathbf{Y}, \Theta^*_{\text{vic}}, \mathbf{F})$ is maximized for effectiveness, and (3) the task-specific loss function $l\left(\widetilde{\mathcal{G}}, \mathbf{Y}, \Theta_{\text{vic}}, \theta_{\text{vic}}\right)$ is minimized for deceptiveness.

# 3 METHODOLOGY

In this section, we first formulate the problem of deceptive fairness attacks on graphs as a bi-level optimization problem, followed by a generic meta learning-based solver named FATE.

## 3.1 PROBLEM FORMULATION

Given an input graph $\mathcal{G} = \{\mathbf{A}, \mathbf{X}\}$ with adjacency matrix $\mathbf{A}$ and node feature matrix $\mathbf{X}$, an attacker aims to learn a poisoned graph $\widetilde{\mathcal{G}} = \left\{ \widetilde{\mathbf{A}}, \widetilde{\mathbf{X}} \right\}$, such that the graph learning model will be maximally biased when trained on $\widetilde{\mathcal{G}}$. In this work, we consider the following settings for the attacker.

**The goal of the attacker.** The attacker aims to amplify the bias of the graph learning results output by a victim graph learning model. And the bias to be amplified is a choice made by the attacker based on which fairness definition the attacker aims to attack.

**The knowledge of the attacker.** Following similar settings in (Hussain et al., 2022), we assume the attacker has access to the adjacency matrix, the feature matrix of the input graph, and the sensitive attribute of all nodes in the graph. For a (semi-)supervised learning problem, we assume that the ground-truth labels of the training nodes are also available to the attacker. For example, for a graph-based financial fraud detection problem, the malicious banker may have access to the demographic information (i.e., sensitive attribute) of the account holders and also know whether some bank accounts are fraudulent or not, which are the ground-truth labels for training nodes. Similar to (Zügner et al., 2018; Zügner & Günnemann, 2019; Hussain et al., 2022), the attacker has no knowledge about the parameters $\Theta_{\text{vic}}$ and $\theta_{\text{vic}}$ of the victim model. Instead, the attacker will perform a gray-box attack by attacking a surrogate graph learning model with learnable parameters $\Theta_{\text{sur}}$ and hyperparameters $\theta_{\text{sur}}$.

**The capabilitiy of the attacker.** The attacker is able to perturb up to $B$ edges/features in the graph (i.e., the entry-wise matrix norms $\left\| \mathbf{A} - \widetilde{\mathbf{A}} \right\|_{1,1} \leq B$ and/or $\left\| \mathbf{X} - \widetilde{\mathbf{X}} \right\|_{1,1} \leq B$).

Based on that, we formulate our problem as a bi-level optimization problem as follows.

$$\widetilde{\mathcal{G}} = \arg\max_{\mathcal{G}} \; b(\mathbf{Y}, \Theta^*_{\text{sur}}, \mathbf{F}) \quad \text{s.t.} \quad \Theta^*_{\text{sur}} = \arg\min_{\Theta_{\text{sur}}} l(\mathcal{G}, \mathbf{Y}, \Theta_{\text{sur}}, \theta_{\text{sur}}), \; d\left(\mathcal{G}, \widetilde{\mathcal{G}}\right) \leq B \qquad (1)$$

where the lower-level problem learns an optimal surrogate graph learning model $\Theta^*_{\text{sur}}$ by minimizing $l(\mathcal{G}, \mathbf{Y}, \Theta_{\text{sur}}, \theta_{\text{sur}})$, the upper-level problem finds a poisoned graph $\widetilde{\mathcal{G}}$ that could maximize a bias function $b(\mathbf{Y}, \Theta^*_{\text{sur}}, \mathbf{F})$ for the victim graph learning model and the distance between the input graph, and the poisoned graph $d\left(\mathcal{G}, \widetilde{\mathcal{G}}\right)$ is constrained to satisfy the setting about the budgeted attack. Note that Eq. 1 is applicable to attack *any* fairness definition on *any* graph learning model, as long as the bias function $b(\mathbf{Y}, \Theta^*_{\text{sur}}, \mathbf{F})$ and the loss function $l(\mathcal{G}, \mathbf{Y}, \Theta_{\text{sur}}, \theta_{\text{sur}})$ are differentiable.

**A – Lower-level optimization problem.** A wide spectrum of graph learning models are essentially solving an optimization problem. For example, graph convolutional network (GCN) (Kipf & Welling, 2017) learns the node representation by aggregating information from its neighborhood and performing nonlinear transformation with model parameters and an activation function. The lower-level optimization problem for an $L$-layer GCN aims to learn the set of model parameters $\Theta^* = \{\mathbf{W}^{(i)} | i = 1, \ldots, L\}$, where $\mathbf{W}^{(i)}$ is the weight matrix in the $i$-th layer, that could minimize a task-specific loss function (e.g., cross-entropy for node classification). For more examples of graph learning models from the optimization perspective, please refers to Appendix A.

**B – Upper-level optimization problem.** To attack the fairness aspect of a graph learning model, we aim to maximize a differentiable bias function $b\left(\mathbf{Y}, \Theta_{\text{sur}}^{*}, \mathbf{F}\right)$ with respect to a user-defined fairness definition in the upper-level optimization problem. For example, for statistical parity (Feldman et al., 2015), the fairness-related auxiliary information matrix $\mathbf{F}$ can be defined as the one-hot demographic membership matrix, where $\mathbf{F}[i, j] = 1$ if and only if node $i$ belongs to $j$-th demographic group. Then the statistical parity is equivalent to the statistical independence between the learning results $\mathbf{Y}$ and $\mathbf{F}$. Based on that, existing studies propose several differentiable measurements of the statistical dependence between $\mathbf{Y}$ and $\mathbf{F}$ as the bias function. For example, Bose et al. (Bose & Hamilton, 2019) use mutual information $I(\mathbf{Y}; \mathbf{F})$ as the bias function; Prost et al. (Prost et al., 2019) define the bias function as the Maximum Mean Discrepancy $MMD\left(\mathcal{Y}_0, \mathcal{Y}_1\right)$ between the learning results of two different demographic groups $\mathcal{Y}_0$ and $\mathcal{Y}_1$.

### 3.2 The Fate Framework

To solve Eq. 1, we propose a generic attacking framework named Fate (Deceptive Fairness Attacks on Graphs via Meta Learning) to learn the poisoned graph. The key idea is to view Eq. 1 as a meta learning problem, which aims to find suitable hyperparameter settings for a learning task (Bengio, 2000), and treat the graph $\mathcal{G}$ as a hyperparameter. With that, we learn the poisoned graph $\widetilde{\mathcal{G}}$ using the meta-gradient of the bias function $b\left(\mathbf{Y}, \Theta_{\text{sur}}^{*}, \mathbf{F}\right)$ with respect to $\mathcal{G}$. In the following, we introduce two key parts of Fate, including meta-gradient computation and graph poisoning with meta-gradient.

**A – Meta-gradient computation.** The key term to learn the poisoned graph is the meta-gradient of the bias function with respect to the graph $\mathcal{G}$. Before computing the meta-gradient, we assume that the lower-level optimization problem converges in $T$ epochs. Thus, we first pre-train the lower-level optimization problem by $T$ epochs to obtain the optimal model $\Theta_{\text{sur}}^{*} = \Theta_{\text{sur}}^{(T)}$ before computing the meta-gradient. The training of the lower-level optimization problem can also be viewed as a dynamic system with $\Theta_{\text{sur}}^{(t+1)} = \text{opt}^{(t+1)}\left(\mathcal{G}, \Theta_{\text{sur}}^{(t)}, \theta_{\text{sur}}, \mathbf{Y}\right)$, $\forall t \in \{1, \ldots, T\}$, where $\Theta_{\text{sur}}^{(1)}$ refers to $\Theta_{\text{sur}}$ at initialization, and $\text{opt}^{(t+1)}(\cdot)$ is an optimizer that minimizes the lower-level loss function $l\left(\mathcal{G}, \mathbf{Y}, \Theta_{\text{sur}}^{(t)}, \theta\right)$ at $(t+1)$-th epoch. From the perspective of the dynamical system, by applying the chain rule and unrolling the training of lower-level problem , the meta-gradient $\nabla_{\mathcal{G}} b$ can be written as $\nabla_{\mathcal{G}} b = \nabla_{\mathcal{G}} b\left(\mathbf{Y}, \Theta_{\text{sur}}^{(T)}, \mathbf{F}\right) + \sum_{t=0}^{T-2} A_t B_{t+1} \ldots B_{T-1} \nabla_{\theta^{(T)}} b\left(\mathbf{Y}, \Theta_{\text{sur}}^{(T)}, \mathbf{F}\right)$, where $A_t = \nabla_{\mathcal{G}} \Theta_{\text{sur}}^{(t+1)}$ and $B_t = \nabla_{\Theta_{\text{sur}}^{(t)}} \Theta_{\text{sur}}^{(t+1)}$. However, it is computationally expensive in both time and space to compute the meta-gradient. To further speed up the computation, we adopt a first-order approximation of the meta-gradient (Finn et al., 2017) and simplify the meta-gradient as

$$\nabla_{\mathcal{G}} b \approx \nabla_{\Theta_{\text{sur}}^{(T)}} b\left(\mathbf{Y}, \Theta_{\text{sur}}^{(T)}, \mathbf{F}\right) \cdot \nabla_{\mathcal{G}} \Theta_{\text{sur}}^{(T)} \tag{2}$$

Since the input graph is undirected, the derivative of the symmetric adjacency matrix $\mathbf{A}$ can be computed as follows by applying the chain rule of a symmetric matrix (Kang et al., 2020).

$$\nabla_{\mathbf{A}} b \leftarrow \nabla_{\mathbf{A}} b + (\nabla_{\mathbf{A}} b)^T - \text{diag}\left(\nabla_{\mathbf{A}} b\right) \tag{3}$$

For the feature matrix $\mathbf{X}$, its derivative equals to the partial derivative since $\mathbf{X}$ is often asymmetric.

**B – Graph poisoning with meta-gradient.** After computing the meta-gradient of the bias function $\nabla_{\mathcal{G}} b$, we aim to poison the input graph guided by $\nabla_{\mathcal{G}} b$. We introduce two poisoning strategies: (1) continuous poisoning and (2) discretized poisoning.

*Continuous poisoning attack.* The continuous poisoning attack is straightforward by reweighting edges in the graph. We first compute the meta-gradient of the bias function $\nabla_{\mathbf{A}} b$, then use it to poison the input graph in a gradient descent-based updating rule as follows.

$$\mathbf{A} \leftarrow \mathbf{A} - \eta \nabla_{\mathbf{A}} b \tag{4}$$

where $\eta$ is a learning rate to control the magnitude of the poisoning attack. Suppose we attack the topology for $k$ attacking steps with budgets $\delta_1, \ldots, \delta_k$ and $\sum_{i=1}^{k} \delta_i = B$. In the $i$-th attacking step, the learning rate should satisfy $\eta \leq \frac{\delta_i}{\|\nabla_{\mathbf{A}}\|_{1,1}}$ to ensure that constraint on the budgeted attack

*Discretized poisoning attack.* The discretized poisoning attack aims to select a set of edges to be added/deleted. It is guided by a poisoning preference matrix defined as follows.

$$\nabla_{\mathbf{A}} = (\mathbf{1} - 2\mathbf{A}) \circ \nabla_{\mathbf{A}} b \tag{5}$$

where $\mathbf{1}$ is an all-one matrix with the same dimension as $\mathbf{A}$ and $\circ$ denotes the Hadamard product. A large positive $\nabla_{\mathbf{A}}\left[i, j\right]$ indicates strong preference in adding an edge if nodes $i$ and $j$ are not connected (i.e., positive $\nabla_{\mathbf{A}}b\left[i, j\right]$, positive $\left(\mathbf{1} - 2\mathbf{A}\right)\left[i, j\right]$) or deleting an edge if nodes $i$ and $j$ are connected (i.e., negative $\nabla_{\mathbf{A}}b\left[i, j\right]$, negative $\left(\mathbf{1} - 2\mathbf{A}\right)\left[i, j\right]$). Then, one strategy to find the set of edges $\mathcal{E}_{\text{attack}}$ to be added/deleted can be greedy selection.

$$\mathcal{E}_{\text{attack}} = \text{topk}\left(\nabla_{\mathbf{A}}, \delta_i\right) \tag{6}$$

where $\text{topk}\left(\nabla_{\mathbf{A}}, \delta_i\right)$ selects $\delta_i$ entries with highest preference score in $\nabla_{\mathbf{A}}$ in the $i$-th attacking step. Note that, if we only want to add edges without any deletion, all negative entries in $\nabla_{\mathbf{A}}b$ should be zeroed out before computing Eq. 5. Likewise, if edges are only expected to be deleted, all positive entries should be zeroed out.

*Remarks.* Poisoning node feature matrix $\mathbf{X}$ follows the same steps as poisoning adjacency matrix $\mathbf{A}$ without applying Eq. 3. And we briefly discuss an alternative edge selection strategy for discretized poisoning attacks via sampling in Appendix G.

**C – Overall framework.** FATE generally works as follows. (1) We first pre-train the surrogate graph learning model and get the corresponding learning model $\Theta_{\text{sur}}^{(T)}$ as well as the learning results $\mathbf{Y}^{(T)}$. (2) Then we compute the meta gradient of the bias function using Eqs. 2 and 3. (3) Finally, we perform the discretized poisoning attack (Eqs. 5 and 6) or continuous poisoning attack (Eq. 4). A detailed pseudo-code of FATE is provided in Appendix B.

**D – Limitations.** Since FATE leverages the meta-gradient to poison the input graph, it requires the bias function $b\left(\mathbf{Y}, \Theta_{\text{sur}}^{(T)}, \mathbf{F}\right)$ to be differentiable in order to calculate the meta-gradient $\nabla_{\mathcal{G}}b$. In Sections 4 and 5, we present two carefully chosen bias functions for FATE. And we leave it for future work on exploring the ability of FATE in attacking other fairness definitions. Moreover, though the meta-gradient can be efficiently computed via auto-differentiation in modern deep learning packages (e.g., PyTorch[2], TensorFlow[3]), it requires $O(n^2)$ space complexity when attacking fairness via edge flipping. It is still a challenging open problem on how to efficiently compute the meta-gradient in terms of space. One possible remedy might be a low-rank approximation on the perturbation matrix formed by $\mathcal{E}_{\text{attack}}$. Since the difference between the benign graph and poisoned graph are often small and budgeted ($d\left(\mathcal{G}, \widetilde{\mathcal{G}}\right) \leq B$), it is likely that the edge manipulations may be around a few set of nodes, which makes the perturbation matrix to be an (approximately) low-rank matrix.

## 4 INSTANTIATION #1: STATISTICAL PARITY ON GRAPH NEURAL NETWORKS

Here, we instantiate FATE framework by attacking statistical parity on graph neural networks in a binary node classification problem with a binary sensitive attribute. We briefly discuss how to choose (A) the surrogate graph learning model used by the attacker, (B) the task-specific loss function in the lower-level optimization problem and (C) the bias function in the upper-level optimization problem.

**A – Surrogate graph learning model.** We assume that the surrogate model is a 2-layer linear GCN (Wu et al., 2019) with different hidden dimensions and model parameters at initialization.

**B – Lower-level loss function.** We consider a semi-supervised node classification task for the graph neural network to be attacked. Thus, the lower-level loss function is chosen as the cross entropy between the ground-truth label and the predicted label: $l\left(\mathcal{G}, \mathbf{Y}, \Theta_{\text{sur}}, \theta_{\text{sur}}\right) = \frac{1}{|\mathcal{V}_{\text{train}}|}\sum_{i \in \mathcal{V}_{\text{train}}}\sum_{j=1}^{c} y_{i,j}\ln\widehat{y}_{i,j}$, where $\mathcal{V}_{\text{train}}$ is the set of training nodes with ground-truth labels with $|\mathcal{V}_{\text{train}}|$ being its cardinality, $c$ is the number of classes, $y_{i,j}$ is a binary indicator of whether node $i$ belongs to class $j$ and $\widehat{y}_{i,j}$ is the prediction probability of node $i$ belonging to class $j$.

**C – Upper-level bias function.** We aim to attack statistical parity in the upper-level problem, which asks the predicted label $\tilde{y}$ to follow $\text{P}\left[\tilde{y} = 1\right] = \text{P}\left[\tilde{y} = 1|s = 1\right]$. Then the bias function is defined as $b\left(\mathbf{Y}, \Theta_{\text{sur}}^{*}, \mathbf{S}\right) = |\text{P}\left[\tilde{y} = 1\right] - \text{P}\left[\tilde{y} = 1|s = 1\right]|$. Suppose $p\left(\widehat{y}_{i,1}\right)$ is the probability density function (PDF) of $\widehat{y}_{i,1}$ for any node $i$ and $p\left(\widehat{y}_{i,1}|s = 1\right)$ is the PDF of $\widehat{y}_{i,1}$ for any node $i$ belong to the demographic group with sensitive attribute value $s = 1$. We observe that $\text{P}\left[\hat{y} = 1\right]$ and $\text{P}\left[\hat{y} = 1|s = 1\right]$ are equivalent to the complementary cumulative distribution functions (CDF)

---

[2]https://pytorch.org/
[3]https://www.tensorflow.org/

of $p\left(\widehat{y}_{i,1} > \frac{1}{2}\right)$ and $p\left(\widehat{y}_{i,1} > \frac{1}{2}|s=1\right)$, respectively. For differentiable estimation of $\mathrm{P}\left[\tilde{y}=1\right]$ and $\mathrm{P}\left[\tilde{y}=1|s=1\right]$, we use kernel density estimation (KDE) for $p\left(\widehat{y}_{i,1} > \frac{1}{2}\right)$ and $p\left(\widehat{y}_{i,1} > \frac{1}{2}|s=1\right)$.

**Definition 1** *(Kernel density estimation (Chen, 2017)) Given a set of $n$ IID samples $\{x_1, \ldots, x_n\}$ drawn from a distribution with an unknown probability density function (PDF) $f$, the kernel density estimation of $f$ at point $\tau$ is defined as follows.*

$$\widetilde{f}\left(\tau\right) = \frac{1}{na} \sum_{i=1}^{n} f_k \left(\frac{\tau - x_i}{a}\right) \tag{7}$$

*where $\widetilde{f}$ is the estimated PDF, $f_k$ is the kernel function and $a$ is a non-negative bandwidth.*

Moreover, we assume the kernel function in KDE is the Gaussian kernel $f_k\left(x\right) = \frac{1}{\sqrt{2\pi}}e^{-x^2/2}$. However, computing the complementary CDF of a Gaussian distribution is non-trivial. Following (Cho et al., 2020), we leverage a tractable approximation of the Gaussian Q-function as follows.

$$Q(\tau) = F_k\left(\tau\right) = \int_{\tau}^{\infty} f_k(x)dx \approx e^{-\alpha\tau^2 - \beta\tau - \gamma} \tag{8}$$

where $f_k\left(x\right) = \frac{1}{\sqrt{2\pi}}e^{-x^2/2}$ is a Gaussian distribution with zero mean, $\alpha = 0.4920$, $\beta = 0.2887$, $\gamma = 1.1893$ (López-Benítez & Casadevall, 2011). How to estimate $\mathrm{P}\left[\hat{y}=1\right]$ is as follows.

- For any node $i$, get its prediction probability $\widehat{y}_{i,1}$ with respect to class 1;
- Estimate the complementary CDF $\mathrm{P}\left[\tilde{y}=1\right]$ using a Gaussian KDE with bandwidth $a$ by $\mathrm{P}\left[\tilde{y}=1\right] = \frac{1}{n}\sum_{i=1}^{n} \exp\left(-\alpha\left(\frac{0.5-\widehat{y}_{i,1}}{a}\right)^2 - \beta\left(\frac{0.5-\widehat{y}_{i,1}}{a}\right) - \gamma\right)$, where $\alpha = 0.4920$, $\beta = 0.2887$, $\gamma = 1.1893$ and $\exp(x) = e^x$.

Note that $\mathrm{P}\left[\tilde{y}=1|s=1\right]$ can be estimated with a similar procedure with minor modifications. The only modifications needed are: (1) get the prediction probability of nodes with $s=1$ and (2) compute the CDF using the Gaussian Q-function over nodes with $s=1$ rather than all nodes in the graph.

## 5 INSTANTIATION #2: INDIVIDUAL FAIRNESS ON GRAPH NEURAL NETWORKS

We provide another instantiation of FATE framework by attacking individual fairness on graph neural networks. Here, we consider the same surrogate graph learning model (i.e., 2-layer linear GCN) and the same lower-level loss function (i.e., cross entropy) as described in Section 4. To attack individual fairness, we define the upper-level bias function following the principles in (Kang et al., 2020): the fairness-related auxiliary information matrix $\mathbf{F}$ is defined as the oracle symmetric pairwise node similarity matrix $\mathbf{S}$ (i.e., $\mathbf{F} = \mathbf{S}$), where $\mathbf{S}\left[i,j\right]$ measures the similarity between node $i$ and node $j$. And the overall individual bias is defined as $\mathrm{Tr}\left(\mathbf{Y}^T\mathbf{L_S}\mathbf{Y}\right)$, where $\mathbf{L_S}$ is the Laplacian matrix of $\mathbf{S}$. Assuming that $\mathbf{Y}$ is the output of an optimization-based graph learning model, $\mathbf{Y}$ can be viewed as a function with respect to the input graph $\mathcal{G}$, which makes $\mathrm{Tr}\left(\mathbf{Y}^T\mathbf{L_S}\mathbf{Y}\right)$ differentiable with respect to $\mathcal{G}$. Thus, the bias function $b\left(\cdot\right)$ can be naturally defined as the overall individual bias of the input graph $\mathcal{G}$, i.e., $b\left(\mathbf{Y}, \Theta_{\mathrm{sur}}^*, \mathbf{S}\right) = \mathrm{Tr}\left(\mathbf{Y}^T\mathbf{L_S}\mathbf{Y}\right)$.

## 6 EXPERIMENTS

### 6.1 ATTACKING STATISTICAL PARITY ON GRAPH NEURAL NETWORKS

**Settings.** We compare FATE with 3 baseline methods: Random, DICE-S, and FA-GNN. Specifically, Random is a heuristic approach that randomly injects edges to the input graph. DICE-S is a variant of DICE (Waniek et al., 2018). It randomly deletes edges between nodes from different demographic groups and injects edges between nodes from the same demographic groups. FA-GNN (Hussain et al., 2022) attacks the fairness of a graph neural network by adversarially injecting edges that connect nodes in consideration of both their class labels and sensitive attribute values. We evaluate all methods under the same setting as in Section 4. That is, the fairness definition to be attacked is statistical parity; the downstream task is binary semi-supervised node classification with binary sensitive attributes. The experiments are conducted on 3 real-world datasets, i.e., Pokec-n, Pokec-z, and Bail. Similar to existing works, we use the 50%/25%/25% splits for training/validation/test sets. For all methods, the victim models are set to GCN (Kipf & Welling, 2017). For each dataset, we use a fixed random seed to learn the poisoned graph corresponding to each baseline method. Then we

Table 1: Attacking statistical parity on GCN under different perturbation rates (Ptb.). FATE poisons the graph via both edge flipping (FATE-flip) and edge addition (FATE-add) while all other baselines poison the graph via edge addition. Higher is better (↑) for micro F1 score (Micro F1) and $\Delta_{SP}$ (bias). Bold font indicates the most deceptive fairness attack, i.e., increasing $\Delta_{SP}$ and highest micro F1. Underlined cell indicates the failure of fairness attack, i.e., decreasing $\Delta_{SP}$ after attack.

| Dataset | Ptb. | Random | | DICE-S | | FA-GNN | | FATE-flip | | FATE-add | |
|---|---|---|---|---|---|---|---|---|---|---|---|
| | | Micro F1 (↑) | $\Delta_{SP}$ (↑) | Micro F1 (↑) | $\Delta_{SP}$ (↑) | Micro F1 (↑) | $\Delta_{SP}$ (↑) | Micro F1 (↑) | $\Delta_{SP}$ (↑) | Micro F1 (↑) | $\Delta_{SP}$ (↑) |
| **Pokec-n** | 0.00 | 67.5 ± 0.3 | 7.1 ± 0.4 | 67.5 ± 0.3 | 7.1 ± 0.4 | 67.5 ± 0.3 | 7.1 ± 0.4 | 67.5 ± 0.3 | 7.1 ± 0.4 | 67.5 ± 0.3 | 7.1 ± 0.4 |
| | 0.05 | 68.0 ± 0.3 | 6.2 ± 0.8 | 67.6 ± 0.3 | 7.1 ± 0.8 | 67.8 ± 0.1 | 3.3 ± 0.4 | 67.9 ± 0.4 | 9.3 ± 1.2 | 67.9 ± 0.4 | 9.3 ± 1.2 |
| | 0.10 | 66.8 ± 0.8 | 7.3 ± 0.7 | 67.9 ± 0.3 | 7.2 ± 0.5 | 66.0 ± 0.2 | 11.5 ± 0.6 | 68.2 ± 0.6 | 9.8 ± 1.5 | 68.2 ± 0.6 | 9.8 ± 1.5 |
| | 0.15 | 66.7 ± 0.4 | 8.1 ± 0.4 | 67.4 ± 0.3 | 7.9 ± 0.5 | 66.0 ± 0.4 | 15.6 ± 3.0 | 68.0 ± 0.3 | 11.5 ± 1.0 | 68.0 ± 0.3 | 11.5 ± 1.0 |
| | 0.20 | 66.3 ± 0.7 | 8.6 ± 1.8 | 66.1 ± 0.6 | 7.1 ± 1.2 | 65.8 ± 0.4 | 18.4 ± 0.7 | 68.2 ± 0.5 | 12.0 ± 1.8 | 68.2 ± 0.5 | 12.0 ± 1.8 |
| | 0.25 | 66.2 ± 0.6 | 8.5 ± 0.8 | 65.9 ± 0.4 | 6.5 ± 1.4 | 66.6 ± 0.2 | 23.3 ± 0.5 | 68.3 ± 0.4 | 12.1 ± 2.1 | 68.3 ± 0.4 | 12.1 ± 2.1 |
| **Pokec-z** | 0.00 | 68.4 ± 0.4 | 6.6 ± 0.9 | 68.4 ± 0.4 | 6.6 ± 0.9 | 68.4 ± 0.4 | 6.6 ± 0.9 | 68.4 ± 0.4 | 6.6 ± 0.9 | 68.4 ± 0.4 | 6.6 ± 0.9 |
| | 0.05 | 68.8 ± 0.4 | 6.4 ± 0.6 | 68.8 ± 0.3 | 5.7 ± 1.1 | 68.1 ± 0.3 | 2.2 ± 0.4 | 68.7 ± 0.4 | 6.7 ± 1.4 | 68.7 ± 0.4 | 6.7 ± 1.4 |
| | 0.10 | **68.7 ± 0.3** | **8.0 ± 0.6** | 67.7 ± 0.3 | 6.5 ± 0.8 | 67.7 ± 0.4 | 13.5 ± 0.9 | 68.7 ± 0.6 | 7.5 ± 0.7 | 68.7 ± 0.6 | 7.5 ± 0.7 |
| | 0.15 | 67.9 ± 0.3 | 9.1 ± 0.8 | 67.8 ± 0.6 | 4.8 ± 0.6 | 66.6 ± 0.4 | 16.9 ± 2.6 | 69.0 ± 0.8 | 8.5 ± 1.1 | 69.0 ± 0.8 | 8.5 ± 1.1 |
| | 0.20 | **68.5 ± 0.4** | **9.3 ± 1.0** | 67.0 ± 0.5 | 5.9 ± 0.7 | 66.1 ± 0.2 | 25.4 ± 1.3 | 68.5 ± 0.6 | 8.8 ± 1.1 | 68.5 ± 0.6 | 8.8 ± 1.1 |
| | 0.25 | 68.3 ± 0.5 | 7.3 ± 0.5 | 67.4 ± 0.6 | 5.8 ± 0.7 | 65.5 ± 0.6 | 22.3 ± 2.8 | 68.5 ± 1.1 | 8.6 ± 2.5 | 68.5 ± 1.1 | 8.6 ± 2.5 |
| **Bail** | 0.00 | 93.1 ± 0.2 | 8.0 ± 0.2 | 93.1 ± 0.2 | 8.0 ± 0.2 | 93.1 ± 0.2 | 8.0 ± 0.2 | 93.1 ± 0.2 | 8.0 ± 0.2 | 93.1 ± 0.2 | 8.0 ± 0.2 |
| | 0.05 | **92.7 ± 0.2** | **8.1 ± 0.0** | 92.3 ± 0.2 | 8.4 ± 0.2 | 91.7 ± 0.1 | 10.0 ± 0.4 | 92.6 ± 0.1 | 8.6 ± 0.1 | 92.5 ± 0.1 | 8.6 ± 0.1 |
| | 0.10 | 92.2 ± 0.2 | 7.8 ± 0.2 | 92.2 ± 0.2 | 8.5 ± 0.3 | 90.5 ± 0.0 | 10.3 ± 0.4 | **92.4 ± 0.1** | **8.9 ± 0.1** | 92.4 ± 0.1 | 8.6 ± 0.1 |
| | 0.15 | 91.9 ± 0.2 | 7.8 ± 0.1 | 92.1 ± 0.1 | 8.9 ± 0.1 | 90.0 ± 0.2 | 8.4 ± 0.2 | 92.2 ± 0.2 | 9.1 ± 0.1 | 92.3 ± 0.1 | 9.1 ± 0.1 |
| | 0.20 | 91.6 ± 0.2 | 7.8 ± 0.1 | 91.8 ± 0.1 | 9.1 ± 0.2 | 89.7 ± 0.1 | 7.4 ± 0.4 | 92.2 ± 0.2 | 9.3 ± 0.1 | 92.3 ± 0.1 | 9.3 ± 0.2 |
| | 0.25 | 91.4 ± 0.1 | 8.3 ± 0.1 | 91.6 ± 0.2 | 9.3 ± 0.1 | 89.8 ± 0.2 | 5.2 ± 0.2 | 92.1 ± 0.1 | 9.1 ± 0.2 | 92.1 ± 0.1 | 9.1 ± 0.3 |

train the victim model 5 times with different random seeds. For a fair comparison, we only attack the adjacency matrix. Please refer to Appendix C for detailed experimental settings.

**Main results.** For FATE, we conduct fairness attacks via both edge flipping (FATE-flip) and edge addition (FATE-add). For all other baseline methods, edges are only added. The effectiveness of fairness attacks on GCN are presented in Table 1. From the table, we have the following key observations: (A) FATE-flip and FATE-add are the only methods that consistently succeed in fairness attacks, while all other baseline methods might fail in some cases (indicated by the underlined $\Delta_{SP}$) because of the decrease in $\Delta_{SP}$. Though DICE-S consistently succeeds in fairness attacks on Pokec-n and Bail, its utility is worse than FATE-flip and FATE-add, making it less deceptive. (B) FATE-flip and FATE-add not only amplify $\Delta_{SP}$ consistently, but also achieve the best micro F1 score on node classification, which makes FATE-flip and FATE-add more deceptive than all baseline methods. Notably, FATE-flip and FATE-add are able to even increase micro F1 score on all datasets, while other baseline methods attack the graph neural networks at the expense of utility (micro F1 score). (C) Though FA-GNN could make the model more biased in some cases, it cannot guarantee consistent success in fairness attacks on all three datasets as shown by the underlined $\Delta_{SP}$ in both tables. All in all, our proposed FATE framework consistently succeeds in fairness attacks while being the most deceptive (i.e., highest micro F1 score).

**Effect of the perturbation rate.** From Table 1, first, $\Delta_{SP}$ tends to increase when the perturbation rate increases, which demonstrates the effectiveness of FATE-flip and FATE-add for attacking fairness. Though in some cases $\Delta_{SP}$ might have a marginal decrease, FATE-flip and FATE-add still successfully attack the fairness compared with GCN trained on the benign graph by being larger to the $\Delta_{SP}$ when perturbation rate (Ptb.) is 0. Second, FATE-flip and FATE-add are deceptive, meaning that the micro F1 scores is close to or even higher than the micro F1 scores on the benign graph compared with the corresponding metrics trained on the poisoned graphs. In summary, across different perturbation rates, FATE-flip and FATE-add are both effective, i.e., amplifying more bias with higher perturbation rate, and deceptive, i.e., achieving similar or even higher micro F1 score.

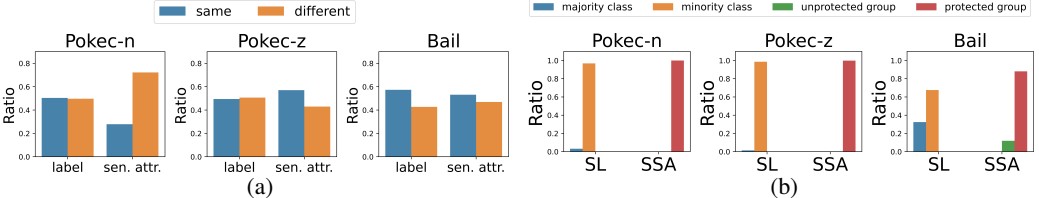

Figure 1: Attacking statistical parity with FATE-flip. (a) Ratios of flipped edges that connect two nodes with same/different label or sensitive attribute (sens. attr.). (b) SL (abbreviation for same label) refers to the ratios of flipped edges whose two endpoints are both from the same class. SSA (abbreviation for same sensitive attribute) refers to the ratios of manipulated edges whose two endpoints are both from the same demographic group. Majority/minority classes are determined by splitting the training nodes based on their class labels. The protected group is the demographic group with fewer nodes.

**Analysis on the manipulated edges.** Here, we aim to characterize the properties of edges that are flipped by FATE (i.e., FATE-flip) in attacking statistical parity with perturbation rate 25%. The reason

Table 2: Attacking individual fairness on GCN under different perturbation rates (Ptb.). FATE poisons the graph via both edge flipping (FATE-flip) and edge addition (FATE-add) while all other baselines poison the graph via edge addition. Higher is better (↑) for micro F1 score (Micro F1) and InFoRM bias (Bias). Bold font indicates the most deceptive fairness attack, i.e., increasing bias and highest micro F1. Underlined cell indicates the failure of fairness attack, i.e., decreasing bias after attack.

| Dataset | Ptb. | Random | | DICE-S | | FA-GNN | | FATE-flip | | FATE-add | |
|---|---|---|---|---|---|---|---|---|---|---|---|
| | | Micro F1 (↑) | Bias (↑) | Micro F1 (↑) | Bias (↑) | Micro F1 (↑) | Bias (↑) | Micro F1 (↑) | Bias (↑) | Micro F1 (↑) | Bias (↑) |
| Pokec-n | 0.00 | 67.5 ± 0.3 | 0.9 ± 0.2 | 67.5 ± 0.3 | 0.9 ± 0.2 | 67.5 ± 0.3 | 0.9 ± 0.2 | 67.5 ± 0.3 | 0.9 ± 0.2 | 67.5 ± 0.3 | 0.9 ± 0.2 |
| | 0.05 | 67.6 ± 0.3 | 1.6 ± 0.3 | **68.1 ± 0.2** | **2.0 ± 0.6** | **67.8 ± 0.5** | **1.9 ± 0.2** | 67.8 ± 0.3 | 1.2 ± 0.4 | 67.6 ± 0.3 | 1.5 ± 0.6 |
| | 0.10 | 67.2 ± 0.5 | 1.4 ± 0.3 | 66.9 ± 1.0 | 1.3 ± 0.3 | 67.4 ± 0.4 | 1.2 ± 0.2 | **67.9 ± 0.4** | **1.3 ± 0.3** | 67.7 ± 0.4 | 1.6 ± 0.4 |
| | 0.15 | 67.2 ± 0.3 | 1.2 ± 0.4 | 67.4 ± 0.3 | 1.3 ± 0.2 | 66.1 ± 0.3 | 1.5 ± 0.3 | **67.8 ± 0.4** | **1.2 ± 0.2** | 67.6 ± 0.2 | 1.1 ± 0.3 |
| | 0.20 | 66.6 ± 0.3 | 1.1 ± 0.2 | 67.3 ± 0.3 | 1.5 ± 0.5 | 65.7 ± 0.6 | 1.5 ± 0.3 | 67.3 ± 0.4 | 1.1 ± 0.3 | **68.2 ± 1.0** | **1.7 ± 0.8** |
| | 0.25 | 66.7 ± 0.3 | 1.3 ± 0.4 | 66.6 ± 0.5 | 1.3 ± 0.1 | 65.2 ± 0.5 | 1.3 ± 0.4 | **67.8 ± 0.8** | **1.4 ± 0.7** | 67.9 ± 0.9 | 1.4 ± 0.7 |
| Pokec-z | 0.00 | 68.4 ± 0.4 | 2.6 ± 0.7 | 68.4 ± 0.4 | 2.6 ± 0.7 | 68.4 ± 0.4 | 2.6 ± 0.7 | 68.4 ± 0.4 | 2.6 ± 0.7 | 68.4 ± 0.4 | 2.6 ± 0.7 |
| | 0.05 | 69.0 ± 0.4 | 3.4 ± 0.5 | **68.9 ± 0.5** | **3.3 ± 0.9** | 68.1 ± 0.4 | 2.9 ± 0.3 | 68.7 ± 0.5 | 2.9 ± 0.5 | 68.7 ± 0.4 | 3.1 ± 1.0 |
| | 0.10 | 68.7 ± 0.1 | 2.4 ± 0.5 | **69.1 ± 0.2** | **3.3 ± 0.8** | 68.2 ± 0.5 | 1.7 ± 0.5 | 69.0 ± 0.6 | 2.9 ± 0.6 | 69.0 ± 0.5 | 3.0 ± 0.6 |
| | 0.15 | 67.9 ± 0.3 | 2.8 ± 0.3 | 68.1 ± 0.2 | 3.6 ± 0.4 | 67.0 ± 0.5 | 1.3 ± 0.2 | 68.6 ± 0.5 | 2.9 ± 0.6 | **69.0 ± 0.7** | **2.7 ± 0.4** |
| | 0.20 | 67.9 ± 0.3 | 2.2 ± 0.6 | 67.8 ± 0.3 | 2.7 ± 0.6 | 66.1 ± 0.1 | 1.6 ± 0.5 | 68.8 ± 0.4 | 3.0 ± 0.4 | **69.2 ± 0.4** | **2.9 ± 0.3** |
| | 0.25 | 67.6 ± 0.3 | 1.9 ± 0.3 | 68.4 ± 0.4 | 2.6 ± 0.7 | 65.1 ± 0.3 | 1.9 ± 0.6 | 69.1 ± 0.3 | 2.9 ± 0.7 | **69.3 ± 0.3** | **2.7 ± 0.6** |
| Bail | 0.00 | 93.1 ± 0.2 | 7.2 ± 0.6 | 93.1 ± 0.2 | 7.2 ± 0.6 | 93.1 ± 0.2 | 7.2 ± 0.6 | 93.1 ± 0.2 | 7.2 ± 0.6 | 93.1 ± 0.2 | 7.2 ± 0.6 |
| | 0.05 | 92.1 ± 0.3 | 8.0 ± 1.9 | 92.3 ± 0.2 | 9.1 ± 2.7 | 91.2 ± 0.2 | 5.6 ± 0.7 | **93.0 ± 0.3** | 7.8 ± 1.0 | 92.9 ± 0.2 | 7.7 ± 1.0 |
| | 0.10 | 91.6 ± 0.1 | 7.3 ± 1.2 | 92.2 ± 0.2 | 8.0 ± 1.8 | 90.3 ± 0.1 | 5.1 ± 0.4 | **93.0 ± 0.1** | **8.0 ± 0.7** | 92.9 ± 0.2 | 7.9 ± 0.8 |
| | 0.15 | 91.3 ± 0.1 | 6.5 ± 0.9 | 92.1 ± 0.2 | 7.7 ± 0.4 | 89.8 ± 0.1 | 5.2 ± 0.1 | **93.1 ± 0.1** | **8.2 ± 0.6** | 93.0 ± 0.2 | 7.8 ± 0.8 |
| | 0.20 | 91.2 ± 0.2 | 6.6 ± 0.6 | 91.8 ± 0.1 | 7.1 ± 1.2 | 89.3 ± 0.1 | 5.3 ± 0.4 | 93.1 ± 0.1 | 7.9 ± 0.6 | **93.1 ± 0.1** | **8.2 ± 0.6** |
| | 0.25 | 90.9 ± 0.1 | 6.8 ± 0.8 | 91.5 ± 0.1 | 6.3 ± 0.9 | 88.9 ± 0.1 | 5.4 ± 0.3 | 92.9 ± 0.1 | 7.6 ± 0.5 | **93.0 ± 0.2** | **7.8 ± 0.7** |

to only analyze FATE-flip is that the majority of edges manipulated by FATE-flip on all datasets is by addition (i.e., flipping from non-existing to existing). Figure 1b suggests that, if two endpoints of a manipulated edge share the same class label or sensitive attribute value, these two endpoints are most likely from the minority class and protected group. Thus, FATE would significantly increase the number of edges that are incident to nodes in the minority class and/or protected group.

**More experimental results.** Due to the space limitation, we defer more experimental results on attacking statistical parity on graph neural networks in Appendix D. More specifically, we present the performance evaluation under Macro F1 and AUC, as well as the effectiveness of FATE with FairGNN (Dai & Wang, 2021) for statistical parity as the victim model.

## 6.2 ATTACKING INDIVIDUAL FAIRNESS ON GRAPH NEURAL NETWORKS

**Settings.** To showcase the ability of FATE on attacking the individual fairness (Section 5), we further compare FATE with the same set of baseline methods (Random, DICE-S, FA-GNN) on the same set of datasets (Pokec-n, Pokec-z, Bail). We follow the settings as in Section 5. We use the 50%/25%/25% splits for train/validation/test sets with GCN being the victim model. For each dataset, we use a fixed random seed to learn the poisoned graph corresponding to each baseline method. Then we train the victim model 5 times with different random seeds. And each entry in the oracle pairwise node similarity matrix is computed by the cosine similarity of the corresponding rows in the adjacency matrix. That is, $\mathbf{S}[i,j] = \cos(\mathbf{A}[i,:], A[j,:])$, where $\cos()$ is the function to compute cosine similarity. For a fair comparison, we only attack the adjacency matrix in all experiments. Please refer to Appendix C for detailed experimental settings.

**Main results.** We test FATE with both edge flipping (FATE-flip) and edge addition (FATE-add), while all other baseline methods only add edges. From Table 2, we have two key observations. (A) FATE-flip and FATE-add are effective: they are the only methods that could consistently attack individual fairness whereas all other baseline methods mostly fail to attack individual fairness. (B) FATE-flip and FATE-add are deceptive: they achieve comparable or even better utility on all datasets compared with the utility on the benign graph. Hence, FATE framework is able to achieve effective and deceptive attacks to exacerbate individual bias.

**Effect of the perturbation rate.** From Table 2, we obtain similar observations as in Section 6.1 for Bail dataset. While for Pokec-n and Pokec-z, the correlation between the perturbation rate (Ptb.) and the individual bias is weaker. One possible reason is that: for Pokec-n and Pokec-z, the discrepancy between the oracle pairwise node similarity matrix and the benign graph is larger. Since the individual bias is computed using the oracle pairwise node similarity matrix rather than the benign/poisoned adjacency matrix, a higher perturbation rate to poison the adjacency matrix may have less impact on the computation of individual bias.

**Analysis on the manipulated edges.** Since the majority of edges manipulated by FATE-flip is through addition, we only analyze FATE-flip here with perturbation rate 25%. From Figure 2, we can find out that FATE tends to manipulate edges from the same class (especially from the minority class). In this way, FATE would find edges that could increase individual bias and improve the utility of the minority class in order to make the fairness attack deceptive.

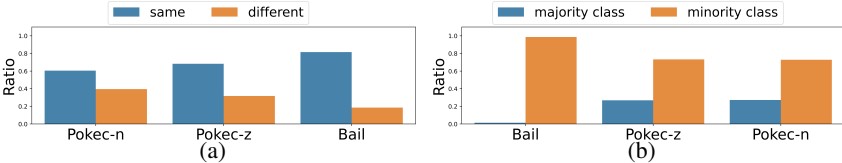

Figure 2: Attacking individual fairness with FATE-flip. (a) Ratios of flipped edges that connect two nodes with same/different label. (b) Ratios of flipped edges whose two endpoints are both from the majority/minority class. Majority/minority classes are formed by splitting the training nodes based on their class labels.

**More experimental results.** Due to the space limitation, we defer more experimental results on attacking individual fairness on graph neural networks in Appendix E. More specifically, we present the performance evaluation under Macro F1 and AUC, as well as the effectiveness of FATE with InFoRM-GNN (Kang et al., 2020), which mitigates individual bias, as the victim model.

## 7 RELATED WORK

**Algorithmic fairness on graphs** aims to obtain debiased graph learning results such that a pre-defined fairness measure can be satisfied with respect to the nodes/edges in the graph. Several definitions of the fairness have been studied so far. Group fairness in graph embedding can be ensured via several strategies, including adversarial learning (Bose & Hamilton, 2019; Dai & Wang, 2021), biased random walk (Rahman et al., 2019; Khajehnejad et al., 2022), bias-free graph generation (Wang et al., 2022), and dropout (Spinelli et al., 2021). Individual fairness on graphs can be ensured via Lipschitz regularization (Kang et al., 2020) and learning-to-rank (Dong et al., 2021). Other than the aforementioned two fairness definitions, several other fairness definitions are studied in the context of graph learning, including counterfactual fairness (Agarwal et al., 2021; Ma et al., 2021), degree fairness (Tang et al., 2020; Kang et al., 2022; Liu et al., 2023b), dyadic fairness (Masrour et al., 2020; Li et al., 2021), and max-min fairness (Rahmattalabi et al., 2019; Tsang et al., 2019). For a comprehensive review of related works, please refer to recent surveys (Zhang et al., 2022; Choudhary et al., 2022; Dong et al., 2022) and tutorials (Kang & Tong, 2021; 2022). It should be noted that our work aims to attack fairness rather than ensuring fairness as in the aforementioned literature.

**Adversarial attacks on graphs** aim to exacerbate the utility of graph learning models by perturbing the input graph topology and/or node features. Several approaches have been proposed to attack graph learning models, including reinforcement learning (Dai et al., 2018), bi-level optimization (Zügner et al., 2018; Zügner & Günnemann, 2019), projected gradient descent (Sun et al., 2018; Xu et al., 2019), spectral distance perturbation (Lin et al., 2022), and edge rewiring/flipping (Bojchevski & Günnemann, 2019; Ma et al., 2021). Other than adversarial attacks that worsen the utility of a graph learning model, a few efforts have been made to attack the fairness of a machine learning model for IID tabular data via label flipping (Mehrabi et al., 2021), adversarial data injection (Solans et al., 2021; Chhabra et al., 2021), adversarial sampling (Van et al., 2022). Different from (Solans et al., 2021; Mehrabi et al., 2021; Chhabra et al., 2021; Van et al., 2022), we aim to poison the input graph via structural modifications on the topology rather than injecting adversarial data sample(s). The most related works to our proposed method are (Hussain et al., 2022) and (Zhang et al., 2023). Hussain et al. (2022) degrades the group fairness of graph neural networks by randomly injecting edges for nodes in different demographic groups and with different class labels. In contrast, our proposed method could attack *any* fairness definition for *any* graph learning models via arbitrary edge manipulation operations in consideration of the utility of the downstream task, as long as the bias function and the utility loss are differentiable. Zhang et al. (2023) is a concurrent study which utilizes zeroth-order optimization instead of gradient-based solution as FATE to solve a similar bi-level problem.

## 8 CONCLUSION

We study deceptive fairness attacks on graphs, whose goal is to amplify the bias while maintaining or improving the utility on the downstream task. We formally define the problem as a bi-level optimization problem, where the upper-level optimization problem maximizes the bias function with respect to a user-defined fairness definition and the lower-level optimization problem minimizes a task-specific loss function. We then propose a meta learning-based framework named FATE to poison the input graph using the meta-gradient of the bias function with respect to the input graph. We instantiate FATE by attacking statistical parity on graph neural networks in a binary node classification problem with binary sensitive attributes. Empirical evaluation demonstrates that FATE is effective (amplifying bias) and deceptive (achieving the highest micro F1 score).

ACKNOWLEDGEMENTS

This work is partially supported by NSF (2134079, 1939725, 2316233, 2238208), DHS (17STQAC00001-07-00, 17STQAC00001-06-00), and NIFA (2020-67021-32799). The views and conclusions contained in this document are those of the authors and should not be interpreted as necessarily representing the official policies, either expressed or implied, of the U.S. Department of Homeland Security.

ETHICAL STATEMENT

The goal of this paper is to investigate the possibility of making the graph learning results more biased, in order to raise the awareness of fairness attacks. Meanwhile, our experiments suggest that existing fair graph neural networks suffer from the fairness attacks, which further highlight the importance of designing robust and fair techniques to protect the civil rights of marginalized individuals. We acknowledge that the proposed method FATE, if misused, could impact the integrity and fairness of graph learning models. When used for commercial purpose, FATE might cause civil rights violation(s) and could be harmful to individuals from certain demographic groups. To prevent the negative societal impacts, the code will be publicly released under CC-BY-NC-ND license upon publication, which prohibits the use of FATE for any commercial purposes, and explicitly highlight in the released code that any use of the developed techniques should be consulted with the authors for permission first.

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

ORGANIZATION OF THE APPENDIX

The supplementary material contains the following information.

- Appendix A provides additional examples of graph learning models from the optimization perspective.
- Appendix B presents the pseudocode of FATE.
- Appendix C offers the detailed parameter settings regarding the reproducibility of this paper.
- Appendix D provides additional experimental results on using FairGNN (Dai & Wang, 2021) and evaluating under macro F1 score and AUC score.
- Appendix E provides additional experimental results on using InFoRM-GNN (Kang et al., 2020) and evaluating under macro F1 score and AUC score.
- Appendix F shows the transferability of using FATE to attack the statistical parity or individual fairness of the non-convolutional aggregation-based graph attention network with linear GCN as the surrogate model.
- Appendix G provides further discussions on (1) the relationship between fairness attacks and the impossibility theorem as well as Metattack (Zügner & Günnemann, 2019), (2) an alternative perturbation set selection strategy via sampling, (3) the potential of FATE on attacking the fairness of a specific demographic group, and (4) justification of applying kernel density estimation on non-IID graph data.
- Appendix H presents more details of statistical parity and individual fairness.

Code can be found at the following repository:

https://github.com/jiank2/FATE.

## A  GRAPH LEARNING MODELS FROM THE OPTIMIZATION PERSPECTIVE

Here, we discuss four additional non-parameterized graph learning models from the optimization perspective, including PageRank, spectral clustering, matrix factorization-based completion and first-order LINE.

**Model #1: PageRank.** It is one of the most successful random walk based ranking algorithm to measure node importance. Mathematically, PageRank solves the linear system

$$\mathbf{r} = c\mathbf{P}\mathbf{r} + (1 - c)\mathbf{e} \tag{9}$$

where $c$ is the damping factor, $\mathbf{P}$ is the propagation matrix and $\mathbf{e}$ is the teleportation vector. In PageRank, the propagation matrix $\mathbf{P}$ is often defined as the row-normalized adjacency matrix of a graph $\mathcal{G}$ and the teleportation vector is a uniform distribution $\frac{1}{n}\mathbf{1}$ with $\mathbf{1}$ being a vector filled with 1. Equivalently, given a damping factor $c$ and a teleportation vector $\mathbf{e}$, the PageRank vector $\mathbf{Y} = \mathbf{r}$ can be learned by minimizing the following loss function

$$\min_{\mathbf{r}} \quad c\mathbf{r}^T(\mathbf{I} - \mathbf{P})\mathbf{r} + (1 - c)\|\mathbf{r} - \mathbf{e}\|_2^2 \tag{10}$$

where $c\left(\mathbf{r}^T\left(\mathbf{I} - \mathbf{P}\right)\mathbf{r}\right)$ is a smoothness term and $(1 - c)\left\|\mathbf{r} - \mathbf{e}\right\|_2^2$ is a query-specific term. To attack the fairness of PageRank with FATE, the attacker could attack a surrogate PageRank with different choices of damping factor $c$ and/or teleportation vector $\mathbf{e}$.

**Model #2: Spectral clustering.** It aims to identify clusters of nodes such that the intra-cluster connectivity are maximized while inter-cluster connectivity are minimized. To find $k$ clusters of nodes, spectral clustering finds a soft cluster membership matrix $\mathbf{Y} = \mathbf{C}$ with orthonormal columns by minimizing the following loss function

$$\min_{\mathbf{C}} \quad \text{Tr}\left(\mathbf{C}^T\mathbf{L}\mathbf{C}\right) \tag{11}$$

where $\mathbf{L}$ is the (normalized) graph Laplacian of the input graph $\mathcal{G}$. It is worth noting that the columns of learning result $\mathbf{C}$ is equivalent to the eigenvectors of $\mathbf{L}$ associated with smallest $k$ eigenvalues. To

attack the fairness of spectral clustering with FATE, the attacker might attack a surrogate spectral clustering with different number of clusters $k$.

**Model #3: Matrix factorization-based completion.** Suppose we have a bipartite graph $\mathcal{G}$ with $n_1$ users, $n_2$ items and $m$ interactions between users and items. Matrix factorization-based completion aims to learn two low-rank matrices an $n_1 \times z$ matrix $\mathbf{U}$ and an $n_2 \times z$ matrix $\mathbf{V}$ such that the following loss function will be minimized

$$\min_{\mathbf{U},\mathbf{V}} \quad \| \operatorname{proj}_\Omega \left( \mathbf{R} - \mathbf{U}\mathbf{V}^T \right) \|_F^2 + \lambda_1 \|\mathbf{U}\|_F^2 \lambda_2 + \|\mathbf{V}\|_F^2 \tag{12}$$

where $\mathbf{A} = \begin{pmatrix} \mathbf{0}_{n_1} & \mathbf{R} \\ \mathbf{R}^T & \mathbf{0}_{n_2} \end{pmatrix}$ with $\mathbf{0}_{n_1}$ being an $n_1 \times n_1$ square matrix filled with 0, $\Omega = \{(i,j)|(i,j) \text{ is observed}\}$ is the set of observed interaction between any user $i$ and any item $j$, $\operatorname{proj}_\Omega (\mathbf{Z}) [i,j]$ equals to $\mathbf{Z}[i,j]$ if $(i,j) \in \Omega$ and 0 otherwise, $\lambda_1$ and $\lambda_2$ are two hyperparameters for regularization. To attack the fairness of matrix factorization-based completion with FATE, the attacker could attack a surrogate model with different number of latent factors $z$.

**Model #4: First-order LINE.** It is a skip-gram based node embedding model. The key idea of first-order LINE is to map each node into a $h$-dimensional space such that the dot product of the embeddings of any two connected nodes will be small. To achieve this goal, first-order LINE essentially optimizes the following loss function

$$\max_{\mathbf{H}} \sum_{i=1}^n \sum_{j=1}^n \mathbf{A}[i,j] \left( \log g \left( \mathbf{H}[j,:]\mathbf{H}[i,:]^T \right) + k\mathbb{E}_{j' \sim P_n}[\log g \left( -\mathbf{H}[j',:]\mathbf{H}[i,:]^T \right)] \right) \tag{13}$$

where $\mathbf{H}$ is the embedding matrix with $\mathbf{H}[i,:]$ being the $h$-dimensional embedding of node $i$, $g(x) = 1/(1 + e^{-x})$ is the sigmoid function, $k$ is the number of negative samples and $P_n$ is the distribution for negative sampling such that the sampling probability for node $i$ is proportional to its degree $\deg_i$. For a victim first-order LINE, the attacker could attack a surrogate LINE (1st) with different dimension $h$ in the embedding space and/or a different number of negative samples $g$.

**Remarks.** Note that, for a non-parameterized graph learning model (e.g., PageRank, spectral clustering, matrix completion, first-order LINE), we have $\Theta = \{\mathbf{Y}\}$ which is the set of learning results. For example, we have $\Theta = \{\mathbf{r}\}$ for PageRank, $\Theta = \{\mathbf{C}\}$ for spectral clustering, $\Theta = \{\mathbf{U}, \mathbf{V}\}$ and $\Theta = \{\mathbf{H}\}$ for LINE (1st). For parameterized graph learning models (e.g., GCN), $\Theta$ refers to the set of learnable weights, e.g., $\Theta = \{\mathbf{W}^{(1)}, \ldots, \mathbf{W}^{(L)}\}$ for an $L$-layer GCN.

# B    PSEUDOCODE OF FATE

Algorithm 1 summarizes the detailed steps on fairness attack with FATE. To be specific, after initialization (line 1), we pre-train the surrogate graph learning model (lines 4 – 6) and get the pre-trained surrogate model $\Theta^{(T)}$ as well as learning results $\mathbf{Y}^{(T)}$ (line 7). After that, we compute the meta gradient of the bias function (lines 8 – 11) and perform either discretized attack or continuous attack based on the interest of attacker (i.e., discretized poisoning attack in lines 12 – 15 or continuous poisoning attack in lines 16 – 18).

# C    EXPERIMENTAL SETTINGS

In this section, we provide more detailed information about the experimental settings. These include the hardware and software specifications, dataset descriptions, evaluation metrics as well as detailed parameter settings. In all experiments, we evaluate our proposed FATE in the task of semi-supervised node classification to answer the following questions:

**Q1.** How effective is FATE in exacerbating bias under different perturbation rates?

**Q2.** How effective is FATE in maintaining node classification accuracy for deceptiveness under different perturbation rates?

**Q3.** Can we characterize the properties of edges perturbed by FATE?

---

**Algorithm 1:** FATE

---

**Given :** an undirected graph $\mathcal{G} = \{\mathbf{A}, \mathbf{X}\}$, the set of training nodes $\mathcal{V}_{\text{train}}$, fairness-related auxiliary information matrix $\mathbf{F}$, total budget $B$, budget in step $i$ $\delta_i$, the bias function $b$, number of pre-training epochs $T$;

**Find :** the poisoned graph $\widetilde{\mathcal{G}}$;

1 poisoned graph $\widetilde{\mathcal{G}} \leftarrow \mathcal{G}$, cumulative budget $\Delta \leftarrow 0$, step counter $i \leftarrow 0$;

2 **while** $\Delta < B$ **do**

3      $\nabla_{\widetilde{\mathcal{G}}} b \leftarrow 0$;

4      **for** $t = 1$ *to* $T$ **do**

5          update $\Theta_{\text{sur}}^{(t)}$ to $\Theta_{\text{sur}}^{(t+1)}$ with a gradient-based optimizer (e.g., Adam);

6      **end**

7      get $\mathbf{Y}^{(T)}$ and $\Theta_{\text{sur}}^{(T)}$;

8      compute meta-gradient $\nabla_{\mathcal{G}} b \leftarrow \nabla_{\Theta_{\text{sur}}^{(T)}} b \left( \mathbf{Y}, \Theta_{\text{sur}}^{(T)}, \mathbf{F} \right) \cdot \nabla_{\mathcal{G}} \Theta_{\text{sur}}^{(T)}$;

9      **if** *attack the adjacency matrix* **then**

10          compute the derivative $\nabla_{\widetilde{\mathbf{A}}} b \leftarrow \nabla_{\widetilde{\mathbf{A}}} b + \left( \nabla_{\widetilde{\mathbf{A}}} b \right)^T - \text{diag} \left( \nabla_{\widetilde{\mathbf{A}}} b \right)$;

11      **end**

12      **if** *discretized poisoning attack* **then**

13          compute the poisoning preference matrix $\nabla_{\widetilde{\mathbf{A}}}$ by Eq. equation 5;

14          select the edges to poison in $\nabla_{\widetilde{\mathbf{A}}}$ with budget $\delta_i$ by Eq. equation 6;

15          update the corresponding entries in $\widetilde{\mathcal{G}}$;

16      **else**

17          update $\widetilde{\mathcal{G}}$ by Eq. equation 4 with budget $\delta_i$;

18      **end**

19      $\Delta \leftarrow \Delta + \delta_i$;

20      $i \leftarrow i + 1$;

21 **end**

22 **return** $\widetilde{\mathcal{G}}$;

---

## C.1 HARDWARE AND SOFTWARE SPECIFICATIONS

All codes are programmed in Python 3.8.13 and PyTorch 1.12.1. All experiments are performed on a Linux server with 2 Intel Xeon Gold 6240R CPUs and 4 Nvidia Tesla V100 SXM2 GPUs, each of which has 32 GB memory.

## C.2 DATASET DESCRIPTIONS

We use three widely-used benchmark datasets for fair graph learning: Pokec-z, Pokec-n and Bail. For each dataset, we use a fixed random seed to split the dataset into training, validation and test sets with the split ratio being 50%, 25%, and 25%, respectively. The statistics of the datasets, including the number of nodes (# Nodes), the number of edges (# Edges), the number of features (# Features), the sensitive attribute (Sensitive Attr.) and the label (Label), are summarized in Table 3.

- Pokec-z and Pokec-n are two datasets collected from the Slovakian social network *Pokec*, each of which represents a sub-network of a province. Each node in these datasets is a user belonging to two major regions of the corresponding provinces, and each edge is the friendship relationship between two users. The sensitive attribute is the user region, and the label is the working field of a user.

- Bail is a similarity graph of criminal defendants during $1990 - 2009$. Each node is a defendant during this time period. Two nodes are connected if they share similar past criminal records and demographics. The sensitive attribute is the race of the defendant, and the label is whether the defendant is on bail or not.

Table 3: Statistics of the datasets.

| Dataset | Pokec-z | Pokec-n | Bail |
|---|---|---|---|
| # Nodes | $7,659$ | $6,185$ | $18,876$ |
| # Edges | $20,550$ | $15,321$ | $311,870$ |
| # Features | $276$ | $265$ | $17$ |
| Sensitive Attr. | Region | Region | Race |
| Label | Working field | Working field | Bail decision |

## C.3 EVALUATION METRICS

In our experiments, we aim to evaluate how effective FATE is in (1) attacking the fairness and (2) maintaining the utility of node classification.

To evaluate the performance of FATE in attacking the group fairness, we evaluate the effectiveness using $\Delta_{\text{SP}}$, which is defined as follows.

$$\Delta_{\text{SP}} = |P\left[\widehat{y} = 1 \mid s = 1\right] - P\left[\widehat{y} = 1 \mid s = 0\right]| \tag{14}$$

where $s$ is the sensitive attribute value of a node and $\widehat{y}$ is the ground-truth and predicted class labels of a node. While to evaluate the performance of FATE in attacking the individual fairness, we evaluate the effectiveness using the InFoRM bias (Bias) measure (Kang et al., 2020), which is defined as follows.

$$\text{Bias} = \sum_{i \in \mathcal{V}_{\text{test}}} \sum_{j \in \mathcal{V}_{\text{test}}} \mathbf{S}\left[i, j\right] \|\mathbf{Y}\left[i, :\right] - \mathbf{Y}\left[j, :\right]\|_F^2 \tag{15}$$

where $\mathcal{V}_{\text{test}}$ is the set of test nodes and $\mathbf{S}$ is the oracle pairwise node similarity matrix. The intuition of Eq. equation 15 is to measure the squared difference between the learning results of two test nodes, weighted by their pairwise similarity.

To evaluate the performance of FATE in maintaining the utility, we use micro F1 score (Micro F1), macro F1 score (Macro F1) and AUC score.

## C.4 DETAILED PARAMETER SETTINGS

**Poisoning the input graph.** During poisoning attacks, we set a fixed random seed to control the randomness. The random seed used for each dataset in attacking group/individual fairness are summarized in Table 4.

- **Surrogate model training.** We run all methods with a perturbation rate from 0.05 to 0.25 with a step size of 0.05. For FA-GNN (Hussain et al., 2022), we follow its official implementation and use the same surrogate 2-layer GCN (Kipf & Welling, 2017) with 16 hidden dimensions for poisoning attack.[4]. The surrogate GCN in FA-GNN is trained for 500 epochs with a learning rate $1e - 2$, weight decay $5e - 4$, and dropout rate 0.5. For FATE, we use a 2-layer linear GCN (Wu et al., 2019) with 16 hidden dimensions for poisoning attacks. And the surrogate linear GCN in FATE is trained for 500 epochs with a learning rate $1e - 2$, weight decay $5e - 4$, and dropout rate 0.5.

- **Graph topology manipulation.** For Random and DICE, we use the implementations provided in the deeprobust package with the default parameters to add the adversarial edges.[5]. For FA-GNN, we add adversarial edges that connect two nodes with different class labels and different sensitive attributes, which provides the most promising performance as shown in (Hussain et al., 2022). For FATE, suppose we poison the input graph in $p$ $(p > 1)$ attacking steps. Then the per-iteration attacking budget in Algorithm 1 is set as $\delta_1 = 1$ and $\delta_i = \frac{r|\mathcal{E}|-1}{p-1}$, $\forall i \in \{2, \ldots, p\}$, where $r$ is the perturbation rate and $|\mathcal{E}|$ is the number of edges. Detailed choices of $p$ for each dataset in attacking group/individual fairness are summarized in Table 4.

**Training the victim model.** We use a fixed list of random seed ($[0, 1, 2, 42, 100]$) to train each victim model 5 times and report the mean and standard deviation. Regarding the victim models in

---

[4]https://github.com/mengcao327/attack-gnn-fairness
[5]https://deeprobust.readthedocs.io/

Table 4: Parameter settings on the random seed for all baseline methods in poisoning attacks (Random Seed) and the number of steps for poisoning attacks in FATE (Attacking Steps).

| Dataset | Fairness Definition | Attacking Steps | Random Seed |
|---------|--------------------|-----------------|-------------|
| Pokec-n | Statistical parity | 3 | 25 |
|         | Individual fairness | 3 | 45 |
| Pokec-z | Statistical parity | 3 | 25 |
|         | Individual fairness | 5 | 15 |
| Bail    | Statistical parity | 3 | 25 |
|         | Individual fairness | 3 | 5 |

group fairness attacks, we train a 2-layer GCN (Kipf & Welling, 2017) for 400 epochs and a 2-layer FairGNN (Dai & Wang, 2021) for 2000 epochs to evaluate the efficacy of fairness attacks. The hidden dimension, learning rate, weight decay and dropout rate of GCN and FairGNN are set to 128, $1e-3$, $1e-5$ and 0.5, respectively. The regularization parameters in FairGNN, namely $\alpha$ and $\beta$, are set to 100 and 1 for all datasets, respectively. Regarding the victim models in individual fairness attacks, we train a 2-layer GCN (Kipf & Welling, 2017) and 2-layer InFoRM-GNN (Kang et al., 2020; Dong et al., 2021) for 400 epochs. The hidden dimension, learning rate, weight decay and dropout rate of GCN and InFoRM-GNN are set to 128, $1e-3$, $1e-5$ and 0.5, respectively. The regularization parameter in InFoRM-GNN is set to 0.1 for all datasets.

## D  ADDITIONAL EXPERIMENTAL RESULTS: ATTACKING STATISTICAL PARITY ON GRAPH NEURAL NETWORKS

**A – FATE with FairGNN as the victim model.** Here, we study how robust FairGNN is in fairness attacks against statistical parity with linear GCN as the surrogate model. Note that FairGNN is a fairness-aware graph neural network that leverages adversarial learning to ensure statistical parity.

**Main results.** Similar to Section 6.1, for FATE, we conduct fairness attacks via both edge flipping (FATE-flip) and edge addition (FATE-add). For all other baseline methods, edges are only added. From Table 5, we have the following key observations: (1) Even though the surrogate model is linear GCN without fairness consideration, FairGNN, which ensures statistical parity on graph neural networks, cannot mitigate the bias caused by fairness attacks and is vulnerable to fairness attack. (2) FATE-flip and FATE-add are effective and the most deceptive method in fairness attacks. (3) DICE-S, FATE-flip, and FATE-add are all capable of successful fairness attacks. But FATE-flip and FATE-add have better utility than DICE-S, making the fairness attacks more deceptive. Both Random and FA-GNN fail in some cases (indicated by the underlined $\Delta_{\mathrm{SP}}$ in both tables). In short, even when the victim model is FairGNN (a fair graph neural network), our proposed FATE framework are effective in fairness attacks while being the most deceptive (i.e., highest micro F1 score).

**Effect of the perturbation rate.** From Table 5, we can find out that: (1) $\Delta_{\mathrm{SP}}$ tends to increase when the perturbation rate increases, indicating the effectiveness of FATE-flip and FATE-add for attacking fairness. (2) There is no clear correlation between the perturbation rate and the micro F1 scores of FATE-flip and FATE-add, meaning that they are deceptive in maintaining the utility. As a consequence, FATE is effective and deceptive in attacking fairness of FairGNN across different perturbation rates.

**B – Performance evaluation under different utility metrics.** Here we provide additional evaluation results of utility using macro F1 score and AUC score. From Tables 6 and 7, we can see that macro F1 scores and AUC scores are less impacted by different perturbation rates. Thus, it provide additional evidence that FATE can achieve deceptive fairness attacks by achieving comparable or even better utility on the semi-supervised node classification.

Table 5: Attacking statistical parity on FairGNN under different perturbation rates (Ptb.). FATE poisons the graph via both edge flipping (FATE-flip) and edge addition (FATE-add) while all other baselines poison the graph via edge addition. Higher is better (↑) for micro F1 score (Micro F1) and $\Delta_{SP}$ (bias). Bold font indicates the most deceptive fairness attack, i.e., increasing $\Delta_{SP}$ and highest micro F1. Underlined cell indicates the failure of fairness attack, i.e., decreasing $\Delta_{SP}$ after fairness attack.

| Dataset | Ptb. | Random | | DICE-S | | FA-GNN | | FATE-flip | | FATE-add | |
|---|---|---|---|---|---|---|---|---|---|---|---|
| | | Micro F1 (↑) | $\Delta_{SP}$ (↑) | Micro F1 (↑) | $\Delta_{SP}$ (↑) | Micro F1 (↑) | $\Delta_{SP}$ (↑) | Micro F1 (↑) | $\Delta_{SP}$ (↑) | Micro F1 (↑) | $\Delta_{SP}$ (↑) |
| Pokec-n | 0.00 | 68.2 ± 0.4 | 6.7 ± 2.0 | 68.2 ± 0.4 | 6.7 ± 2.0 | 68.2 ± 0.4 | 6.7 ± 2.0 | 68.2 ± 0.4 | 6.7 ± 2.0 | 68.2 ± 0.4 | 6.7 ± 2.0 |
| | 0.05 | 67.4 ± 0.8 | 8.2 ± 2.5 | 66.9 ± 0.9 | 7.4 ± 1.7 | 66.7 ± 1.2 | 2.8 ± 1.3 | 68.4 ± 0.2 | 8.9 ± 1.8 | 68.4 ± 0.2 | 8.9 ± 1.8 |
| | 0.10 | 67.5 ± 0.5 | 8.3 ± 1.5 | 67.6 ± 0.3 | 8.4 ± 1.2 | 66.6 ± 0.5 | 5.9 ± 1.3 | 68.5 ± 0.4 | 9.5 ± 1.4 | 68.5 ± 0.4 | 9.5 ± 1.4 |
| | 0.15 | 65.9 ± 0.6 | 10.4 ± 2.3 | 67.3 ± 0.3 | 9.9 ± 2.4 | 64.8 ± 1.6 | 9.0 ± 3.3 | 68.5 ± 0.8 | 10.5 ± 2.6 | 68.5 ± 0.8 | 10.5 ± 2.6 |
| | 0.20 | 65.4 ± 0.5 | 10.0 ± 1.5 | 66.5 ± 0.4 | 9.0 ± 2.3 | 65.2 ± 0.2 | 11.6 ± 2.6 | 68.3 ± 0.3 | 10.7 ± 2.3 | 68.3 ± 0.3 | 10.7 ± 2.3 |
| | 0.25 | 65.8 ± 1.1 | 7.5 ± 1.9 | 66.5 ± 0.8 | 9.7 ± 3.0 | 64.8 ± 0.8 | 14.2 ± 2.3 | 68.5 ± 0.3 | 9.1 ± 3.6 | 68.5 ± 0.3 | 9.1 ± 3.6 |
| Pokec-z | 0.00 | 68.7 ± 0.3 | 7.0 ± 0.9 | 68.7 ± 0.3 | 7.0 ± 0.9 | 68.7 ± 0.3 | 7.0 ± 0.9 | 68.7 ± 0.3 | 7.0 ± 0.9 | 68.7 ± 0.3 | 7.0 ± 0.9 |
| | 0.05 | 67.3 ± 0.6 | 8.7 ± 2.8 | 68.0 ± 0.7 | 9.4 ± 4.1 | 67.1 ± 1.0 | 1.7 ± 1.3 | 68.7 ± 0.4 | 8.0 ± 0.9 | 68.7 ± 0.4 | 8.0 ± 0.9 |
| | 0.10 | 67.1 ± 0.2 | 8.6 ± 2.7 | 68.1 ± 0.5 | 8.2 ± 5.0 | 65.9 ± 0.8 | 6.8 ± 1.7 | 68.5 ± 0.5 | 9.0 ± 1.8 | 68.5 ± 0.5 | 9.0 ± 1.8 |
| | 0.15 | 66.8 ± 0.8 | 8.9 ± 2.2 | 67.6 ± 0.6 | 9.6 ± 3.4 | 64.9 ± 0.9 | 10.0 ± 1.7 | 68.7 ± 0.5 | 9.5 ± 2.2 | 68.7 ± 0.5 | 9.5 ± 2.2 |
| | 0.20 | 66.8 ± 0.7 | 8.6 ± 3.0 | 67.4 ± 0.7 | 9.1 ± 4.9 | 64.6 ± 0.8 | 14.2 ± 3.1 | 68.8 ± 0.2 | 10.4 ± 1.6 | 68.8 ± 0.2 | 10.4 ± 1.6 |
| | 0.25 | 66.4 ± 0.4 | 7.9 ± 2.8 | 67.1 ± 0.6 | 8.7 ± 4.3 | 64.0 ± 1.1 | 14.0 ± 2.0 | 68.5 ± 0.3 | 10.3 ± 2.1 | 68.5 ± 0.3 | 10.3 ± 2.1 |
| Bail | 0.00 | 93.9 ± 0.1 | 8.4 ± 0.2 | 93.9 ± 0.1 | 8.4 ± 0.2 | 93.9 ± 0.1 | 8.4 ± 0.2 | 93.9 ± 0.1 | 8.4 ± 0.2 | 93.9 ± 0.1 | 8.4 ± 0.2 |
| | 0.05 | 90.6 ± 1.2 | 8.3 ± 0.2 | 90.5 ± 1.0 | 8.9 ± 0.5 | 89.1 ± 2.0 | 10.8 ± 1.1 | 93.6 ± 0.1 | 9.2 ± 0.2 | 93.6 ± 0.1 | 9.1 ± 0.2 |
| | 0.10 | 90.1 ± 2.0 | 8.5 ± 0.6 | 90.1 ± 1.0 | 8.6 ± 0.2 | 87.3 ± 2.2 | 12.2 ± 1.2 | 93.4 ± 0.1 | 9.3 ± 0.2 | 93.4 ± 0.1 | 9.3 ± 0.2 |
| | 0.15 | 90.0 ± 2.0 | 8.1 ± 0.5 | 90.6 ± 1.7 | 9.5 ± 0.6 | 87.8 ± 2.0 | 10.9 ± 2.1 | 93.3 ± 0.1 | 9.2 ± 0.3 | 93.3 ± 0.1 | 9.2 ± 0.3 |
| | 0.20 | 89.2 ± 2.4 | 8.4 ± 0.7 | 90.0 ± 1.7 | 9.9 ± 0.6 | 86.0 ± 2.7 | 11.7 ± 2.4 | 93.1 ± 0.2 | 9.3 ± 0.3 | 93.0 ± 0.1 | 9.4 ± 0.2 |
| | 0.25 | 88.8 ± 2.3 | 8.2 ± 0.7 | 89.9 ± 1.8 | 9.6 ± 0.5 | 87.0 ± 1.9 | 8.5 ± 2.6 | 93.0 ± 0.1 | 9.2 ± 0.4 | 93.0 ± 0.2 | 9.3 ± 0.3 |

Table 6: Macro F1 score and AUC score of attacking statistical parity on GCN under different perturbation rates (Ptb.). FATE poisons the graph via both edge flipping (FATE-flip) and edge addition (FATE-add) while all other baselines poison the graph via edge addition. Higher is better (↑) for macro F1 score (Macro F1) and AUC score (AUC). Bold font indicates the highest macro F1 score or AUC score.

| Dataset | Ptb. | Random | | DICE-S | | FA-GNN | | FATE-flip | | FATE-add | |
|---|---|---|---|---|---|---|---|---|---|---|---|
| | | Macro F1 (↑) | AUC (↑) | Macro F1 (↑) | AUC (↑) | Macro F1 (↑) | AUC (↑) | Macro F1 (↑) | AUC (↑) | Macro F1 (↑) | AUC (↑) |
| Pokec-n | 0.00 | 65.3 ± 0.3 | 69.9 ± 0.5 | 65.3 ± 0.3 | 69.9 ± 0.5 | 65.3 ± 0.3 | 69.9 ± 0.5 | 65.3 ± 0.3 | 69.9 ± 0.5 | 65.3 ± 0.3 | 69.9 ± 0.5 |
| | 0.05 | 65.7 ± 0.3 | 70.4 ± 0.4 | 65.4 ± 0.3 | 70.3 ± 0.3 | 64.9 ± 0.2 | 70.4 ± 0.2 | 66.0 ± 0.3 | 70.3 ± 0.6 | 66.0 ± 0.3 | 70.3 ± 0.6 |
| | 0.10 | 64.6 ± 0.4 | 69.6 ± 0.3 | 65.7 ± 0.2 | 70.2 ± 0.2 | 64.1 ± 0.3 | 70.0 ± 0.1 | 66.1 ± 0.6 | 70.4 ± 0.6 | 66.1 ± 0.6 | 70.4 ± 0.6 |
| | 0.15 | 65.1 ± 0.4 | 69.6 ± 0.1 | 64.9 ± 0.3 | 69.0 ± 0.3 | 64.3 ± 0.6 | 69.1 ± 0.5 | 66.1 ± 0.2 | 70.6 ± 0.6 | 66.1 ± 0.2 | 70.6 ± 0.6 |
| | 0.20 | 64.5 ± 0.5 | 69.1 ± 0.1 | 64.2 ± 0.3 | 68.7 ± 0.4 | 63.5 ± 0.2 | 68.0 ± 0.2 | 66.4 ± 0.3 | 70.7 ± 0.4 | 66.4 ± 0.3 | 70.7 ± 0.4 |
| | 0.25 | 64.5 ± 0.6 | 68.8 ± 0.1 | 63.7 ± 0.2 | 68.8 ± 0.2 | 65.0 ± 0.2 | 69.5 ± 0.3 | 66.3 ± 0.3 | 70.6 ± 0.6 | 66.3 ± 0.3 | 70.6 ± 0.6 |
| Pokec-z | 0.00 | 68.2 ± 0.4 | 75.1 ± 0.3 | 68.2 ± 0.4 | 75.1 ± 0.3 | 68.2 ± 0.4 | 75.1 ± 0.3 | 68.2 ± 0.4 | 75.1 ± 0.3 | 68.2 ± 0.4 | 75.1 ± 0.3 |
| | 0.05 | 68.5 ± 0.4 | 74.5 ± 0.4 | 68.7 ± 0.3 | 75.4 ± 0.4 | 67.9 ± 0.3 | 74.5 ± 0.2 | 68.6 ± 0.4 | 75.2 ± 0.4 | 68.6 ± 0.4 | 75.2 ± 0.4 |
| | 0.10 | 68.5 ± 0.3 | 74.8 ± 0.3 | 67.6 ± 0.2 | 74.5 ± 0.3 | 67.5 ± 0.5 | 73.8 ± 0.3 | 68.6 ± 0.6 | 75.2 ± 0.3 | 68.6 ± 0.6 | 75.2 ± 0.3 |
| | 0.15 | 67.8 ± 0.3 | 74.4 ± 0.3 | 67.6 ± 0.4 | 74.1 ± 0.4 | 66.1 ± 0.6 | 72.7 ± 0.2 | 68.9 ± 0.7 | 75.3 ± 0.2 | 68.9 ± 0.7 | 75.3 ± 0.2 |
| | 0.20 | 68.2 ± 0.4 | 74.5 ± 0.6 | 66.8 ± 0.5 | 73.6 ± 0.3 | 66.1 ± 0.2 | 71.9 ± 0.1 | 68.4 ± 0.5 | 75.1 ± 0.3 | 68.4 ± 0.5 | 75.1 ± 0.3 |
| | 0.25 | 68.0 ± 0.4 | 74.4 ± 0.4 | 67.1 ± 0.7 | 74.4 ± 0.3 | 65.6 ± 0.3 | 71.2 ± 0.3 | 68.4 ± 1.1 | 74.4 ± 1.4 | 68.4 ± 1.1 | 74.4 ± 1.4 |
| Bail | 0.00 | 92.3 ± 0.2 | 97.4 ± 0.1 | 92.3 ± 0.2 | 97.4 ± 0.1 | 92.3 ± 0.2 | 97.4 ± 0.1 | 92.3 ± 0.2 | 97.4 ± 0.1 | 92.3 ± 0.2 | 97.4 ± 0.1 |
| | 0.05 | 92.0 ± 0.2 | 95.3 ± 0.2 | 91.4 ± 0.3 | 95.1 ± 0.4 | 90.8 ± 0.1 | 94.4 ± 0.2 | 91.8 ± 0.1 | 97.1 ± 0.1 | 91.7 ± 0.1 | 97.1 ± 0.2 |
| | 0.10 | 91.4 ± 0.2 | 94.7 ± 0.3 | 91.4 ± 0.3 | 94.7 ± 0.4 | 89.5 ± 0.1 | 93.5 ± 0.1 | 91.6 ± 0.2 | 96.9 ± 0.1 | 91.6 ± 0.2 | 96.9 ± 0.1 |
| | 0.15 | 91.1 ± 0.2 | 94.2 ± 0.2 | 91.2 ± 0.2 | 94.5 ± 0.2 | 88.7 ± 0.3 | 92.5 ± 0.2 | 91.4 ± 0.2 | 96.9 ± 0.1 | 91.5 ± 0.1 | 96.9 ± 0.1 |
| | 0.20 | 90.7 ± 0.2 | 94.1 ± 0.1 | 90.9 ± 0.2 | 94.4 ± 0.3 | 88.4 ± 0.1 | 92.2 ± 0.1 | 91.3 ± 0.2 | 96.8 ± 0.1 | 91.4 ± 0.2 | 96.8 ± 0.1 |
| | 0.25 | 90.4 ± 0.2 | 93.4 ± 0.3 | 90.6 ± 0.3 | 94.3 ± 0.3 | 88.5 ± 0.2 | 92.0 ± 0.1 | 91.2 ± 0.1 | 96.8 ± 0.1 | 91.3 ± 0.2 | 96.8 ± 0.1 |

Table 7: Macro F1 score and AUC score of attacking statistical parity on FairGNN under different perturbation rates (Ptb.). FATE poisons the graph via both edge flipping (FATE-flip) and edge addition (FATE-add) while all other baselines poison the graph via edge addition. Higher is better (↑) for macro F1 score (Macro F1) and AUC score (AUC). Bold font indicates the highest macro F1 score or AUC score.

| Dataset | Ptb. | Random | | DICE-S | | FA-GNN | | FATE-flip | | FATE-add | |
|---|---|---|---|---|---|---|---|---|---|---|---|
| | | Macro F1 (↑) | AUC (↑) | Macro F1 (↑) | AUC (↑) | Macro F1 (↑) | AUC (↑) | Macro F1 (↑) | AUC (↑) | Macro F1 (↑) | AUC (↑) |
| Pokec-n | 0.00 | 65.6 ± 0.3 | 70.4 ± 0.5 | 65.6 ± 0.3 | 70.4 ± 0.5 | 65.6 ± 0.3 | 70.4 ± 0.5 | 65.6 ± 0.3 | 70.4 ± 0.5 | 65.6 ± 0.3 | 70.4 ± 0.5 |
| | 0.05 | 64.3 ± 0.6 | 68.3 ± 1.1 | 64.5 ± 0.4 | 69.5 ± 0.8 | 63.6 ± 0.7 | 68.2 ± 0.5 | 65.8 ± 0.5 | 70.7 ± 0.4 | 65.8 ± 0.5 | 70.7 ± 0.4 |
| | 0.10 | 63.8 ± 0.2 | 67.3 ± 1.1 | 64.3 ± 0.7 | 69.6 ± 0.4 | 63.9 ± 0.4 | 68.3 ± 0.2 | 66.0 ± 0.2 | 70.8 ± 0.5 | 66.0 ± 0.2 | 70.8 ± 0.5 |
| | 0.15 | 63.5 ± 0.2 | 67.8 ± 0.4 | 64.1 ± 0.7 | 68.5 ± 0.4 | 63.1 ± 0.6 | 67.2 ± 0.5 | 65.8 ± 1.0 | 70.8 ± 0.5 | 65.8 ± 1.0 | 70.8 ± 0.5 |
| | 0.20 | 63.1 ± 0.6 | 67.8 ± 1.1 | 62.4 ± 1.5 | 67.5 ± 1.1 | 62.3 ± 0.6 | 66.7 ± 0.9 | 65.7 ± 0.7 | 70.4 ± 0.5 | 65.7 ± 0.7 | 70.4 ± 0.5 |
| | 0.25 | 62.4 ± 0.3 | 66.8 ± 0.8 | 62.4 ± 1.6 | 67.2 ± 0.9 | 62.4 ± 1.4 | 67.6 ± 1.3 | 65.1 ± 1.2 | 70.1 ± 0.5 | 65.1 ± 1.2 | 70.1 ± 0.5 |
| Pokec-z | 0.00 | 68.4 ± 0.4 | 75.1 ± 0.3 | 68.4 ± 0.4 | 75.1 ± 0.3 | 68.4 ± 0.4 | 75.1 ± 0.3 | 68.4 ± 0.4 | 75.1 ± 0.3 | 68.4 ± 0.4 | 75.1 ± 0.3 |
| | 0.05 | 66.3 ± 0.9 | 73.5 ± 0.9 | 67.2 ± 0.7 | 73.9 ± 1.5 | 66.5 ± 1.4 | 72.6 ± 1.4 | 68.4 ± 0.4 | 74.7 ± 0.9 | 68.4 ± 0.4 | 74.7 ± 0.9 |
| | 0.10 | 66.0 ± 0.7 | 72.9 ± 1.1 | 67.1 ± 0.5 | 73.4 ± 0.2 | 65.2 ± 0.9 | 71.3 ± 1.7 | 68.2 ± 0.8 | 75.3 ± 0.8 | 68.2 ± 0.8 | 75.3 ± 0.8 |
| | 0.15 | 66.0 ± 0.8 | 71.8 ± 2.1 | 66.5 ± 0.9 | 73.4 ± 0.6 | 63.4 ± 1.5 | 70.0 ± 1.8 | 68.3 ± 0.5 | 75.2 ± 0.6 | 68.3 ± 0.5 | 75.2 ± 0.6 |
| | 0.20 | 65.6 ± 0.9 | 71.9 ± 1.4 | 66.4 ± 1.0 | 73.0 ± 0.8 | 63.7 ± 0.9 | 68.9 ± 1.6 | 68.3 ± 0.3 | 75.5 ± 0.3 | 68.3 ± 0.3 | 75.5 ± 0.3 |
| | 0.25 | 65.0 ± 0.7 | 71.2 ± 1.7 | 66.3 ± 1.0 | 73.3 ± 0.8 | 62.8 ± 1.8 | 69.4 ± 1.5 | 68.0 ± 0.5 | 75.3 ± 0.3 | 68.0 ± 0.5 | 75.3 ± 0.3 |
| Bail | 0.00 | 93.3 ± 0.2 | 97.4 ± 0.1 | 93.3 ± 0.2 | 97.4 ± 0.1 | 93.3 ± 0.2 | 97.4 ± 0.1 | 93.3 ± 0.2 | 97.4 ± 0.1 | 93.3 ± 0.2 | 97.4 ± 0.1 |
| | 0.05 | 89.5 ± 1.5 | 92.8 ± 1.8 | 89.5 ± 1.1 | 92.3 ± 1.7 | 87.8 ± 2.2 | 91.2 ± 1.8 | 93.0 ± 0.1 | 97.3 ± 0.1 | 93.0 ± 0.1 | 97.3 ± 0.1 |
| | 0.10 | 89.1 ± 2.2 | 92.7 ± 2.5 | 88.8 ± 1.3 | 92.3 ± 1.6 | 85.6 ± 2.7 | 90.5 ± 1.7 | 92.7 ± 0.1 | 97.1 ± 0.1 | 92.7 ± 0.1 | 97.1 ± 0.1 |
| | 0.15 | 88.8 ± 2.2 | 92.4 ± 2.5 | 89.6 ± 1.9 | 92.8 ± 2.2 | 86.1 ± 2.4 | 90.3 ± 2.2 | 92.6 ± 0.1 | 97.0 ± 0.1 | 92.6 ± 0.1 | 97.0 ± 0.1 |
| | 0.20 | 87.8 ± 2.8 | 91.6 ± 2.5 | 88.9 ± 1.8 | 92.2 ± 1.6 | 84.1 ± 3.0 | 89.0 ± 1.5 | 92.5 ± 0.2 | 97.0 ± 0.1 | 92.3 ± 0.1 | 97.0 ± 0.1 |
| | 0.25 | 87.5 ± 2.6 | 91.5 ± 2.6 | 88.7 ± 2.1 | 92.5 ± 2.3 | 85.1 ± 2.3 | 89.6 ± 1.3 | 92.3 ± 0.1 | 97.0 ± 0.1 | 92.3 ± 0.2 | 97.0 ± 0.1 |

# E    ADDITIONAL EXPERIMENTAL RESULTS: ATTACKING INDIVIDUAL FAIRNESS ON GRAPH NEURAL NETWORKS

**A – FATE with InFoRM-GNN as the victim model.** InFoRM-GNN is an individually fair graph neural network that ensures individual fairness through regularizing the individual bias measure defined in Section 5. Here, we study how robust InFoRM-GNN is in fairness attacks against individual fairness with linear GCN as the surrogate model.

**Main results.** We attack individual fairness using FATE via both edge flipping (FATE-flip) and edge addition (FATE-add), whereas edges are only added for all other baseline methods. From Table 8, we can see that: (1) for Pokec-n and Pokec-z, FATE-flip and FATE-add are effective: they are the only methods that could consistently attack individual fairness across different perturbation rates; FATE-flip and FATE-add are deceptive by achieving comparable or higher micro F1 scores compared with the micro F1 score on the benign graph (when perturbation rate is 0.00). (2) For Bail, almost all methods fail the fairness attacks, except for FA-GNN with perturbation rates 0.20 and 0.25. A possible reason is that the adjacency matrix **A** of *Bail* is essentially a similarity graph, which causes pairwise node similarity matrix **S** being close to the adjacency matrix **A**. Even though FATE and other baseline methods add adversarial edges to attack individual fairness, regularizing the individual bias defined by **S** (a) not only helps to ensure individual fairness (b) but also provide useful supervision signal in learning a representative node representation due to the closeness between **S** and **A**. (3) Compared with the results in Table 2 where GCN is the victim model, InFoRM-GNN is more robust against fairness attacks against individual fairness due to smaller individual bias in Table 8.

**Effect of the perturbation rate.** From Table 8, we can see that FATE can always achieve comparable or even better micro F1 scores across different perturbation rates. In the meanwhile, the correlation between the perturbation rate and the individual bias is relatively weak. One possible reason is that the individual bias is computed using the pairwise node similarity matrix, which is not impacted by poisoning the adjacency matrix. Though poisoning the adjacency matrix could affect the learning results, the goal of achieving deceptive fairness attacks (i.e., the lower-level optimization problem in FATE) may not cause the learning results obtained by training on the benign graph to deviate much from the learning results obtained by training on the poisoned graph. Consequently, a higher perturbation rate may have less impact on the computation of individual bias.

Table 8: Attacking individual fairness on InFoRM-GNN under different perturbation rates (Ptb.). FATE poisons the graph via both edge flipping (FATE-flip) and edge addition (FATE-add) while all other baselines poison the graph via edge addition. Higher is better ($\uparrow$) for micro F1 score (Micro F1) and InFoRM bias (Bias). Bold font indicates the most deceptive fairness attack, i.e., increasing bias and highest micro F1. Underlined cell indicates the failure of fairness attack, i.e., decreasing bias after fairness attack.

| Dataset | Ptb. | Random | | DICE-S | | FA-GNN | | FATE-flip | | FATE-add | |
|---|---|---|---|---|---|---|---|---|---|---|---|
| | | Micro F1 ($\uparrow$) | Bias ($\uparrow$) | Micro F1 ($\uparrow$) | Bias ($\uparrow$) | Micro F1 ($\uparrow$) | Bias ($\uparrow$) | Micro F1 ($\uparrow$) | Bias ($\uparrow$) | Micro F1 ($\uparrow$) | Bias ($\uparrow$) |
| Pokec-n | 0.00 | 68.0±0.4 | 0.5±0.1 | 68.0±0.4 | 0.5±0.1 | 68.0±0.4 | 0.5±0.1 | 68.0±0.4 | 0.5±0.1 | 68.0±0.4 | 0.5±0.1 |
| | 0.05 | 67.3±0.5 | 0.5±0.0 | 68.0±0.4 | 0.5±0.1 | 68.3±0.2 | 0.5±0.0 | **68.4±0.4** | **0.6±0.1** | 68.3±0.4 | 0.5±0.1 |
| | 0.10 | 67.0±0.2 | 0.5±0.1 | 67.4±0.4 | 0.5±0.1 | 67.2±0.2 | 0.4±0.0 | 68.3±0.6 | 0.5±0.1 | **68.4±0.5** | **0.6±0.1** |
| | 0.15 | 66.7±0.5 | 0.5±0.1 | 67.7±0.4 | 0.4±0.1 | 66.1±0.2 | 0.4±0.0 | 68.3±0.6 | 0.6±0.1 | 68.1±0.7 | 0.6±0.1 |
| | 0.20 | 66.9±0.3 | 0.4±0.1 | 67.2±0.2 | 0.5±0.1 | 66.5±0.2 | 0.4±0.0 | 67.9±0.8 | 0.5±0.1 | **68.1±0.7** | **0.6±0.1** |
| | 0.25 | 66.6±0.5 | 0.5±0.0 | 66.7±0.6 | 0.5±0.1 | 65.1±0.2 | 0.4±0.0 | **68.7±0.3** | **0.6±0.0** | 68.5±0.8 | 0.6±0.1 |
| Pokec-z | 0.00 | 68.4±0.5 | 0.5±0.0 | 68.4±0.5 | 0.5±0.0 | 68.4±0.5 | 0.5±0.0 | 68.4±0.5 | 0.5±0.0 | 68.4±0.5 | 0.5±0.0 |
| | 0.05 | 68.9±0.2 | 0.6±0.1 | 68.9±0.5 | 0.5±0.1 | 68.1±0.7 | 0.5±0.1 | 68.7±0.7 | 0.7±0.1 | **68.9±0.5** | **0.6±0.0** |
| | 0.10 | 67.9±0.2 | 0.6±0.1 | **69.0±0.1** | **0.6±0.1** | 68.0±0.6 | 0.5±0.0 | 68.9±0.6 | 0.6±0.0 | 68.8±0.6 | 0.6±0.0 |
| | 0.15 | 67.6±0.3 | 0.6±0.1 | 68.2±0.5 | 0.6±0.1 | 66.8±0.3 | 0.5±0.1 | **69.1±0.5** | **0.6±0.0** | 69.0±0.7 | 0.6±0.1 |
| | 0.20 | 67.7±0.5 | 0.6±0.1 | 68.5±0.2 | 0.5±0.0 | 66.4±0.6 | 0.4±0.1 | 69.1±0.2 | 0.6±0.0 | **69.3±0.3** | **0.6±0.0** |
| | 0.25 | 66.8±0.4 | 0.5±0.1 | 68.5±0.2 | 0.5±0.0 | 65.3±0.4 | 0.4±0.0 | 68.9±0.7 | 0.6±0.0 | **69.4±0.4** | **0.6±0.0** |
| Bail | 0.00 | 92.8±0.1 | 1.7±0.1 | 92.8±0.1 | 1.7±0.1 | 92.8±0.1 | 1.7±0.1 | 92.8±0.1 | 1.7±0.1 | 92.8±0.1 | 1.7±0.1 |
| | 0.05 | 91.9±0.1 | 0.4±0.0 | 92.1±0.1 | 1.7±0.0 | 91.3±0.1 | 1.5±0.1 | 92.8±0.3 | 1.7±0.1 | 92.7±0.1 | 1.6±0.1 |
| | 0.10 | 91.7±0.1 | 0.3±0.0 | 92.0±0.1 | 1.6±0.1 | 90.4±0.2 | 1.5±0.1 | 92.8±0.1 | 1.6±0.0 | 92.8±0.1 | 1.6±0.0 |
| | 0.15 | 91.5±0.1 | 0.3±0.0 | 91.9±0.1 | 1.6±0.1 | 90.0±0.1 | 1.7±0.1 | 92.8±0.0 | 1.6±0.1 | 92.8±0.1 | 1.6±0.0 |
| | 0.20 | 91.5±0.1 | 0.3±0.0 | 91.8±0.1 | 1.7±0.0 | **89.1±0.1** | **1.7±0.1** | 92.8±0.1 | 1.6±0.0 | 92.7±0.1 | 1.5±0.1 |
| | 0.25 | 91.1±0.2 | 0.3±0.0 | 91.5±0.1 | 1.6±0.0 | **88.9±0.1** | **1.8±0.1** | 92.6±0.1 | 1.6±0.1 | 92.7±0.0 | 1.6±0.1 |

**B – Performance evaluation under different utility metrics.** Similar to Appendix D, we provide additional results on evaluating the utility of FATE in attacking individual fairness with macro F1 score and AUC score. From Tables 9 and 10, we can draw a conclusion that FATE can achieve comparable or even better macro F1 scores and AUC scores for both GCN and InFoRM-GNN across different perturbation rates. It further proves the ability of FATE on deceptive fairness attacks in the task of semi-supervised node classification.

Table 9: Macro F1 score and AUC score of attacking individual fairness on GCN under different perturbation rates (Ptb.). FATE poisons the graph via both edge flipping (FATE-flip) and edge addition (FATE-add) while all other baselines poison the graph via edge addition. Higher is better (↑) for macro F1 score (Macro F1) and AUC score (AUC). Bold font indicates the highest macro F1 score or AUC score.

| Dataset | Ptb. | Random | | DICE-S | | FA-GNN | | FATE-flip | | FATE-add | |
|---|---|---|---|---|---|---|---|---|---|---|---|
| | | Macro F1 (↑) | AUC (↑) | Macro F1 (↑) | AUC (↑) | Macro F1 (↑) | AUC (↑) | Macro F1 (↑) | AUC (↑) | Macro F1 (↑) | AUC (↑) |
| Pokec-n | 0.00 | $65.3 \pm 0.3$ | $69.9 \pm 0.5$ | $65.3 \pm 0.3$ | $69.9 \pm 0.5$ | $65.3 \pm 0.3$ | $69.9 \pm 0.5$ | $65.3 \pm 0.3$ | $69.9 \pm 0.5$ | $65.3 \pm 0.3$ | $69.9 \pm 0.5$ |
| | 0.05 | $65.2 \pm 0.3$ | $70.1 \pm 0.2$ | $65.7 \pm 0.3$ | $70.2 \pm 0.2$ | $65.6 \pm 0.6$ | $\mathbf{71.1 \pm 0.2}$ | $\mathbf{65.7 \pm 0.4}$ | $70.1 \pm 0.6$ | $65.5 \pm 0.3$ | $70.2 \pm 0.8$ |
| | 0.10 | $65.2 \pm 0.3$ | $69.6 \pm 0.5$ | $64.7 \pm 0.5$ | $69.9 \pm 0.2$ | $65.4 \pm 0.6$ | $70.2 \pm 0.3$ | $65.5 \pm 0.3$ | $70.2 \pm 0.7$ | $\mathbf{65.8 \pm 0.5}$ | $\mathbf{70.7 \pm 0.6}$ |
| | 0.15 | $65.4 \pm 0.2$ | $69.4 \pm 0.3$ | $64.9 \pm 0.2$ | $70.1 \pm 0.4$ | $64.6 \pm 0.2$ | $69.4 \pm 0.1$ | $\mathbf{65.6 \pm 0.4}$ | $\mathbf{70.0 \pm 0.5}$ | $65.4 \pm 0.1$ | $69.8 \pm 0.7$ |
| | 0.20 | $64.9 \pm 0.2$ | $69.6 \pm 0.3$ | $65.1 \pm 0.4$ | $70.2 \pm 0.3$ | $63.7 \pm 0.5$ | $69.0 \pm 0.1$ | $65.2 \pm 0.3$ | $69.7 \pm 0.6$ | $\mathbf{65.6 \pm 0.6}$ | $\mathbf{70.2 \pm 0.7}$ |
| | 0.25 | $64.7 \pm 0.1$ | $69.4 \pm 0.2$ | $64.1 \pm 0.1$ | $69.4 \pm 0.2$ | $63.3 \pm 0.5$ | $68.4 \pm 0.3$ | $65.4 \pm 0.6$ | $69.7 \pm 0.7$ | $\mathbf{65.6 \pm 0.8}$ | $\mathbf{69.8 \pm 0.8}$ |
| Pokec-z | 0.00 | $68.2 \pm 0.4$ | $75.1 \pm 0.3$ | $68.2 \pm 0.4$ | $75.1 \pm 0.3$ | $68.2 \pm 0.4$ | $75.1 \pm 0.3$ | $68.2 \pm 0.4$ | $75.1 \pm 0.3$ | $68.2 \pm 0.4$ | $75.1 \pm 0.3$ |
| | 0.05 | $\mathbf{68.7 \pm 0.4}$ | $75.0 \pm 0.4$ | $\mathbf{68.7 \pm 0.6}$ | $75.2 \pm 0.4$ | $68.0 \pm 0.4$ | $75.1 \pm 0.5$ | $68.5 \pm 0.5$ | $\mathbf{75.4 \pm 0.2}$ | $68.5 \pm 0.3$ | $75.2 \pm 0.4$ |
| | 0.10 | $68.5 \pm 0.1$ | $75.1 \pm 0.5$ | $\mathbf{68.9 \pm 0.2}$ | $75.3 \pm 0.1$ | $67.9 \pm 0.6$ | $74.4 \pm 0.5$ | $68.8 \pm 0.5$ | $75.5 \pm 0.3$ | $68.8 \pm 0.4$ | $\mathbf{75.6 \pm 0.2}$ |
| | 0.15 | $67.5 \pm 0.4$ | $74.4 \pm 0.3$ | $67.9 \pm 0.3$ | $73.8 \pm 0.1$ | $66.8 \pm 0.4$ | $72.6 \pm 0.2$ | $68.4 \pm 0.5$ | $75.5 \pm 0.4$ | $\mathbf{68.8 \pm 0.7}$ | $\mathbf{75.6 \pm 0.3}$ |
| | 0.20 | $67.5 \pm 0.4$ | $74.7 \pm 0.4$ | $67.7 \pm 0.3$ | $74.7 \pm 0.2$ | $66.1 \pm 0.1$ | $71.8 \pm 0.2$ | $68.7 \pm 0.5$ | $75.5 \pm 0.3$ | $\mathbf{69.0 \pm 0.4}$ | $\mathbf{75.6 \pm 0.3}$ |
| | 0.25 | $67.2 \pm 0.3$ | $74.1 \pm 0.3$ | $68.0 \pm 0.5$ | $74.7 \pm 0.2$ | $64.8 \pm 0.4$ | $70.5 \pm 0.4$ | $68.9 \pm 0.3$ | $75.6 \pm 0.2$ | $\mathbf{69.1 \pm 0.3}$ | $\mathbf{75.7 \pm 0.3}$ |
| Bail | 0.00 | $92.3 \pm 0.2$ | $97.4 \pm 0.1$ | $92.3 \pm 0.2$ | $97.4 \pm 0.1$ | $92.3 \pm 0.2$ | $97.4 \pm 0.1$ | $92.3 \pm 0.2$ | $97.4 \pm 0.1$ | $92.3 \pm 0.2$ | $97.4 \pm 0.1$ |
| | 0.05 | $91.2 \pm 0.3$ | $94.8 \pm 0.2$ | $91.4 \pm 0.2$ | $95.0 \pm 0.3$ | $90.3 \pm 0.2$ | $94.1 \pm 0.2$ | $\mathbf{92.3 \pm 0.4}$ | $97.3 \pm 0.1$ | $92.1 \pm 0.3$ | $\mathbf{97.3 \pm 0.1}$ |
| | 0.10 | $90.6 \pm 0.1$ | $94.2 \pm 0.3$ | $91.3 \pm 0.2$ | $94.9 \pm 0.4$ | $89.1 \pm 0.1$ | $92.9 \pm 0.3$ | $\mathbf{92.3 \pm 0.1}$ | $\mathbf{97.3 \pm 0.4}$ | $92.2 \pm 0.2$ | $\mathbf{97.3 \pm 0.1}$ |
| | 0.15 | $90.3 \pm 0.1$ | $94.1 \pm 0.2$ | $91.3 \pm 0.3$ | $94.7 \pm 0.3$ | $88.6 \pm 0.2$ | $92.4 \pm 0.3$ | $\mathbf{92.4 \pm 0.1}$ | $\mathbf{97.3 \pm 0.0}$ | $92.3 \pm 0.2$ | $\mathbf{97.3 \pm 0.1}$ |
| | 0.20 | $90.2 \pm 0.0$ | $93.9 \pm 0.1$ | $90.9 \pm 0.2$ | $94.2 \pm 0.2$ | $87.9 \pm 0.2$ | $91.8 \pm 0.2$ | $\mathbf{92.4 \pm 0.1}$ | $\mathbf{97.3 \pm 0.0}$ | $\mathbf{92.4 \pm 0.2}$ | $\mathbf{97.3 \pm 0.1}$ |
| | 0.25 | $90.9 \pm 0.1$ | $93.5 \pm 0.2$ | $90.5 \pm 0.1$ | $94.1 \pm 0.4$ | $87.6 \pm 0.1$ | $91.6 \pm 0.2$ | $\mathbf{92.2 \pm 0.2}$ | $97.2 \pm 0.1$ | $\mathbf{92.2 \pm 0.2}$ | $\mathbf{97.3 \pm 0.1}$ |

Table 10: Macro F1 score and AUC score of attacking individual fairness on InFoRM-GNN under different perturbation rates (Ptb.). FATE poisons the graph via both edge flipping (FATE-flip) and edge addition (FATE-add) while all other baselines poison the graph via edge addition. Higher is better (↑) for macro F1 score (Macro F1) and AUC score (AUC). Bold font indicates the highest macro F1 score or AUC score.

| Dataset | Ptb. | Random | | DICE-S | | FA-GNN | | FATE-flip | | FATE-add | |
|---|---|---|---|---|---|---|---|---|---|---|---|
| | | Macro F1 (↑) | AUC (↑) | Macro F1 (↑) | AUC (↑) | Macro F1 (↑) | AUC (↑) | Macro F1 (↑) | AUC (↑) | Macro F1 (↑) | AUC (↑) |
| Pokec-n | 0.00 | $65.4 \pm 0.4$ | $70.5 \pm 0.8$ | $65.4 \pm 0.4$ | $70.5 \pm 0.8$ | $65.4 \pm 0.4$ | $70.5 \pm 0.8$ | $65.4 \pm 0.4$ | $70.5 \pm 0.8$ | $65.4 \pm 0.4$ | $70.5 \pm 0.8$ |
| | 0.05 | $65.1 \pm 0.3$ | $69.9 \pm 0.2$ | $65.7 \pm 0.1$ | $70.3 \pm 0.1$ | $65.6 \pm 0.3$ | $\mathbf{70.8 \pm 0.1}$ | $\mathbf{65.9 \pm 0.4}$ | $70.7 \pm 0.8$ | $65.8 \pm 0.5$ | $70.5 \pm 0.9$ |
| | 0.10 | $64.9 \pm 0.2$ | $69.6 \pm 0.5$ | $65.2 \pm 0.4$ | $70.2 \pm 0.2$ | $64.8 \pm 0.4$ | $69.8 \pm 0.3$ | $65.8 \pm 0.5$ | $70.3 \pm 1.0$ | $\mathbf{66.0 \pm 0.4}$ | $\mathbf{70.9 \pm 1.1}$ |
| | 0.15 | $64.8 \pm 0.4$ | $69.6 \pm 0.4$ | $65.1 \pm 0.3$ | $70.0 \pm 0.5$ | $64.4 \pm 0.1$ | $69.2 \pm 0.3$ | $65.7 \pm 0.6$ | $\mathbf{70.3 \pm 0.7}$ | $65.8 \pm 0.4$ | $70.3 \pm 0.9$ |
| | 0.20 | $65.1 \pm 0.2$ | $69.5 \pm 0.3$ | $64.9 \pm 0.5$ | $69.9 \pm 0.2$ | $63.4 \pm 0.4$ | $69.0 \pm 0.2$ | $65.5 \pm 0.8$ | $70.2 \pm 0.9$ | $\mathbf{65.6 \pm 0.6}$ | $\mathbf{70.5 \pm 0.7}$ |
| | 0.25 | $64.6 \pm 0.3$ | $69.6 \pm 0.2$ | $64.5 \pm 0.3$ | $69.4 \pm 0.2$ | $63.6 \pm 0.3$ | $68.6 \pm 0.2$ | $\mathbf{66.0 \pm 0.5}$ | $\mathbf{70.8 \pm 0.3}$ | $65.9 \pm 0.5$ | $70.4 \pm 0.8$ |
| Pokec-z | 0.00 | $68.3 \pm 0.4$ | $75.2 \pm 0.2$ | $68.3 \pm 0.4$ | $75.2 \pm 0.2$ | $68.3 \pm 0.4$ | $75.2 \pm 0.2$ | $68.3 \pm 0.4$ | $75.2 \pm 0.2$ | $68.3 \pm 0.4$ | $75.2 \pm 0.2$ |
| | 0.05 | $68.6 \pm 0.2$ | $75.1 \pm 0.3$ | $\mathbf{68.9 \pm 0.4}$ | $\mathbf{75.5 \pm 0.2}$ | $67.8 \pm 0.6$ | $75.0 \pm 0.3$ | $68.6 \pm 0.7$ | $75.1 \pm 0.6$ | $68.7 \pm 0.4$ | $75.4 \pm 0.3$ |
| | 0.10 | $67.6 \pm 0.2$ | $74.3 \pm 0.4$ | $\mathbf{68.9 \pm 0.2}$ | $75.3 \pm 0.3$ | $67.7 \pm 0.6$ | $73.9 \pm 0.6$ | $68.6 \pm 0.6$ | $\mathbf{75.6 \pm 0.3}$ | $68.6 \pm 0.6$ | $75.5 \pm 0.3$ |
| | 0.15 | $67.2 \pm 0.3$ | $74.1 \pm 0.4$ | $67.9 \pm 0.4$ | $74.4 \pm 0.3$ | $66.7 \pm 0.3$ | $72.3 \pm 0.1$ | $\mathbf{68.9 \pm 0.4}$ | $75.4 \pm 0.4$ | $68.9 \pm 0.6$ | $75.4 \pm 0.4$ |
| | 0.20 | $67.3 \pm 0.6$ | $74.4 \pm 0.4$ | $68.3 \pm 0.1$ | $75.1 \pm 0.3$ | $66.0 \pm 0.5$ | $71.7 \pm 0.2$ | $69.0 \pm 0.2$ | $\mathbf{75.5 \pm 0.2}$ | $\mathbf{69.2 \pm 0.4}$ | $75.4 \pm 0.4$ |
| | 0.25 | $66.3 \pm 0.4$ | $73.9 \pm 0.4$ | $68.2 \pm 0.3$ | $74.8 \pm 0.1$ | $65.0 \pm 0.5$ | $70.8 \pm 0.2$ | $68.8 \pm 0.7$ | $75.6 \pm 0.3$ | $\mathbf{69.3 \pm 0.4}$ | $\mathbf{75.8 \pm 0.1}$ |
| Bail | 0.00 | $91.9 \pm 0.1$ | $97.2 \pm 0.0$ | $91.9 \pm 0.1$ | $97.2 \pm 0.0$ | $91.9 \pm 0.1$ | $97.2 \pm 0.0$ | $91.9 \pm 0.1$ | $97.2 \pm 0.0$ | $91.9 \pm 0.1$ | $97.2 \pm 0.0$ |
| | 0.05 | $91.0 \pm 0.1$ | $94.2 \pm 0.2$ | $91.2 \pm 0.1$ | $95.0 \pm 0.1$ | $90.4 \pm 0.1$ | $94.2 \pm 0.1$ | $\mathbf{92.0 \pm 0.0}$ | $\mathbf{97.1 \pm 0.1}$ | $91.9 \pm 0.2$ | $97.0 \pm 0.2$ |
| | 0.10 | $90.7 \pm 0.2$ | $93.9 \pm 0.3$ | $91.1 \pm 0.1$ | $94.7 \pm 0.3$ | $89.4 \pm 0.2$ | $93.3 \pm 0.1$ | $\mathbf{92.0 \pm 0.1}$ | $\mathbf{97.0 \pm 0.0}$ | $91.9 \pm 0.1$ | $\mathbf{97.0 \pm 0.0}$ |
| | 0.15 | $90.5 \pm 0.1$ | $93.8 \pm 0.3$ | $91.0 \pm 0.2$ | $94.5 \pm 0.3$ | $88.8 \pm 0.2$ | $92.4 \pm 0.1$ | $\mathbf{92.0 \pm 0.1}$ | $\mathbf{97.0 \pm 0.0}$ | $91.9 \pm 0.1$ | $\mathbf{97.0 \pm 0.1}$ |
| | 0.20 | $90.5 \pm 0.2$ | $93.7 \pm 0.2$ | $90.8 \pm 0.1$ | $94.3 \pm 0.2$ | $87.8 \pm 0.1$ | $91.8 \pm 0.1$ | $\mathbf{92.0 \pm 0.1}$ | $96.9 \pm 0.0$ | $91.9 \pm 0.1$ | $96.8 \pm 0.1$ |
| | 0.25 | $90.1 \pm 0.2$ | $93.4 \pm 0.3$ | $90.6 \pm 0.1$ | $94.0 \pm 0.1$ | $87.4 \pm 0.1$ | $91.4 \pm 0.1$ | $91.8 \pm 0.1$ | $96.8 \pm 0.1$ | $\mathbf{91.9 \pm 0.1}$ | $\mathbf{96.9 \pm 0.0}$ |

# F    TRANFERABILITY OF FAIRNESS ATTACKS BY FATE

For the evaluation results shown in Sections 6.1 and 6.2 as well as Appendices D and E, both the surrogate model (linear GCN) and the victim models (i.e., GCN, FairGNN, InFoRM-GNN) are convolutional aggregation-based graph neural networks. In this section, we aim to test the transferability of FATE by generating poisoned graphs on the convolutional aggregation-based surrogate model (i.e., linear GCN) and testing on graph attention network (GAT), which is a non-convolutional aggregation-based graph neural network (Veličković et al., 2018).

More specifically, we train a graph attention network (GAT) with 8 attention heads for 400 epochs. The hidden dimension, learning rate, weight decay and dropout rate of GAT are set to 64, $1e - 3$, $1e - 5$ and 0.5, respectively.

The results on attacking statistical parity or individual fairness with GAT as the victim model are shown in Table 11. Even though the surrogate model used by the attacker is a convolutional aggregation-based linear GCN, from the table, it is clear that FATE can consistently succeed in (1) effective fairness attack by increasing $\Delta_{\text{SP}}$ and the individual bias (Bias) and (2) deceptive attack by offering comparable or even better micro F1 score (Micro F1) when the victim model is not a convolutional aggregation-based model. Thus, it shows that the adversarial edges flipped/added by FATE is able to transfer to graph neural networks with different type of aggregation function.

Table 11: Transferability of attacking statical parity and individual fairness with FATE on GAT under different perturbation rates (Ptb.). FATE poisons the graph via both edge flipping (FATE-flip) and edge addition (FATE-add). Higher is better (↑) for micro F1 score (Micro F1), $\Delta_{\text{SP}}$ (bias for statistical parity), and InFoRM bias (Bias, bias for individual fairness).

| | | Pokec-n | | Pokec-z | | Bail | |
|---|---|---|---|---|---|---|---|
| **Dataset** | **Ptb.** | **Micro F1 (↑)** | **$\Delta_{\text{SP}}$ (↑)** | **Micro F1 (↑)** | **$\Delta_{\text{SP}}$ (↑)** | **Micro F1 (↑)** | **$\Delta_{\text{SP}}$ (↑)** |
| | | | | | | | |
| | 0.00 | $63.8 \pm 5.3$ | $4.0 \pm 3.2$ | $68.2 \pm 0.5$ | $8.6 \pm 1.1$ | $89.7 \pm 4.2$ | $7.5 \pm 0.6$ |
| | 0.05 | $63.9 \pm 5.5$ | $6.4 \pm 5.1$ | $68.3 \pm 0.4$ | $10.5 \pm 1.3$ | $90.1 \pm 3.8$ | $8.1 \pm 0.6$ |
| **FATE-flip** | 0.10 | $63.6 \pm 5.3$ | $7.9 \pm 6.7$ | $67.8 \pm 0.4$ | $11.2 \pm 1.7$ | $90.3 \pm 3.2$ | $8.5 \pm 0.6$ |
| | 0.15 | $63.7 \pm 5.3$ | $7.5 \pm 6.1$ | $68.2 \pm 0.6$ | $11.2 \pm 1.5$ | $90.2 \pm 2.7$ | $8.8 \pm 0.3$ |
| | 0.20 | $64.1 \pm 5.6$ | $7.7 \pm 6.3$ | $67.8 \pm 0.6$ | $11.1 \pm 0.9$ | $90.0 \pm 2.7$ | $8.7 \pm 0.6$ |
| | 0.25 | $63.6 \pm 5.2$ | $8.5 \pm 7.0$ | $68.0 \pm 0.4$ | $11.5 \pm 1.2$ | $89.9 \pm 3.0$ | $8.8 \pm 0.5$ |
| | 0.00 | $63.8 \pm 5.3$ | $4.0 \pm 3.2$ | $68.2 \pm 0.5$ | $8.6 \pm 1.1$ | $89.7 \pm 4.2$ | $7.5 \pm 0.6$ |
| | 0.05 | $63.9 \pm 5.5$ | $6.4 \pm 5.1$ | $68.3 \pm 0.4$ | $10.5 \pm 1.3$ | $90.2 \pm 3.7$ | $8.1 \pm 0.7$ |
| **FATE-add** | 0.10 | $63.6 \pm 5.3$ | $7.9 \pm 6.7$ | $67.8 \pm 0.4$ | $11.2 \pm 1.7$ | $90.3 \pm 3.2$ | $8.5 \pm 0.6$ |
| | 0.15 | $63.7 \pm 5.3$ | $7.5 \pm 6.1$ | $68.2 \pm 0.6$ | $11.2 \pm 1.5$ | $90.3 \pm 2.6$ | $8.8 \pm 0.3$ |
| | 0.20 | $64.1 \pm 5.6$ | $7.7 \pm 6.3$ | $67.8 \pm 0.6$ | $11.1 \pm 0.9$ | $90.1 \pm 2.6$ | $8.8 \pm 0.5$ |
| | 0.25 | $63.6 \pm 5.2$ | $8.5 \pm 7.0$ | $68.0 \pm 0.4$ | $11.5 \pm 1.2$ | $89.9 \pm 2.9$ | $8.8 \pm 0.5$ |
| **Attacking Individual Fairness** | | | | | | | |
| | | Pokec-n | | Pokec-z | | Bail | |
| **Dataset** | **Ptb.** | **Micro F1 (↑)** | **Bias (↑)** | **Micro F1 (↑)** | **Bias (↑)** | **Micro F1 (↑)** | **Bias (↑)** |
| | 0.00 | $63.8 \pm 5.3$ | $0.4 \pm 0.2$ | $68.2 \pm 0.5$ | $0.5 \pm 0.1$ | $89.7 \pm 4.2$ | $2.5 \pm 1.2$ |
| | 0.05 | $63.6 \pm 5.3$ | $0.5 \pm 0.2$ | $68.2 \pm 0.8$ | $0.6 \pm 0.1$ | $90.0 \pm 4.2$ | $2.7 \pm 1.1$ |
| **FATE-flip** | 0.10 | $63.7 \pm 5.3$ | $0.5 \pm 0.2$ | $67.8 \pm 0.5$ | $0.6 \pm 0.1$ | $90.0 \pm 4.0$ | $2.8 \pm 1.3$ |
| | 0.15 | $63.7 \pm 5.4$ | $0.5 \pm 0.2$ | $68.2 \pm 0.5$ | $0.6 \pm 0.2$ | $90.2 \pm 3.6$ | $2.8 \pm 1.4$ |
| | 0.20 | $63.5 \pm 5.1$ | $0.5 \pm 0.2$ | $68.5 \pm 0.5$ | $0.6 \pm 0.2$ | $90.2 \pm 3.4$ | $2.8 \pm 1.2$ |
| | 0.25 | $63.5 \pm 5.1$ | $0.5 \pm 0.2$ | $68.0 \pm 0.6$ | $0.6 \pm 0.1$ | $90.2 \pm 3.1$ | $2.7 \pm 1.2$ |
| | 0.00 | $63.8 \pm 5.3$ | $0.4 \pm 0.2$ | $68.2 \pm 0.5$ | $0.5 \pm 0.1$ | $89.7 \pm 4.2$ | $2.5 \pm 1.2$ |
| | 0.05 | $63.9 \pm 5.4$ | $0.5 \pm 0.2$ | $68.2 \pm 0.7$ | $0.6 \pm 0.1$ | $90.0 \pm 4.6$ | $2.7 \pm 1.4$ |
| **FATE-add** | 0.10 | $63.8 \pm 5.4$ | $0.5 \pm 0.2$ | $68.2 \pm 0.5$ | $0.6 \pm 0.2$ | $90.1 \pm 4.0$ | $2.8 \pm 1.2$ |
| | 0.15 | $63.8 \pm 5.4$ | $0.5 \pm 0.2$ | $68.3 \pm 0.2$ | $0.6 \pm 0.2$ | $90.1 \pm 3.9$ | $2.8 \pm 1.2$ |
| | 0.20 | $63.7 \pm 5.3$ | $0.5 \pm 0.2$ | $68.4 \pm 0.3$ | $0.6 \pm 0.1$ | $90.3 \pm 3.2$ | $2.8 \pm 1.3$ |
| | 0.25 | $63.7 \pm 5.3$ | $0.5 \pm 0.2$ | $68.4 \pm 0.3$ | $0.6 \pm 0.1$ | $90.2 \pm 3.1$ | $2.8 \pm 1.2$ |

# G   Further Discussions about Fate

**A – Relationship between fairness attacks and the impossibility theorem of fairness.** The impossibility theorems show that some fairness definitions may not be satisfied at the same time.[6] However, this may not always be regarded as fairness attacks. To our best knowledge, the impossibility theorems prove that two fairness definitions (e.g., statistical parity and predictive parity) cannot be fully satisfied at the same time, i.e., biases for two fairness definitions are both zero). However, there is no formal theoretical guarantees that ensuring one fairness definition will *always* amplify the bias of another fairness definition. Such formal guarantees might be nontrivial and beyond the scope of our paper. As we pointed out in the abstract, the main goal of this paper is to provide insights into the adversarial robustness of fair graph learning and can shed light for designing robust and fair graph learning in future studies.

**B – Relationship between FATE and Metattack.** FATE bears subtle differences with Metattack (Zügner & Günnemann, 2019), which utilizes meta learning for adversarial attacks on utility. Note that Metattack aims to degrade the utility of a graph neural network by maximizing the task-specific utility loss (e.g., cross entropy for node classification) in the upper-level optimization problem. Different from Metattack, FATE aims to attack the fairness instead of utility by setting the upper-level optimization problem as maximizing a bias function rather than a task-specific utility loss.

**C – Alternative edge selection strategy via sampling.** Here we introduce an alternative perturbation set selection strategy that is different from the greedy selection described in Section 3.2. The key idea is to view each edge in the graph as a Bernoulli random variable (Lin et al., 2022; Liu et al., 2023a). And the general workflow is as follows. First, we follow Eq. 5 to get a poisoning preference matrix $\nabla_{\mathbf{A}}$. Then, we normalize $\nabla_{\mathbf{A}}$ to a probability matrix $\mathbf{P_A}$. Finally, for the $i$-th attacking step, we can sample $\delta_i$ entries without replacement using $\mathbf{P_A}$ as the set of edges to be manipulated (i.e., added/deleted/flipped).

**D – The potential of FATE on attacking a specific demographic group in group fairness.** To attack a specific group, there can be two possible strategies: (1) decreasing the acceptance rate of the corresponding group and (2) increasing the gap between the group to be attacked and another demographic group. For (1), following our strategy of modeling acceptance rate as the CDF of Gaussian KDE, we can set the bias function to be maximize as the negative of acceptance rate, i.e., $b\left(\mathbf{Y}, \Theta^*, \mathbf{F}\right) = -\mathrm{P}\left[\tilde{y} = 1 \mid s = a\right]$, where $a$ is the sensitive attribute value denoting the demographic group to be attacked. For (2), suppose we want to attack the group with sensitive attribute value. We can also attack this demographic group by setting the bias function to be $b\left(\mathbf{Y}, \Theta^*, \mathbf{F}\right) = \mathrm{P}\left[\tilde{y} = 1 \mid s = 1\right] - \mathrm{P}\left[\tilde{y} = 1 \mid s = 0\right]$. In this way, we can increase the acceptance rate of demographic group ($s = 1$) while minimizing the acceptance rate of the group ($s = 0$).

**E – The potential of FATE on attacking the best/worst accuracy group.** To attack the best/worst accuracy group, the general idea is to set the bias function to be the loss of the best/worst group. It is worth noting that such attack is conceptually similar to adversarial attacks on the utility as shown in Metattack (Zügner & Günnemann, 2019), but only focusing on a subgroup of nodes determined by the sensitive attribute rather than the validation set.

**F – Justification of applying kernel density estimation on non-IID graph data.** To date, it remains an open problem whether the learned node representations follow IID assumption on the low-dimensional manifold or not. Empirically from the experimental results, using KDE-based bias approximation effectively helps maximize the bias for fairness attacks. Meanwhile, relaxing the IID assumption is a common strategy in computing the distributional discrepancy of node representations. For example, MMD is a widely used distributional discrepancy measures, whose accurate approximation also requires IID assumption (Chérief-Abdellatif & Alquier, 2020), and recent studies (Zhu et al., 2021; 2023) show that we can also adapt it on non-IID data which shows promising empirical performance.

**G – Possible defense strategies against deceptive fairness attacks.** FATE demonstrate that it is possible to achieve deceptive fairness attacks on graph learning models by deliberately perturbing the input graph. Given its potential negative societal impacts, we discuss few possible defense strategies against such deceptive fairness attacks. To defend against deceptive fairness attacks for statistical parity, one possible strategy is to preprocess the input graph by either learning a bias-free graph (e.g.,

---

[6]https://machinesgonewrong.com/fairness/

Wang et al. (2022)) or sampling over the neighborhood (e.g., Spinelli et al. (2021); Chen et al. (2022); Lin et al. (2023)) to control which node representations to aggregate during message passing. The reason for such possible design is that Figure 1 reveals the properties of injected edges that are likely to be incident to nodes in the minority class and/or protected group. Following similar principles, it is also possible to develop a selective or probabilistic message passing strategy to achieve the same goal during model optimization. To defend against deceptive fairness attacks for individual fairness, we can apply similar neighborhood sampling strategy or selective/probabilistic message passing strategy. Instead, for individual fairness, the neighborhood sampling or selective message passing would consider the class label rather than the sensitive attribute (i.e., sample edges that connect nodes in the minority class as shown in Figure 2).

**H – How does FATE maintain the performance for deceptiveness?** We assume there is a divergence in optimizing the task-specific loss function $l\left(\mathcal{G}, \mathbf{Y}, \Theta, \theta\right)$ and optimizing the bias function $b\left(\mathbf{Y}, \Theta^{*}, \mathbf{F}\right)$. Thus, maximizing $b\left(\mathbf{Y}, \Theta^{*}, \mathbf{F}\right)$ may not affect $l\left(\mathcal{G}, \mathbf{Y}, \Theta, \theta\right)$ too much. Since we are minimizing the task-specific loss function in the inner loop (i.e., lower-level optimization), it helps to maintain the performance in the downstream task for deceptive fairness attacks. We think such assumption is reasonable for the following reason. In fair machine learning, a common strategy is to solve a regularized optimization problem, where the objective function to be minimized is often defined as $l\left(\mathcal{G}, \mathbf{Y}, \Theta, \theta\right) + \alpha b\left(\mathbf{Y}, \Theta^{*}, \mathbf{F}\right)$ with $\alpha$ being the regularization hyperparameter. If there is no divergence between the optimization of $l\left(\mathcal{G}, \mathbf{Y}, \Theta, \theta\right)$ and $b\left(\mathbf{Y}, \Theta^{*}, \mathbf{F}\right)$, it would be sufficient to optimize one of them to obtain fair and high-utility learning results, or it would be impossible to achieve a good trade-off between fairness and utility if they are completely conflicting with each other. All in all, we believe that optimizing the task-specific loss function $l\left(\mathcal{G}, \mathbf{Y}, \Theta, \theta\right)$ in the lower-level optimization problem could help maintain deceptiveness both intuitively and empirically as shown in Section 6.1, Section 6.2, Appendix D, and Appendix E.

## H  MORE DETAILS ON FAIRNESS DEFINITIONS

We discuss more details about statistical parity and individual fairness here.

**A – Statistical parity.** Mathematically, statistical parity is equivalent to the statistical independence between the learning results (e.g., predicted labels of a classification algorithm) and the sensitive attribute. Consider a classification problem with $\tilde{y}$ being the predicted label, $s$ being the sensitive attribute whose attribute value is in the set $\mathcal{S}$. Statistical parity is defined as follows.

$$\mathrm{P}\left[\tilde{y}=1\right]=\mathrm{P}\left[\tilde{y}=1 | s=a\right], \quad \forall a \in \mathcal{S} \tag{16}$$

**B – Individual fairness.** Other than group fairness, individual fairness studies fairness in a finer-grained individual level. It asks for similar individuals to be treated similarly Dwork et al. (2012). Such principle is often formulated as a Lipschitz inequality

$$d_1\left(\mathbf{Y}\left[i,:\right], \mathbf{Y}\left[j,:\right]\right) \leq \epsilon d_2\left(i, j\right) \tag{17}$$

where $\mathbf{Y}$ is the learning results, $\epsilon$ is the Lipschitz constant, the left hand side $d_1\left(\mathbf{Y}\left[i,:\right], \mathbf{Y}\left[j,:\right]\right)$ measures the distance between the learning results $\mathbf{Y}\left[i,:\right]$ and $\mathbf{Y}\left[j,:\right]$ of data points $i$ and $j$, respectively, and $d_2\left(i, j\right)$ measures the distance between the two data points. Given a graph $\mathcal{G}=\{\mathbf{A}, \mathbf{X}\}$ with adjacency matrix $\mathbf{A}$ and node feature matrix $\mathbf{X}$, Kang et al. Kang et al. (2020) further define $d_1$ as the squared Frobenius distance and assume the existence of an oracle pairwise node similarity matrix $\mathbf{S}$. Then, the overall individual bias of $\mathcal{G}$ is further defined as $\mathrm{Tr}\left(\mathbf{Y}^T \mathbf{L_S} \mathbf{Y}\right)$ where $\mathbf{L_S}$ is the graph Laplacian of similarity matrix $\mathbf{S}$.

