Macro F1 (↑) | Random AUC (↑) | DICE-S Macro F1 (↑) | DICE-S AUC (↑) | FA-GNN Macro F1 (↑) | FA-GNN AUC (↑) | FATE-flip Macro F1 (↑) | FATE-flip AUC (↑) | FATE-add Macro F1 (↑) | FATE-add AUC (↑) |
|---|---|---|---|---|---|---|---|---|---|---|---|
| Pokec-n | 0.00 | 65.3 ± 0.3 | 69.9 ± 0.5 | 65.3 ± 0.3 | 69.9 ± 0.5 | 65.3 ± 0.3 | 69.9 ± 0.5 | 65.3 ± 0.3 | 69.9 ± 0.5 | 65.3 ± 0.3 | 69.9 ± 0.5 |
| | 0.05 | 65.2 ± 0.3 | 70.1 ± 0.2 | 65.7 ± 0.3 | 70.2 ± 0.2 | 65.6 ± 0.6 | **71.1 ± 0.2** | **65.7 ± 0.4** | 70.1 ± 0.6 | 65.5 ± 0.3 | 70.2 ± 0.8 |
| | 0.10 | 65.2 ± 0.3 | 69.6 ± 0.5 | 64.7 ± 0.5 | 69.9 ± 0.2 | 65.4 ± 0.6 | 70.2 ± 0.3 | 65.5 ± 0.3 | 70.2 ± 0.7 | **65.8 ± 0.5** | **70.7 ± 0.6** |
| | 0.15 | 65.4 ± 0.2 | 69.4 ± 0.3 | 64.9 ± 0.2 | 70.1 ± 0.4 | 64.6 ± 0.2 | 69.4 ± 0.1 | **65.6 ± 0.4** | **70.0 ± 0.5** | 65.4 ± 0.1 | 69.8 ± 0.7 |
| | 0.20 | 64.9 ± 0.2 | 69.6 ± 0.3 | 65.1 ± 0.4 | 70.2 ± 0.3 | 63.7 ± 0.5 | 69.0 ± 0.1 | 65.2 ± 0.3 | 69.7 ± 0.6 | **65.6 ± 0.6** | **70.2 ± 0.7** |
| | 0.25 | 64.7 ± 0.1 | 69.4 ± 0.2 | 64.1 ± 0.1 | 69.4 ± 0.2 | 63.3 ± 0.5 | 68.4 ± 0.3 | 65.4 ± 0.6 | 69.7 ± 0.7 | **65.6 ± 0.8** | 69.8 ± 0.8 |
| Pokec-z | 0.00 | 68.2 ± 0.4 | 75.1 ± 0.3 | 68.2 ± 0.4 | 75.1 ± 0.3 | 68.2 ± 0.4 | 75.1 ± 0.3 | 68.2 ± 0.4 | 75.1 ± 0.3 | 68.2 ± 0.4 | 75.1 ± 0.3 |
| | 0.05 | **68.7 ± 0.4** | 75.0 ± 0.4 | **68.7 ± 0.6** | 75.2 ± 0.4 | 68.0 ± 0.4 | 75.1 ± 0.5 | 68.5 ± 0.5 | **75.4 ± 0.2** | 68.5 ± 0.3 | 75.2 ± 0.4 |
| | 0.10 | 68.5 ± 0.1 | 75.1 ± 0.5 | **68.9 ± 0.2** | 75.3 ± 0.1 | 67.9 ± 0.6 | 74.4 ± 0.5 | 68.8 ± 0.5 | 75.5 ± 0.3 | 68.8 ± 0.4 | **75.6 ± 0.2** |
| | 0.15 | 67.5 ± 0.4 | 74.4 ± 0.3 | 67.9 ± 0.3 | 73.8 ± 0.1 | 66.8 ± 0.4 | 72.6 ± 0.2 | 68.4 ± 0.5 | 75.5 ± 0.4 | **68.8 ± 0.7** | **75.6 ± 0.3** |
| | 0.20 | 67.5 ± 0.4 | 74.7 ± 0.4 | 67.7 ± 0.3 | 74.7 ± 0.2 | 66.1 ± 0.1 | 71.8 ± 0.2 | 68.7 ± 0.5 | 75.5 ± 0.3 | **69.0 ± 0.4** | **75.6 ± 0.3** |
| | 0.25 | 67.2 ± 0.3 | 74.1 ± 0.3 | 68.0 ± 0.5 | 74.7 ± 0.2 | 64.8 ± 0.4 | 70.5 ± 0.4 | 68.9 ± 0.3 | 75.6 ± 0.3 | **69.1 ± 0.3** | **75.7 ± 0.3** |
| Bail | 0.00 | 92.3 ± 0.2 | 97.4 ± 0.1 | 92.3 ± 0.2 | 97.4 ± 0.1 | 92.3 ± 0.2 | 97.4 ± 0.1 | 92.3 ± 0.2 | 97.4 ± 0.1 | 92.3 ± 0.2 | 97.4 ± 0.1 |
| | 0.05 | 91.2 ± 0.3 | 94.8 ± 0.2 | 91.4 ± 0.2 | 95.0 ± 0.3 | 90.3 ± 0.2 | 94.1 ± 0.2 | **92.3 ± 0.4** | **97.3 ± 0.1** | 92.1 ± 0.3 | **97.3 ± 0.1** |
| | 0.10 | 90.6 ± 0.1 | 94.2 ± 0.3 | 91.3 ± 0.2 | 94.9 ± 0.4 | 89.1 ± 0.1 | 92.9 ± 0.3 | **92.3 ± 0.1** | **97.3 ± 0.4** | 92.2 ± 0.2 | **97.3 ± 0.1** |
| | 0.15 | 90.3 ± 0.1 | 94.1 ± 0.2 | 91.3 ± 0.3 | 94.7 ± 0.3 | 88.6 ± 0.2 | 92.4 ± 0.3 | **92.4 ± 0.1** | **97.3 ± 0.0** | 92.3 ± 0.2 | **97.3 ± 0.1** |
| | 0.20 | 90.2 ± 0.0 | 93.9 ± 0.1 | 90.9 ± 0.2 | 94.2 ± 0.2 | 87.9 ± 0.2 | 91.8 ± 0.2 | **92.4 ± 0.1** | **97.3 ± 0.0** | **92.4 ± 0.2** | **97.3 ± 0.1** |
| | 0.25 | 90.9 ± 0.1 | 93.5 ± 0.2 | 90.5 ± 0.1 | 94.1 ± 0.4 | 87.6 ± 0.1 | 91.6 ± 0.2 | **92.2 ± 0.2** | 97.2 ± 0.1 | **92.2 ± 0.2** | **97.3 ± 0.1** |

Table 8: Macro F1 score and AUC score of attacking individual fairness on InFoRM-GNN under different perturbation rates (Ptb.). FATE poisons the graph via both edge flipping (FATE-flip) and edge addition (FATE-add) while all other baselines poison the graph via edge addition. Higher is better (↑) for macro F1 score (Macro F1) and AUC score (AUC). Bold font indicates the highest macro F1 score or AUC score.

| Dataset | Ptb. | Random Macro F1 (↑) | Random AUC (↑) | DICE-S Macro F1 (↑) | DICE-S AUC (↑) | FA-GNN Macro F1 (↑) | FA-GNN AUC (↑) | FATE-flip Macro F1 (↑) | FATE-flip AUC (↑) | FATE-add Macro F1 (↑) | FATE-add AUC (↑) |
|---|---|---|---|---|---|---|---|---|---|---|---|
| Pokec-n | 0.00 | 65.4 ± 0.4 | 70.5 ± 0.8 | 65.4 ± 0.4 | 70.5 ± 0.8 | 65.4 ± 0.4 | 70.5 ± 0.8 | 65.4 ± 0.4 | 70.5 ± 0.8 | 65.4 ± 0.4 | 70.5 ± 0.8 |
| | 0.05 | 65.1 ± 0.3 | 69.9 ± 0.2 | 65.7 ± 0.1 | 70.3 ± 0.1 | 65.6 ± 0.3 | **70.8 ± 0.1** | **65.9 ± 0.4** | 70.7 ± 0.8 | 65.8 ± 0.5 | 70.5 ± 0.9 |
| | 0.10 | 64.9 ± 0.2 | 69.6 ± 0.5 | 65.2 ± 0.4 | 70.2 ± 0.2 | 64.8 ± 0.4 | 69.8 ± 0.3 | 65.8 ± 0.5 | 70.3 ± 1.0 | **66.0 ± 0.4** | **70.9 ± 1.1** |
| | 0.15 | 64.8 ± 0.4 | 69.6 ± 0.4 | 65.1 ± 0.3 | 70.0 ± 0.5 | 64.4 ± 0.1 | 69.2 ± 0.3 | 65.7 ± 0.6 | **70.3 ± 0.7** | **65.8 ± 0.4** | **70.3 ± 0.9** |
| | 0.20 | 65.1 ± 0.2 | 69.5 ± 0.3 | 64.9 ± 0.5 | 69.9 ± 0.2 | 63.4 ± 0.4 | 69.0 ± 0.2 | 65.5 ± 0.8 | 70.2 ± 0.9 | **65.6 ± 0.6** | **70.5 ± 0.7** |
| | 0.25 | 64.6 ± 0.3 | 69.6 ± 0.2 | 64.5 ± 0.3 | 69.4 ± 0.2 | 63.6 ± 0.3 | 68.6 ± 0.2 | **66.0 ± 0.5** | **70.8 ± 0.3** | 65.9 ± 0.5 | 70.4 ± 0.8 |
| Pokec-z | 0.00 | 68.3 ± 0.4 | 75.2 ± 0.2 | 68.3 ± 0.4 | 75.2 ± 0.2 | 68.3 ± 0.4 | 75.2 ± 0.2 | 68.3 ± 0.4 | 75.2 ± 0.2 | 68.3 ± 0.4 | 75.2 ± 0.2 |
| | 0.05 | 68.6 ± 0.2 | 75.1 ± 0.3 | **68.9 ± 0.4** | **75.5 ± 0.2** | 67.8 ± 0.6 | 75.0 ± 0.3 | 68.6 ± 0.7 | 75.1 ± 0.6 | 68.7 ± 0.4 | 75.4 ± 0.3 |
| | 0.10 | 67.6 ± 0.2 | 74.3 ± 0.4 | **68.9 ± 0.2** | 75.3 ± 0.3 | 67.7 ± 0.6 | 73.9 ± 0.6 | 68.6 ± 0.6 | **75.6 ± 0.3** | 68.6 ± 0.6 | 75.5 ± 0.3 |
| | 0.15 | 67.2 ± 0.3 | 74.1 ± 0.4 | 67.9 ± 0.4 | 74.4 ± 0.3 | 66.7 ± 0.3 | 72.3 ± 0.1 | 68.9 ± 0.4 | 75.4 ± 0.4 | **69.2 ± 0.6** | **75.4 ± 0.4** |
| | 0.20 | 67.3 ± 0.6 | 74.4 ± 0.4 | 68.3 ± 0.1 | 75.1 ± 0.3 | 66.0 ± 0.5 | 71.7 ± 0.2 | 69.0 ± 0.2 | **75.5 ± 0.2** | **69.2 ± 0.4** | 75.4 ± 0.4 |
| | 0.25 | 66.3 ± 0.4 | 73.9 ± 0.4 | 68.2 ± 0.3 | 74.8 ± 0.1 | 65.0 ± 0.5 | 70.8 ± 0.2 | 68.8 ± 0.7 | 75.6 ± 0.3 | **69.3 ± 0.4** | **75.8 ± 0.1** |
| Bail | 0.00 | 91.9 ± 0.1 | 97.2 ± 0.0 | 91.9 ± 0.1 | 97.2 ± 0.0 | 91.9 ± 0.1 | 97.2 ± 0.0 | 91.9 ± 0.1 | 97.2 ± 0.0 | 91.9 ± 0.1 | 97.2 ± 0.0 |
| | 0.05 | 91.0 ± 0.1 | 94.2 ± 0.2 | 91.2 ± 0.1 | 95.0 ± 0.1 | 90.4 ± 0.1 | 94.2 ± 0.1 | **92.0 ± 0.0** | **97.1 ± 0.1** | 91.9 ± 0.2 | 97.0 ± 0.2 |
| | 0.10 | 90.7 ± 0.2 | 93.9 ± 0.3 | 91.1 ± 0.1 | 94.7 ± 0.3 | 89.4 ± 0.2 | 93.3 ± 0.1 | **92.0 ± 0.1** | **97.0 ± 0.0** | 91.9 ± 0.1 | **97.0 ± 0.0** |
| | 0.15 | 90.5 ± 0.1 | 93.8 ± 0.3 | 91.0 ± 0.2 | 94.5 ± 0.3 | 88.8 ± 0.2 | 92.4 ± 0.1 | **92.0 ± 0.1** | **97.0 ± 0.0** | 91.9 ± 0.1 | **97.0 ± 0.1** |
| | 0.20 | 90.5 ± 0.2 | 93.7 ± 0.2 | 90.8 ± 0.1 | 94.3 ± 0.2 | 87.8 ± 0.1 | 91.8 ± 0.1 | **92.0 ± 0.1** | **96.9 ± 0.0** | 91.9 ± 0.1 | 96.8 ± 0.1 |
| | 0.25 | 90.1 ± 0.2 | 93.4 ± 0.3 | 90.6 ± 0.1 | 94.0 ± 0.1 | 87.4 ± 0.1 | 91.4 ± 0.1 | 91.8 ± 0.1 | 96.8 ± 0.1 | **91.9 ± 0.1** | **96.9 ± 0.0** |