# OpenReview forum: "Deceptive Fairness Attacks on Graphs via Meta Learning"
_ICLR.cc/2024/Conference — ICLR 2024 poster_

### Official Review · Reviewer_5Ww3 · 2023-10-27

**Soundness:** 4 excellent
**Presentation:** 2 fair
**Contribution:** 3 good
**Rating:** 8
**Confidence:** 3

**Summary:**

The paper "Deceptive Fairness Attacks on Graphs via Meta Learning" proposes a meta-learning approach for exacerbating demographic bias in the prediction of a given, unknown, victim model operating on the input graph; It is based on the optimization of a surogate model to define edges (or features to modify). It shows that their approach is able to manipulate edges and features  of the graph so that the performance of different victim models remains constant while the bias is increased.

**Strengths:**

Very interesting approach and results.

Looks theoretically sound.

**Weaknesses:**

The paper sometimes lacks of carity, with some defintions not given (e.g., Y never defined in sec 2) or only in the appendix. In fact I had to go many times in the appendix to understand the core concepts of the approach, which is not very comfortable for the reader. I feel the formalization of the problem/method should include some separated notation for the victim and the surogate model, cause it was a bit confusing. Maybe what we want is that we are good for any victim model from a given distribution of victims ? Setting the problem like this would made the problem easier to understand from my point of view. Results are very hard to follow as some notation are not specified at all (e.g., Ptb which is the perturbation budget but we have to infer it) and every important details as the evaluation metrics are only given in the appendix (I understand the choice for a lack of space but only give some words on it (e.g. simply say delta sp is the bias) and refer to the corresponding section in the appendix would be helpful. Also please better specify the experimental questions you are inspecting in each section of the experimental part, because we can be a little lost at the first read.

**Questions:**

1. I wish the paper could have inspected some possible ways to prevent such attacks. The approach announce that the goal is to better understand. Ok, but what can we do, that would be positive, from the outcomes of the model ? Would it be possible to give some insights for good ways to defend from such attacks ?

2. The considered bias is not fully defined in sec 4, for instance for the instantiation 1: you give the way you estimate both pdf but finally what is the score considered ? an mse, a kl, or what ?  Also, would it be possible to consider a targeted attack that would favor a specific population ?

3. Would it be possible to specify more deeply $\nabla_{\cal G} \Theta^{(T)}$ ? is it equal to $ \epsilon \nabla_{\cal G} \nabla_\Theta l({\cal G}, Y,\Theta,\theta)|_{\Theta^{(T-1)}}$, with $\epsilon$ the step size of update of the surogate ?

4. I have difficulties to well understand the recursive definition of (3) and its connection with (2). Could you clarify please ?

**Details Of Ethics Concerns:**

There can be ethical concerns with this approach which is designed to exacerbate biases on graph models. This is done to better envision defense actions, but it could be misused online. However, authors are aware of this and will only release the code under the CC-BY-NC-ND license upon publication, which prohibits the use of our method for any commercial purposes.

---

> ### Author Response · Authors · 2023-11-20
> **Response to reviewer 5Ww3 (Part 1)**
>
> We appreciate your constructive comments and valuable feedback for further improving our work. Please see our point-to-point responses as follows.
>
> **Q1. Lack of clarity, including definition of $\mathbf{Y}$, separating victim and surrogate model, reference to Ptb. and bias measures, and experimental questions.**
>
> Thank you for your suggestion in further improving the clarity.
>
> - Definition of $\mathbf{Y}$: In the last paragraph of Section 2, we have defined $\mathbf{Y}$ as the output of graph learning model, which is highlighted in blue.
>
> - Separating victim and surrogate model: We would like to clarify that the attacker only have access to the surrogate model, i.e., solving Eq. (1) is solely based on the surrogate model. The victim model is only used when evaluate the performance of FATE in Section 6. Thus, we believe it it not necessary to differentiate the notations for victim model and the surrogate model in Sections 3 -- 5.
>
> - Reference to Ptb. and bias measure: We have revised the table caption to reflect the meaning of Ptb. and $\Delta_{\text{SP}}$ in the revised version, highlighted in blue. For Table 1, the new caption is "Attacking statistical parity on GCN under different perturbation rates (Ptb.). FATE poisons the graph via both edge flipping (FATE-flip) and edge addition (FATE-add) while all other baselines poison the graph via edge addition. Higher is better ($\uparrow$) for micro F1 score (Micro F1) and $\Delta_{\text{SP}}$ (bias). Bold font indicates the most deceptive fairness attack, i.e., increasing $\Delta_{\text{SP}}$ and highest micro F1. Underlined cell indicates the failure of fairness attack, i.e., decreasing $\Delta_{\text{SP}}$ after fairness attack." Similar changes have also been made to Tables 2 and 5 -- 11.
>
> - Add experimental questions: Given the space limitation, we have added the philosophical experimental questions at the beginning of Appendix D, which are (1) How effective is FATE in exacerbating bias under different perturbation rates? (2) How effective is FATE in maintaining node classification accuracy for deceptiveness under different perturbation rates? And (3) can we characterize the properties of edges perturbed by FATE?
>
>
> **Q2. Possible defense strategy.**
>
> Thank you for your suggestion in discussing how to defend against fairness attacks. Following our analysis on the manipulated edges in Sections 6.1 and 6.2, we believe possible defense strategies could be as follows. To defend against deceptive fairness attacks for statistical parity, one possible strategy is to preprocess the input graph by either learning a bias-free graph (e.g., [1]) or sampling over the neighborhood (e.g., [2, 3, 4]) to control which node representations to aggregate during message passing. The reason for such possible design is that Figure 1 reveals the properties of injected edges that are likely to be incident to nodes in the minority class and/or protected group. Following similar principles, it is also possible to develop a selective or probabilistic message passing strategy to achieve the same goal during model optimization. To defend against deceptive fairness attacks for individual fairness, we can apply similar neighborhood sampling strategy or selective/probabilistic message passing strategy. Instead, for individual fairness, the neighborhood sampling or selective message passing would consider the class label rather than the sensitive attribute (i.e., sample edges that connect nodes in the minority class as shown in Figure 2). Please note that the aforementioned discussion has been added in our revised version (Appendix H.G).
>
> **Q3. Bias measure in Section 4.**
>
> Thank you for your suggestion in further improving the clarity. We have added the definition of bias function (i.e., $b\left(\mathbf{Y}, \Theta^*, \mathbf{S}\right) = |\operatorname{P}\left[{\tilde y} = 1\right] = \operatorname{P}\left[{\tilde y} = 1|s = 1\right]|$) in Section 4-C, which is highlighted in blue.

---

> > ### Author Response · Authors · 2023-11-20
> > **Response to reviewer 5Ww3 (Part 2)**
> >
> > **Q4. Attacking specific subpopulation.**
> >
> > Thank you for your suggestion for attacking specific subpopulation. In fact, we have discussed this situation in Appendix H.D (The potential of FATE on attacking a specific demographic group in group fairness) in the original manuscript. More specifically, to attack a specific group, there can be two possible strategies: (1) decreasing the acceptance rate of the corresponding group and (2) increasing the gap between the group to be attacked and another demographic group. For (1), following our strategy of modeling acceptance rate as the CDF of Gaussian KDE, we can set the bias function to be maximize as the negative of acceptance rate, i.e., $b\left(\mathbf{Y}, \Theta^*, \mathbf{F}\right) = - P\left[\widehat{y} = 1 \mid s = a\right]$, where $a$ is the sensitive attribute value denoting the demographic group to be attacked. For (2), suppose we want to attack the group with sensitive attribute value. We can also attack this demographic group by setting the bias function to be $b\left(\mathbf{Y}, \Theta^*, \mathbf{F}\right) = P\left[\widehat{y} = 1 \mid s = 1\right] - P\left[\widehat{y} = 1 \mid s = 0\right]$. In this way, we can increase the acceptance rate of demographic group ($s = 1$) while minimizing the acceptance rate of the group ($s = 0$).
> >
> > **Q5. Clarification on $\nabla_{\mathcal{G}} \Theta^{(T)}$.**
> >
> > Thank you for your question. We would like to clarify that $\nabla\_{\mathcal{G}} \Theta^{(T)}$ is not the same as $\epsilon \nabla\_{\mathcal{G}} \nabla\_{\Theta} l(\mathcal{G}, \mathbf{Y}, \Theta, \theta)|\_{\Theta^{(T-1)}}$. It should be the partial derivative of model parameters at time step $T$ with respect to the input graph, which is derived from the chain rule. The detailed steps are as follows. Our goal is to take the derivative of the bias function $b(\mathbf{Y}, \Theta, \theta)$. Since $\Theta$ is learned by an optimizer $\Theta^{(t+1)} = \operatorname{opt}^{(t+1)}\left(\mathcal{G}, \Theta^{(t)}, \theta, \mathbf{Y}\right),~\forall t\in\{1, \ldots, T\}$, $\Theta$ can be regarded as a function over the input graph $\mathcal{G}$. Thus, when taking the derivative of the bias function with respect to $\mathcal{G}$, we apply the chain rule by calculating the partial derivative of bias function with respect to the model parameter $\Theta$ and then calculating the partial derivative of $\Theta$ with respect to $\mathcal{G}$. This gives us $$\nabla_{\mathcal{G}} b = \nabla_{\mathcal{G}} b\left(\mathbf{Y}, \Theta^{(T)}, \mathbf{F}\right) + \sum_{t=0}^{T-2} \nabla_{\mathcal{G}}\Theta^{(t+1)} \nabla_{\Theta^{(t+1)}}\Theta^{(t+2)} \dots \nabla_{\Theta^{(T-1)}}\Theta^{(T)}\nabla_{\theta^{(T)}} b\left(\mathbf{Y}, \Theta^{(T)}, \mathbf{F}\right)$$ We then apply first-order approximation which has been used in MAML and Metattack and simplifies the equation above to $\nabla_{\mathcal{G}} b \approx \nabla_{\Theta^{(T)}} b\left(\mathbf{Y}, \Theta^{(T)}, \mathbf{F}\right) \cdot \nabla_{\mathcal{G}} \Theta^{(T)}$.
> >
> > **Q6. Connection between Eq. (2) and Eq. (3).**
> >
> > Thank you for your question. Eq. (2) calculates the **partial derivative** of the bias function, while Eq. (3) calculates the **derivative** of the bias function with respect to the adjancency matrix. The reason we need Eq. (3) is that the input graph is undirected causing the adjacency matrix to be symmetric. If Eq. (3) is not applied for symmetric adjacency matrix, the partial derivative with respect to $(i, j)$-th entry in $\mathbf{A}$ could be unequal to the partial derivative $(j, i)$-th entry in $\mathbf{A}$. Then after one-step of continuous attack, the perturbed adjacency matrix will become to an asymmetric matrix, which is inconsistent with the nature of input graph being undirected.
> >
> >
> > **References**
> >
> > [1] Wang, Nan, et al. "Unbiased graph embedding with biased graph observations." Proceedings of the ACM Web Conference 2022. 2022.
> >
> > [2] Spinelli, Indro, et al. "Fairdrop: Biased edge dropout for enhancing fairness in graph representation learning." IEEE Transactions on Artificial Intelligence 3.3 (2021): 344-354.
> >
> > [3] Chen, April, et al. "Graph Learning with Localized Neighborhood Fairness." arXiv preprint arXiv:2212.12040 (2022).
> >
> > [4] Lin, Xiao, et al. "BeMap: Balanced Message Passing for Fair Graph Neural Network." arXiv preprint arXiv:2306.04107 (2023).

---

> > > ### Comment · Reviewer_5Ww3 · 2023-11-21
> > > **thanks**
> > >
> > > Thanks for the insightful answers. However, I wish authors made more to improve clarity, as in many cases they only answered my questions here without impacting the paper. I think these answers would help the reader understand every component of the approach. I am not fully convinced by the answer about their claim of no need of separation of notations for victim and surrogate. I well understood that the victim is only used for evaluation, but I feel the global objective could include it in its formulation. It would be greatly better from my point of view.
> > >
> > > Anyway, I keep my score unchanged.

---

> > > > ### Author Response · Authors · 2023-11-22
> > > > **Further response to reviewer 5Ww3**
> > > >
> > > > We appreciate the reviewer for your suggestion in improving our paper's clarity. Based on your suggestion, we have separated the notation of victim model and surrogate model. Regarding the problem definition in Section 2, our global objective is to exacerbate the bias and maintain the performance of the victim model. Thus, we represent the model parameters and hyperparameters as $\Theta_{\text{vic}}$ and $\theta_{\text{vic}}$. Starting from Section 3, since the attacker only has access to the surrogate model, the attacker's goal becomes to exacerbate the bias and maintain the performance of the surrogate model, whose model parameters and hyperparameters are denoted as $\Theta_{\text{sur}}$ and $\theta_{\text{sur}}$. These changes has been updated in the updated version on OpenReview. We would highly appreciate it, if the reviewer could let us whether any further changes or clarifications are needed.
> > > >
> > > > Besides, we would like to point out that most of our responses to your concerns have been updated in the updated version on OpenReview (highlighted in blue). For example, our response to your Q1 (except for experimental questions) and Q3 has been updated in the main body of updated version (highlighted in blue), and our response to Q4 is presented in Appendix H.D of the original manuscript. Additionally, our responses to your suggestion about experimental questions and Q2 have been updated at the beginning of Appendix D and in Appendix H.G (due to the page limitation).
> > > >
> > > > Again, we sincerely appreciate the reviewer for your suggestion on better clarity. We will keep polishing the paper in the remaining days. If there is any specific contents that the reviewer would like us to move to the main body, we will try our best to move them to the main body for better clarity. Should you have any remaining concerns, we are more than happy to discuss with you.

---

### Official Review · Reviewer_Jcma · 2023-10-30

**Soundness:** 4 excellent
**Presentation:** 3 good
**Contribution:** 3 good
**Rating:** 6
**Confidence:** 5

**Summary:**

This paper investigates deceptive fairness attacks on graph learning, introducing a meta-learning-based framework, FATE. Designed with broad applicability, FATE targets biases in graph neural networks. Using real-world datasets for semi-supervised node classification, the authors demonstrate FATE's ability to amplify biases without compromising downstream task utility. This emphasizes the covert nature of these attacks. The study hopes to spotlight adversarial robustness in fair graph learning, guiding future endeavours towards creating robust and equitable graph models. The paper is a commendable exploration of the crossroads of fairness, adversarial attacks, and graph learning.

**Strengths:**

The topic of this paper is attacking the fairness of GNNs by a poisoning attack setting, which is valuable for building trustworthy GNNs. The method in this paper is easy to follow. In the experimental evaluations, the authors attempt to provide deep analysis upon validating the effectiveness of the proposed method, which is great for the audience to obtain insights and understanding of the vulnerability of GNN fairness.

**Weaknesses:**

1.Limited novelty. The limitation of existing works is weak (section 2-c). For limitation 1, this paper still only attacks one fairness concept when generating the adversarial graph. For limitation 2, the operation space depends on the practicality of the attack. For limitation 3, the specially designed module for maintaining performance is not presented in this paper. Authors are suggested to revise this part to highlight the contribution of this paper.

2.Inaccurate/Unclear statements. For the budget limitation in the “capability of the attacker”, please be aware that the ||.||_{1,1} (according to the common definition of matrix norm) focuses on the maximum number of perturbations among nodes and cannot reflect the number of total perturbations. The definition of d(G,\tilde G) should also include the changes on X. In the 4-c, the \hat{y_(i,1)} discussing the sensitive attributes is repeated with that defining the label in 4-B.

3.The design for keeping the performance of GNNs is not presented in the equation (1). Authors are suggested to discuss why the proposed method can achieve the performance goal, as presented at the beginning of this paper.

4.The operation in equation 6 should be explained with more details to show how to handle different connection statuses and gradient statuses. Moreover, the flip and add operations should also be explained to show how they happen.

5.For definition 1, authors are suggested to explain why the conclusion from IID assumption can be used in the graph data, where nodes are non-IID. Moreover, according to the definition of CDF, the integral starts from -infinity. Authors are suggested to revise related symbols or statements in section 4.

6.In section 5, authors are suggested to explain that L_s is the Laplacian matrix of S, avoiding the confusion of audience who are not familiar with fairness.

7.Limited evaluation. It is worth noting that the surrogate model is SGC while the target model is GCN, and they almost have extremely similar architecture. Authors are suggested to evaluate the effectiveness of the proposed method on other target GNN architectures (e.g., GAT, GraphSAGE) to follow the attack setting of this paper (section 3.1, attackers have only access to the graph data).

8.Evaluation metric. It is worth noting that the fairness of GNNs is at the cost of performance. Although it is great to see the fairness deduction with limited performance cost. However, FA-GNN works well in a lot of cases. Considering that both fairness and model performance is the design of this paper, authors are suggested to compare the (\delta SP)/(\delta Acc) to evaluate the bias increment on unit acc cost.

9.Confusing analysis on the effect of the perturbation rate. I understand that the authors attempt to show insights on the adversarial graph. However, it is hard to digest the operation (delete or add) on the same/different label/attribute. The discussion in Figure 1-b is also confusing. I suppose that detailed explanations of what exactly happened on each bar will make a more clear discussion. Authors are suggested to revise the paragraphs related to figures 1 and 2.

10.According to the effectiveness table results, authors are suggested to explain why the proposed method can improve the performance of GNNs.

11.On the “Effect of the perturbation rate” in section 6.2. Intuitively, more perturbations will make the learned parameters farther from that learned from the original graph. Thus, the availability scope of the learned adversarial graph will be limited by the perturbations, i.e., the higher budget will result in more unexpected behaviours of GNN trained on the poisoned graph. Authors are suggested to revise and discuss it in this paper, which is the limitation of the gradient-based attack method.

**Questions:**

Refer to the weakness part.

---

> ### Author Response · Authors · 2023-11-20
> **Response to reviewer Jcma (Part 1)**
>
> We appreciate your constructive comments and valuable feedback for further improving our work. Please see our point-to-point responses as follows.
>
> **Q1. Limited novelty and discussion on three limitations for existing work.**
>
> Thank you for your questions.
> - For limitation 1, we would like to clarify that we attack **two** fairness definitions, i.e., statistical parity (Sections 4 and 6.1) and individual fairness (Sections 5 and 6.2), to generate the adversarial graphs.
> - For limitation 2, we agree with the reviewer that the operation space would be affected by the practicability of the attack. However, we believe both edge injection, edge deletion, and edge flipping are almost equally practical. For example, in the given malicious banker example, edge injection refers to injecting adversarial transaction record(s) in the transaction network, edge deletion refers to removing transaction record(s), and edge flipping is the combination of two.
> - For limitation 3, we implicitly assume there is a divergence in optimizing the task-specific loss function $l(\mathcal{G},\mathbf{Y},\Theta,\theta)$ and optimizing the bias function $b\left(\mathbf{Y},\Theta^*,\mathbf{F}\right)$. Thus, maximizing $b(\mathbf{Y},\Theta^*,\mathbf{F})$ may not affect $l\left(\mathcal{G},\mathbf{Y},\Theta,\theta\right)$ too much. Since we are minimizing the task-specific loss function in the inner loop (i.e., lower-level optimization), it helps to maintain the performance in the downstream task for deceptive fairness attacks. We think such assumption is reasonable for the following reason. In fair machine learning, a common strategy is to solve a regularized optimization problem, where the objective function to be minimized is often defined as $l(\mathcal{G},\mathbf{Y}, \Theta, \theta)+\alpha b(\mathbf{Y},\Theta^*,\mathbf{F})$ with $\alpha$ being the regularization hyperparameter. If there is no divergence between the optimization of $l(\mathcal{G},\mathbf{Y},\Theta,\theta)$ and $b(\mathbf{Y},\Theta^*,\mathbf{F})$, it would be sufficient to optimize one of them to obtain fair and high-utility learning results, or it would be impossible to achieve a good trade-off between fairness and utility if they are completely conflicting with each other. All in all, we believe that optimizing the task-specific loss function $l(\mathcal{G},\mathbf{Y},\Theta,\theta)$ in the lower-level optimization problem could help maintain deceptiveness both intuitively and empirically as shown in Section 6.1, Section 6.2, Appendix E, and Appendix F.
>
>
> **Q2. Inaccurate/unclear statements about matrix norm, definition of distance between graphs, and repeated discussion about $\hat{y}$.**
>
> Thank you for your suggestions.
> - Regarding the matrix norm: we believe the review is referring to matrix-induced norm. For the induced matrix norms, $\|\mathbf{A}\|_{1,1}$ would refer to the maximum absolute column sum of the matrix difference. However, as we mentioned at the end of Section 2.C, we use the entry-wise matrix norms. It treat the elements of the matrix as one big vector (see [here](https://en.wikipedia.org/wiki/Matrix_norm#%22Entry-wise%22_matrix_norms)). Following the definition of entry-wise matrix norms, our notation means the absolute sum of each entry as shown at the end of Section 2.C.
>
> - Regarding the definition of distance between graphs: we agree with the reviewer that the distance between two graphs might also consider the distance between feature matrices in some cases. However, we believe how to define the distance is based on the attacker's capability in attacking adjacency matrix and/or feature matrix. If the attacker would only attack the adjacency matrix, it is sufficient to define the distance between two graphs as the distance between two adjacency matrices, because the distance between two feature matrices will always be 0. If the attacker would want to attack both adjacency matrix and feature matrix, the distance between two graphs should consider both the distance between adjacency matrices and the distance between feature matrices. In fact, when we describe the distance between two graphs at the end of Section 2.C, we use e.g. (for example) to provide an example. And when we describe the distance in Section 3.1 ("The capability of attacker"), we use i.e. (in another word) to explain the distance constraints for edges/features.
>
> - Regarding the repeated discussion about $\hat{y}$: we would like to clarify that this is not repeated definition. We acknowledge that the previous descriptions might confuse the reviewer. In the revised version, we use ${\widehat y}\_{i,j}$ to denote the prediction probability of node $i$ belonging to class $j$ and ${\tilde y}$ to denote the predicted label. Then, $P[{\tilde y} = 1]$ refers to the complementary CDF of $p({\widehat y}_{i,1} > \frac{1}{2})$ because we would classify node $i$ to class 1 if the corresponding prediction probability is larger than $\frac{1}{2}$ in a binary classification problem.

---

> > ### Author Response · Authors · 2023-11-20
> > **Response to reviewer Jcma (Part 2)**
> >
> > **Q3. Keeping the performance of GNN in Eq. (1).**
> >
> > Thank you for your question. Please refer to our response to the third limitation in your Q1. More specifically, we implicitly assume there is a divergence in optimizing the task-specific loss function $l\left(\mathcal{G},\mathbf{Y}, \Theta, \theta\right)$ and optimizing the bias function $b\left(\mathbf{Y}, \Theta^*, \mathbf{F}\right)$. Thus, maximizing $b\left(\mathbf{Y}, \Theta^*, \mathbf{F}\right)$ may not affect $l\left(\mathcal{G},\mathbf{Y}, \Theta, \theta\right)$ too much. Since we are minimizing the task-specific loss function in the inner loop (i.e., lower-level optimization), it helps to maintain the performance in the downstream task for deceptive fairness attacks. We think such assumption is reasonable for the following reason. In fair machine learning, a common strategy is to solve a regularized optimization problem, where the objective function to be minimized is often defined as $l\left(\mathcal{G},\mathbf{Y}, \Theta, \theta\right) + \alpha b\left(\mathbf{Y}, \Theta^*, \mathbf{F}\right)$ with $\alpha$ being the regularization hyperparameter. If there is no divergence between the optimization of $l\left(\mathcal{G},\mathbf{Y}, \Theta, \theta\right)$ and $b\left(\mathbf{Y}, \Theta^*, \mathbf{F}\right)$, it would be sufficient to optimize one of them to obtain fair and high-utility learning results, or it would be impossible to achieve a good trade-off between fairness and utility if they are completely conflicting with each other. All in all, we believe that optimizing the task-specific loss function $l\left(\mathcal{G},\mathbf{Y}, \Theta, \theta\right)$ in the lower-level optimization problem could help maintain deceptiveness both intuitively and empirically as shown in Section 6.1, Section 6.2, Appendix E, and Appendix F.
> >
> >
> > **Q4. Explain how to handle different connection status and gradient status in Eq. (6).**
> >
> > Thank you for your question. To flip the edges, we first apply Eq. (6) to select edges with highest preference scores and then flip the corresponding value in the adjacency matrix (i.e., 0 is flipped to 1, and 1 is flipped to 0). To add edges without any deletion, all negative entries in $\nabla_\mathbf{A} b$ should be zeroed out before computing the preference score in Eq. (5). Likewise, if edges are only expected to be deleted, all positive entries in $\nabla_\mathbf{A} b$ should be zeroed out before computing the preference score in Eq. (5)
> >
> > The rationale of such choice is as follows. Given that $\nabla_{\mathbf{A}} = \left(\mathbf{1} - 2\mathbf{A}\right) \circ \nabla_\mathbf{A} b$, when $\Delta_{\mathbf{A}}b$ is positive, it indicates that adding an edge will help increase the bias. Since $\left(\mathbf{I}-2\mathbf{A}\right)\left[i,j\right] > 0$ corresponds to adding an edge that doesn't exist before, positive $\Delta_{\mathbf{A}}b$ and positive $\left(\mathbf{I}-2\mathbf{A}\right)\left[i,j\right] > 0$ indicates a strong preference in adding edge $\left(i,j\right)$ to increase the edge. However, if $\left(\mathbf{I}-2\mathbf{A}\right)\left[i,j\right] < 0$ in this case, it suggests remove edge $\left(i,j\right)$ instead of adding it, thus decreasing the bias due to the positive gradient value and conflicting with the goal of fairness attacks. So, it should not have a high preference to be selected. Similarly, if we have negative $\Delta_{\mathbf{A}}b$ and negative $\left(\mathbf{I}-2\mathbf{A}\right)\left[i,j\right]$, then removing edge $\left(i,j\right)$ would help increase the edge. However, a positive $\left(\mathbf{I}-2\mathbf{A}\right)\left[i,j\right]$ suggests to add the edge instead of removing it. It is also conflicting with the goal of fairness attacks, and our method would not select this edge.
> >
> >
> > **Q5. Non-IID assumption in Definition 1.**
> >
> > Thank you for your feedback about non-IID assumption for applying kernel density estimation. In our original manuscript, we have provided the justifications in Appendix H.F. Specifically, we believe it remains an open problem whether the learned node representations follow IID assumption on the low-dimensional manifold or not. Empirically from the experimental results, using KDE-based bias approximation effectively helps maximize the bias for fairness attacks. Meanwhile, relaxing the IID assumption is a common strategy in computing the distributional discrepancy of node representations. For example, MMD is a widely used distributional discrepancy measures, whose accurate approximation also requires IID assumption [1], and recent studies [2, 3] show that we can also adapt it on non-IID data which shows promising empirical performance.
> >
> > **Q6. Definition of CDF.**
> >
> > Thank you for pointing it out. The reviewer is correct that CDF corresponds to the integral that starts from negative infinity. We have corrected the corresponding to complementary CDF, which starts from $\tau$ to positive infinity in the revised version (highlighted in blue in Section 4.C).

---

> > > ### Author Response · Authors · 2023-11-20
> > > **Response to reviewer Jcma (Part 3)**
> > >
> > > **Q7. Explain the Laplacian matrix $\mathbf{L}_{\mathbf{S}}$.**
> > >
> > > Thank you for your suggestion in further improving the clarity. We have added ", where $\mathbf{L}_{\mathbf{S}}$ is the Laplacian matrix of $\mathbf{S}$" in Section 5, highlighted in blue.
> > >
> > >
> > > **Q8. Limited evaluation with SGC being the surrogate model and GCN being the victim model.**
> > >
> > > Thank you for your comments. We agree that SGC (the surrogate model) and GCN (the victim model in the main body) are both convolutional aggregation-based graph neural networks. Thus, in Appendix H of the original manuscript, we evaluate the tranferability of fairness attacks of our method by setting the victim model to graph attention network (GAT). From the results, we observe that our method can still consistently succeed in effective fairness attack by increasing $\Delta_{\text{SP}}$ and the individual bias (Bias) and deceptive attack by offering comparable or even better micro F1 score (Micro F1) when the victim model is not a convolutional aggregation-based model. Thus, it shows that the adversarial edges flipped/added by our method is able to transfer to graph neural networks with different type of aggregation function.
> > >
> > > **Q9. Performance of FA-GNN and unit accuracy cost.**
> > >
> > > Thank you for your suggestion in calculating the unit accuracy cost. However, we would highly appreciate the reviewer to clarify how $\Delta_{\text{Acc}}$ is defined because $\Delta_{\text{Acc}}$ as the denominator might cause different conclusions if it is defined in different ways. And we would be more than happy to provide additional results on the unit accuracy cost as soon as possible once the reviewer clarifies it.
> > >
> > >
> > > **Q10. Explanation on the effect of perturbation rate and revise the paragraphs related to Figures 1 and 2.**
> > >
> > > Thank you for your question. We would like to clarify that Figures 1 and 2 refers to our analysis on the manipulated edges, which aims to investigate whether the manipulated edges have certain properties, rather than the effect of perturbation rate. As we mentioned in "Analysis on the manipulated edges", we only analyze FATE-flip because, in practice, we found that the majority of edges manipulated by FATE-flip on all three datasets is by addition. And we choose to analyze the case with perturbation rate 25\% because the cases with smaller perturbation rates is essentially manipulating subsets of edges in the case with perturbation rate 25\%.
> > >
> > >
> > > **Q11. More explanation on why fairness attacks improve GCN performance.**
> > >
> > > Thank you for your question. We think one possible reason is that FATE would favor injecting edges that connect two nodes with the same class labels (as shown in Figures 1 and 2), which would increase the homophily of the perturbed graph. Given that (1) the surrogate model (SGC) and all victim models (GCN, FairGNN, InFoRM-GNN, GAT) are proved to work well due to the homophily assumption and (2) all datasets are homophilous, such homophily-favored manipulations might improve the GCN performance.
> > >
> > >
> > > **Q12. Higher budget will result in more unexpected behaviors of GNN trained on poisoned graph.**
> > >
> > > Thank you for your question. We agree with the reviewer that higher perturbation budget might cause more unexpected behaviors of GNN trained on the poisoned graph. However, please note that, the individual bias in Section 6.2 is computed based on an oracle pairwise node similarity matrix, which is different from the input graph and not affected by the perturbation on the input graph. By attacking individual fairness, the learning results (i.e., prediction probabilities) might be shifted to increase the individual bias to some extent. However, all compared datasets are of high homophily, which might cause the similarity between connected nodes to be higher in the oracle similarity matrix $\mathbf{S}$. Meanwhile, FATE tries to maintain the performance on such homophilious graphs. These combined effects might cause a limited shift/deviation in prediction probabilities, otherwise the performance in the downstream tasks might be dropped which goes against the lower-level optimization problem that maintains the utility.
> > >
> > > Additionally, we would also highly appreciate the reviewer for clarification on the term "availability scope", and we would be happy to discuss with the reviewer further.
> > >
> > >
> > > **References**
> > >
> > > [1] Chérief-Abdellatif, Badr-Eddine, and Pierre Alquier. "MMD-Bayes: Robust Bayesian estimation via maximum mean discrepancy." Symposium on Advances in Approximate Bayesian Inference. PMLR, 2020.
> > >
> > > [2] Zhu, Qi, et al. "Shift-robust gnns: Overcoming the limitations of localized graph training data." Advances in Neural Information Processing Systems 34 (2021): 27965-27977.
> > >
> > > [3] Zhu, Qi, et al. "Explaining and Adapting Graph Conditional Shift." arXiv preprint arXiv:2306.03256 (2023).

---

> > > > ### Comment · Reviewer_Jcma · 2023-11-21
> > > >
> > > > Dear Authors,
> > > >
> > > > For limitation 1, I would recommend revising it as the generalisation of the proposed method, e.g., it can be adapted to attack different fairness. The current statement “attack statistical parity and individual fairness” is suggested to be revised as “attack statistical parity or individual fairness” to avoid confusion.
> > > >
> > > > For the \Delta Acc, a potential available manner of definition could be the |Acc_{before}-Acc_{after}| to evaluate the accuracy changes.
> > > >
> > > > Thanks for your efforts in clarifying your paper. Your answers make sense to me and I have raised the score.

---

> ### Author Response · Authors · 2023-11-22
> **Further response to reviewer Jcma (Part 1)**
>
> We appreciate the reviewer for your suggestion for better clarity and for further clarification on defining the unit accuracy cost. We have followed your suggestion and have changed "attack statistical parity and individual fairness" to "attack statistical parity or individual fairness" in the updated version (in abstract, organization of Appendix at the beginning of Appendix page, and Appendix G).
>
> Regarding the unit accuracy cost, as the reviewer suggested, we define the **absolute** unit accuracy cost as $\text{unit-acc-cost} = \frac{|\Delta_{\text{SP}}^{(\text{before})} - \Delta_{\text{SP}}^{(\text{after})}|}{|\text{Acc}^{(\text{before})} - \text{Acc}^{(\text{after})}|}$, which helps measure the absolute change in $\Delta_{\text{SP}}$ with one unit absolute change in accuracy. And higher is better for absolute unit accuracy, because a higher absolute unit accuracy cost means that one unit change in accuracy could cause more bias increase. The unit accuracy costs of FA-GNN, FATE-flip, and FATE-add are as follows. Please note that, in each cell of table below, the absolute unit accuracy costs are reported when the perturbation are from 0.05 to 0.25 with a step size of 0.05 in an ascending order, i.e., the leftmost value indicates the absolute unit accuracy cost when the perturbation rate is 0.05, and the rightmost value indicates the absolute unit accuracy cost when the perturbation rate is 0.25. (fail) means that the absolute unit accuracy cost refers to the case of failed fairness attack (i.e., decreasing bias after attack).
>
> ||FA-GNN|FATE-flip|FATE-add|
> |-|-|-|-|
> |Pokec-n|12.666 (fail) -> 2.933 -> 5.667 -> 6.647 -> 18.000|5.500 -> 3.857 -> 8.800 -> 7.000 -> 6.250|5.500 -> 3.857 -> 8.800 -> 7.000 -> 6.250|
> |Pokec-z|14.667 (fail) -> 9.857 -> 5.722 -> 8.174 -> 5.414|0.333 -> 3.000 -> 3.167 -> 22.000 -> 20.000|0.333 -> 3.000 -> 3.167 -> 22.000 -> 20.000|
> |Bail|1.429 -> 0.885 -> 0.129 -> 0.176 (fail) -> 0.848 (fail)|1.200 -> 1.286 -> 1.222 -> 1.444 -> 1.100| 1.000 -> 0.857 -> 1.375 -> 1.625 -> 1.100|
>
> For Pokec-n, FATE-flip and FATE-add achieves higher absolute unit accuracy cost when the perturbation rate is smaller than 0.25 (please note that FA-GNN fails the fairness attack when the perturbation rate is 0.05). For Pokec-z, FATE-flip and FATE-add achieves a much higher absolute unit accuracy cost when the perturbation rates are 0.20 and 0.25. While for Bail, FATE-flip and FATE-add consistently offers higher absolute unit accuracy cost (except for a marginal decrease for FATE-add with the perturbation rate being 0.10).

---

> ### Author Response · Authors · 2023-11-22
> **Further response to reviewer Jcma (Part 2)**
>
> In addition, we would like to point out that an absolute unit accuracy cost may be unable to capture whether the accuracy is increased or decreased after attack. Thus, we further provide additional results on unit accuracy bias increase, which is defined as $\text{unit-acc-bias-increase} = \frac{\Delta_{\text{SP}}^{(\text{after})} - \Delta_{\text{SP}}^{(\text{before})}}{\text{Acc}^{(\text{before})} - \text{Acc}^{(\text{after})}}$. The reason for the numerator being $\Delta_{\text{SP}}^{(\text{after})} - \Delta_{\text{SP}}^{(\text{before})}$ is that we want the bias to increase as much as possible after fairness attack, thus measuring the bias increase after attack. While the reason for the denominator being $\text{Acc}^{(\text{before})} - \text{Acc}^{(\text{after})}$ is that we want the accuracy to be decrease as few as possible, thus measuring the accuracy decrease after attack. With this definition, it is clear that (1) a negative unit accuracy bias increase is always better than positive value, because we could increase the bias without costing accuracy; (2) higher is better if unit accuracy bias increase is positive (we increase more bias with a unit decrease in accuracy), and (3) smaller is better if unit accuracy bias increase is negative (i.e., we increase more bias with a unit increase in accuracy). The results for unit accuracy bias increase of FA-GNN, FATE-flip, and FATE-add are as follows (with the same format as the table above). It should be noted that, when the bias decreases and the accuracy increases (i.e., failed fairness attacks), it is also possible to achieve a negative unit accuracy bias increase, which will be denoted by (fail).
>
> ||FA-GNN|FATE-flip|FATE-add|
> |-|-|-|-|
> |Pokec-n|12.666 (fail) -> 2.933 -> 5.667 -> 6.647 -> 18.000|-5.500 -> -3.857 -> -8.800 -> -7.000 -> -6.250|-5.500 -> -3.857 -> -8.800 -> -7.000 -> -6.250|
> |Pokec-z|-14.667 (fail) -> 9.857 -> 5.722 -> 8.174 -> 5.414|-0.333 -> -3.000 -> 3.167 -> -22.000 -> -20.000|-0.333 -> -3.000 -> 3.167 -> -22.000 -> -20.000|
> |Bail|1.429 -> 0.885 -> 0.129 -> -0.176 (fail) -> -0.848 (fail)|1.200 -> 1.286 -> 1.222 -> 1.444 -> 1.100| 1.000 -> 0.857 -> 1.375 -> 1.625 -> 1.100|
>
> From the table above, for Pokec-n, FATE-flip and FATE-add always achieves negative unit accuracy bias increase, which is better than positive values achieved by FA-GNN. For Pokec-z, other than the case when perturbation rate is 0.15, FATE-flip and FATE-add always achieves negative unit accuracy bias increase, which is better than positive values achieved by FA-GNN. For Bail, FATE-flip and FATE-add always achieve a better unit accuracy bias increase than FATE.
>
> We hope this could demonstrate the effectiveness of our proposed method, as we can achieve better absolute unit accuracy cost or unit accuracy bias increase in many cases. In the remaining days, we will calculate both metrics for all experimental settings (statistical parity vs. individual fairness, GCN vs. FairGNN/InFoRM-GNN) and update the manuscript with the additional results.
>
> Again, we sincerely appreciate the reviewer for your kind suggestion on better clarity and new evaluation metric. Should you have any remaining concerns, we are more than happy to discuss with you.

---

### Official Review · Reviewer_RLh1 · 2023-11-01

**Soundness:** 3 good
**Presentation:** 4 excellent
**Contribution:** 4 excellent
**Rating:** 6
**Confidence:** 3

**Summary:**

In the paper Deceptive Fairness Attacks on Graphs via Meta Learning, the authors delve into the realm of deceptive fairness attacks on graph learning models, aiming to amplify bias while simultaneously maintaining or improving utility in downstream tasks. They formalize this challenge as a bi-level optimization problem, where the upper level seeks to maximize the bias function according to a user-defined fairness definition, and the lower level aims to minimize a task-specific loss function.

To address this challenge, the authors introduce FATE, a meta-learning based framework designed to poison the input graph using the meta-gradient of the bias function with respect to the input graph. FATE is rigorously evaluated and compared with three baseline methods: Random, DICE-S, and FA-GNN, across three real-world datasets in the context of binary semi-supervised node classification with binary sensitive attributes.

FATE distinguishes itself through two strategies: edge flipping (FATE-flip) and edge addition (FATE-add), as opposed to other baseline methods which only perform edge addition, and consistently outperform the baseline methods in fairness attacks. Unlike some baseline methods that sometimes fail to attack individual fairness, FATE maintains its effectiveness, amplifying bias while achieving the highest micro F1 scores in node classification. This dual success highlights FATE's deceptiveness, as it can increase bias while simultaneously enhancing or maintaining model performance.

**Strengths:**

1. Clear Structure and Argumentation: The paper is well-organized, providing a clear and compelling argument for the importance of studying deceptive fairness attacks in graph learning models.
2. Robust Experimental Design: The experimental design is of high quality, with all mathematical symbols and notations meticulously explained.
3. Transparent Results Presentation: Results are presented clearly and concisely, with visual aids and detailed explanations that make the findings accessible and easy to understand, showcasing the effectiveness of the FATE framework.
4. Problem Novelty: The proposed work focuses on adversarial robustness of fair graph learning, which needs more exploration.

**Weaknesses:**

It is noted the below are minor weaknesses:
1. Grammatical Typos: There are minor grammatical errors present, such as in Section 6.1 on page 6, where the phrase "It randomly deleting edges..." should be corrected to "It randomly deletes edges...".
2. Incomplete Introduction of Fairness Definitions: In Section 2B, when algorithmic fairness definitions are introduced, the paper provides English definitions but omits corresponding mathematical definitions. Including these mathematical definitions would enhance clarity, especially as the individual fairness definition is important when analyzing results.
3. Inconsistent Organization: The paper utilizes a letter-based organization in some sections and a numerical organization in others. For instance, in Section 4, ideas are laid out numerically but then referred to using letters (A, B,...).

**Questions:**

1. Can more be discussed on the potential other bias functions besides statistical parity?
2. Do the authors believe this attack scenario is reasonable in real-world settings? Although an example is provided of a banker and has been used in prior work, but it feels somehow contrived and would look forward to perhaps more realistic settings where the attackers have more restricted access (e.g., visibility of the existing graph and only able to construct and connect new nodes, etc.).

---

> ### Author Response · Authors · 2023-11-20
> **Response to reviewer RLh1 (Part 1)**
>
> We appreciate your constructive comments and valuable feedback for further improving our work. Please see our point-to-point responses as follows.
>
> **Q1. Grammatical typos.**
>
> Thank you for catching the grammatical errors. We have proofread the paper and fixed them in the revised version (highlighted in blue).
>
>
> **Q2. Missing mathematical formulation of fairness definitions.**
>
> Thank you for your kind suggestion. Given the space limitation, we have added a new section in Appendix (i.e., Appendix B) for mathematical formulations and formal definitions of statistical parity and individual fairness. A pointer to Appendix B is also added in Section 2-B.
>
>
> **Q3. Inconsistent organization between letter-based indexing and numerical indexing.**
>
> Thank you for your suggestion about consistent indexing. We have changed the indexing in Section 4 to letter-based indexing. Meanwhile, for the second-level indexing, we are using numerical indexing to avoid confusion. For example, numerical indexing ((1), (2), ...) are used in Section 2-C. Please let us know if you would like use to change the second-level indexing to other format and we would be more than happy to accommodate it.
>
>
> **Q4. More discussion about other bias function other than statistical parity.**
>
> Thank you for your suggestion. We would like to clarify that, in addition to statistical parity, we instantiate our method by attacking individual fairness in Section 5 and discuss strategies on attacking a specific demographic group in statistical parity and the best/worst accuracy group in Appendix H.D and H.E in the original manuscript. More specifically, for attacking individual fairness, we define the bias function as $b\left(\mathbf{Y}, \Theta^*, \mathbf{S}\right) = \operatorname{Tr}\left(\mathbf{Y}^T \mathbf{L}_{\mathbf{S}} \mathbf{Y}\right)$. For attacking a specific demographic group in statical parity (Appendix H.D), we can set the bias function to either $b\left(\mathbf{Y}, \Theta^*, \mathbf{F}\right) = - \operatorname{P}\left[{\tilde y} = 1 \mid s = a\right]$ to decrease the acceptance rate of demographic group with sensitive attribute value $s=a$ or $b\left(\mathbf{Y}, \Theta^*, \mathbf{F}\right) = \operatorname{P}\left[{\tilde y} = 1 \mid s = 1\right] - \operatorname{P}\left[{\tilde y} = 1 \mid s = 0\right]$ to increase the gap of acceptance rate between group $s = 1$ and group $s = 0$. For attacking the best/worst accuracy group (Appendix H.E), we can set the bias function to be the task-specific loss (e.g., cross-entropy) of the demographic of nodes with best/worst accuracy, which is conceptually similar to adversarial attacks on the utility as shown in Metattack [1], but focusing on a subgroup of nodes determined by the sensitive attribute rather than the validation set. We would highly appreciate if the reviewer could clarify the specific bias function that needs to be discussed, and we would be more than happy to provide additional discussion if possible.

---

> > ### Author Response · Authors · 2023-11-20
> > **Response to reviewer RLh1 (Part 2)**
> >
> > **Q5. How reasonable and realistic is the attack scenario? Possibility for a more restricted setting, e.g., limited visibility, node injection.**
> >
> > Thank you for your question about the attack scenario. Please note that we follow the same assumption compared with existing work FA-GNN [2], in which it requires the sensitive attributes of all nodes to determine which edge to be injected. We leave for future work on fairness attacks with limited visibility of the input graph and with adversarial node injection.
> >
> > - For limited visibility, in practice, we find that, for both debiasing and attacking, statistical parity relies on accurate estimation of the learning results' distributions. This is because statistical parity requires the statistical independence between the learning results and the sensitive attribute. With limited visibility of the graph, it is likely that the distribution of invisible subgraph is different from that of the visible subgraph, which would result in unstable model performance. In terms of fairness on graphs, how to achieve (and attack) statistical parity under potential distribution shift is still a less explored open problem.
> >
> > - For node injection, as we mentioned in the second paragraph in introduction (Section 1), adversarial node injection not only requires generating the node attributes for the injected nodes, but also requires connecting the injected adversarial node(s) to nodes in the original graph. Though existing works [3, 4, 5, 6] have investigated how to generated adversarial data points on tabular data (which might be used to generate features of the adversarially injected nodes on graphs), it still remains non-trivial to connect the injected nodes to the existing nodes to effectively attack the fairness of graph learning models.
> >
> > **References**
> >
> > [1] Zügner, Daniel and Stephan Günnemann. “Adversarial Attacks on Graph Neural Networks via Meta Learning.” ArXiv abs/1902.08412 (2019)
> >
> > [2] Hussain, Hussain, et al. "Adversarial Inter-Group Link Injection Degrades the Fairness of Graph Neural Networks." 2022 IEEE International Conference on Data Mining (ICDM). IEEE, 2022.
> >
> > [3] Solans, David, Battista Biggio, and Carlos Castillo. "Poisoning attacks on algorithmic fairness." Joint European Conference on Machine Learning and Knowledge Discovery in Databases. Cham: Springer International Publishing, 2020.
> >
> > [4] Mehrabi, Ninareh, et al. "Exacerbating algorithmic bias through fairness attacks." Proceedings of the AAAI Conference on Artificial Intelligence. Vol. 35. No. 10. 2021.
> >
> > [5] Chhabra, Anshuman, Adish Singla, and Prasant Mohapatra. "Fairness degrading adversarial attacks against clustering algorithms." arXiv preprint arXiv:2110.12020 (2021).
> >
> > [6] Van, Minh-Hao, et al. "Poisoning attacks on fair machine learning." International Conference on Database Systems for Advanced Applications. Cham: Springer International Publishing, 2022

---

> > ### Comment · Reviewer_RLh1 · 2023-11-23
> >
> > Thank you for your replies. For Q4, this was referring to other group fairness definitions besides statistical parity.
> > I am inclined to keep my score based on your feedback here and upon seeing the other reviews/responses.

---

> > > ### Author Response · Authors · 2023-11-23
> > > **Further response to reviewer RLh1**
> > >
> > > Thank you for your clarification on the bias function for other group fairness definition. We discuss equal opportunity and equalized odds here.
> > >
> > > - For equal opportunity, it aims to satisfy equal true positive rate, i.e., $P\left[\widehat{y} = 1 \mid s = 1, y = 1\right] = P\left[\widehat{y} = 1 \mid s = 0, y = 1\right]$, where $y$ is the ground-truth label. It is similar to statistical parity but conditioned on on the subpopulation of data with ground-truth label $y = 1$. To define the bias function, similar to statistical parity, we can define it as $b\left(\mathbf{Y}, \Theta^*, \mathbf{F}\right) = |P\left[\widehat{y} = 1 \mid s = 1, y = 1\right] - P\left[\widehat{y} = 1 \mid s = 0, y = 1\right]|$. To calculate the bias in a differentiable way, we can follow similar step as described in Section 4 with minor modification. The only modifcation needed is to only use nodes with known ground-truth label $y = 1$ to estimate the complementary CDF with Q-function, instead of using all nodes as shown in Section 4.
> > >
> > > - For equalized odds, it generalizes equal opportunity to all class labels, i.e., $P\left[\widehat{y} = 1 \mid s = 1, y\right] = P\left[\widehat{y} = 1 \mid s = 0, y\right],~\forall y$. The bias function can then be defined as $b\left(\mathbf{Y}, \Theta^*, \mathbf{F}\right) = \sum_y |P\left[\widehat{y} = 1 \mid s = 1, y\right] - P\left[\widehat{y} = 1 \mid s = 0, y\right]|$. Then we can loop over each class label and follow the steps to estimate bias for equal opportunity (as described above) w.r.t. the corresponding class label, and then sum up the estimated bias over all class labels.
> > >
> > > We will incorporate the above discussion in Appendix H of the revised version. If the reviewer would like more discussion on some other bias function for group fairness, please kindly let us know and we will be more than happy to provide additional discussion if necessary.

---

### Official Review · Reviewer_CLho · 2023-11-01

**Soundness:** 3 good
**Presentation:** 3 good
**Contribution:** 3 good
**Rating:** 6
**Confidence:** 3

**Summary:**

The paper first defines deceptive fairness attacks on graphs, which is a new type of threat model that is compatible with many of the existing settings. Then the paper proposes FATE, a framework that models this problem via a bi-level optimization treatment, and offers a solution using meta-learning. Finally, the paper empirically confirmed the effectiveness of their framework on real-world datasets in the task of semi-supervised node classification.

**Strengths:**

1. Very clear motivation and problem formulation
2. Extensive experimental results and clear explanations.
3. Theoretical analysis is necessary and appears to be correct.

**Weaknesses:**

The terminology of "deceptiveness" may be confusing and is not clearly explained, from an intuitive level, in the manuscript.

**Questions:**

What's the intuition behind the method being "meta-learning"? It appears that this is not a typical treatment of meta-learning in graph representation learning.

---

> ### Author Response · Authors · 2023-11-20
> **Response to reviewer CLho**
>
> We appreciate your constructive comments and valuable feedback for further improving our work. Please see our point-to-point responses as follows.
>
> **Q1. Explain the term `deceptiveness' from an intuitive level.**
>
> Thank you for your question. In our revised version (Section 1 paragraph 3), we highlight that "the lower-level problem optimizes a task-specific loss function to maintain the performance of the downstream learning task and enforces budgeted perturbations to make the fairness attacks deceptive". Budgeted perturbations (i.e., imperceptible changes) have been widely adopted in existing works on adversarial attacks for graph neural networks. In addition, we believe a deceptive fairness attack is also important in maintaining the performance of downstream learning task, which is mentioned in Section 2-C in the original manuscript "a performance degradation in the utility would make the fairness attacks not deceptive from the perspective of a utility-maximizing institution." The reason is that it is easy for a utility-maximizing institution to detect the abnormal performance of the graph learning model, which is trained on the poisoned data. Such abnormal performance may trigger the auditing toward model behaviors, which may further lead to the investigation on the data used to trained the model. With careful auditing, it is possible to identify fairness attacks as shown in our analysis of manipulated edges due to significantly more edges that have certain properties (e.g., connecting two nodes with a specific sensitive attribute value). Then the institution might take actions (e.g., data cleansing) to avoid the negative consequences caused by fairness attacks.
>
> **Q2. Intuition behind the method being `meta learning'.**
>
> Thank you for your question. In our work, as we mentioned in the manuscript, we treat the input graph as a hyperparameter to be learned other than model parameters and apply meta learning to find the suitable hyperparameter (i.e., the input graph) that maximizes the bias function. Similar strategy has been adopted in hyperparameter optimization [1] and in adversarial attacks on the utility of graph neural networks [2], as well as in graph structure learning [3, 4]
>
> **References**
>
> [1] Franceschi, Luca, et al. "Bilevel programming for hyperparameter optimization and meta-learning." International conference on machine learning. PMLR, 2018.
>
> [2] Zügner, Daniel and Stephan Günnemann. “Adversarial Attacks on Graph Neural Networks via Meta Learning.” ArXiv abs/1902.08412 (2019)
>
> [3] Franceschi, Luca, et al. "Learning discrete structures for graph neural networks." International conference on machine learning. PMLR, 2019.
>
> [4] Xu, Zhe, Boxin Du, and Hanghang Tong. "Graph sanitation with application to node classification." Proceedings of the ACM Web Conference 2022. 2022.

---

> > ### Comment · Reviewer_CLho · 2023-11-21
> >
> > Thank you for your response. I've maintained my score.

---

> > > ### Author Response · Authors · 2023-11-23
> > > **Thank you!**
> > >
> > > Thank you for your comment! We appreciate your comments to further improve our paper's quality.

---

### Meta-Review · Program_Chairs · 2023-12-04

**Metareview:**

Program chair note: The paper received unanimously positive reviews. The AC raised valid concerns. The program chairs deliberated extensively over this paper and found the merits to outweigh the AC's concerns in this case. The authors are advised to take into account the points raised and clarify them accordingly in their paper.

I have read all the materials of this paper including the manuscript, appendix, comments, and response. Based on collected information from all reviewers and my personal judgment, I can make the recommendation on this paper, *reject*.

**Research Problem**

This paper addresses the fairness attach on graphs.

**Motivation**

The authors pointed out two drawbacks of existing methods in the fairness attack category. (1) Adversarial data injection methods are often designed for tabular data. This is not true. The authors provide a paragraph on adversarial attacks on graphs in the related work. Therefore, the authors need to illustrate the drawbacks of literation on adversarial attacks on graphs, especially what the research challenges are to extend the current methods for tackling fairness attack. (2) The current method only attacks group fairness. The authors believe it is crucial to study the attack of different notations of fairness for a variety of graph learning models. More comprehensiveness does not mean better. If a paper considers the group fairness, which is its research question, it is improper to criticize it cannot handle individual fairness. Just like this paper that can only tackle the differentiable bias functions, but I do not regard this point as a drawback. If some simple extensions can achieve comprehensiveness, it does not count for novelty or contributions. In this paper, the authors also consider the individual fairness. Since there is a huge body of literature on individual fairness on graphs, it also needs to discuss the difficulty of extending these works in the attack scenario.

I did notice the third point in problem definition, which considers attacking fairness and preserve the algorithmic utility. This is good; unfortunately, the authors did not have any action or design in their framework.

**Challenge Analysis**

Beyond motivations, I did not find specific challenges in this paper.

**Philosophy**

For the above first point, the challenge is not specific, and I did not find any philosophy to tackle this. For the second point, the authors employed the bi-level optimization for generalization on fairness notations and graph models, where the third point is also embedded as a constraint.

**Related Work**

Since the authors also address the individual fairness, it would be to provide a paragraph on that.

**Technique**

Respectfully, I did not see too much innovation in the technical part. Moreover, the lower-level optimization is problematic, which minimizes the loss function towards utility. As the authors mentioned, performance degradation in utility makes the attack deceptive. However, utility increase will also raise the deceptiveness. One reviewer has a similar concern, while the authors responded that minimizing the utility loss is enough to maintain the performance. Clearly, the experiments in Table 1 demonstrate it is not true. Note that some vanilla graph learning models without taking fairness into consideration can achieve higher utility but low fairness, compared with fair graph models.

**Experiments**

1. Since there are many graph attack methods, why the authors only compare with DICE? Please check the methods in the related work part.
2. It is suggested to provide the significant analysis.
3. In Table 1, the definition of bold font should be the one with remained utility but decreased fairness. Actually they are two dimensions. It is suggested to just use bold to highlight one dimension for better understanding; otherwise, an equation is needed to demonstrate how to get the best result in two dimensions for bolding.
4. I noticed some utility gain of FATE in Table 1, which makes the model not deceptive (See my point above). In most of cases, there exist utility-fairness tradeoffs, where it might be not deceptive with utility gains.

**Presentation**

The presentation and organization is very good.

**Justification For Why Not Higher Score:**

This paper is not self-standing and does not reach the bar of ICLR.

**Justification For Why Not Lower Score:**

N/A

---

### Decision · Program_Chairs · 2024-01-16

Accept (poster)